# Positive feedbacks drive the Greenland ice sheet evolution in millennial-length MAR–GISM simulations under a high-end warming scenario

Chloë Marie Paice[1], Xavier Fettweis[2], Philippe Huybrechts[1]

[1]Earth System Science and Department of Geography, Vrije Universiteit Brussel, Brussels, Belgium
[2]Department of Geography, Laboratory of Climatology, SPHERES, University of Liège, Liège, Belgium

*Correspondence to*: Chloë Marie Paice (chloe.marie.paice@vub.be)

**Abstract.** Understanding the complex interactions between the Greenland ice sheet (GrIS) and the atmosphere is crucial for projecting its future sea level contribution. However, studying these interactions remains challenging, as it requires high-resolution climate or atmospheric models to be run over extended timescales before their influence on the ice sheet–climate system becomes evident. Therefore, in this study, we coupled an ice sheet model (GISM) with a regional climate model (MAR) and conducted millennial-length simulations. The simulations consist of a zero-way, a one-way, and a two-way coupled configuration, which were forced by the IPSL-CM6A-LR global climate model output under the SSP5-8.5 scenario until 2300 and extended until the year 3000 by randomly sampling the last 51 years of forcing. They represent the first coupled simulations of an ice sheet model (ISM) and regional climate model (RCM) that extend beyond the centennial timescale and allow us to assess the evolving role of ice sheet–atmosphere feedbacks. Our results reveal that the ice sheet evolution is determined by positive as well as negative feedback mechanisms, that act over different timescales. The main observed negative feedback in our simulations is related to changing wind speeds at the ice sheet margin, due to which the integrated ice mass loss differs by only 2.4 % by 2300 between the two- and one-way coupled simulations, regardless of the differently evolving ice sheet geometries. Beyond this time however, positive feedback mechanisms related to decreasing surface elevation, namely the melt–elevation feedback and changes in cloudiness and orographic precipitation, dominate the ice sheet–climate system and strongly accelerate the integrated ice mass loss in the two-way coupled simulation. As a result, by the end of the simulations, the ice sheet has almost entirely disappeared in the two-way coupled simulation, with a sea level contribution of 7.135 m s.l.e., compared to the significantly smaller contributions of 5.635 m s.l.e. and 5.122 m s.l.e. for the one-way and zero-way coupled simulations, respectively. This highlights the importance of accurately representing the ice sheet–atmosphere interactions for long-term assessments of the Greenland ice sheet and climate.

## 1 Introduction

As the second largest ice body atop the largest island on earth, the Greenland ice sheet (GrIS) comprises a volume of 7.42 m sea level equivalent (s.l.e.) and is one of the main contributors to global sea level rise, with an ice mass loss of 4892 ± 457 Gt

30 between 1992 and 2020 (Morlighem et al., 2017; Goelzer et al., 2017, 2020b; Fox-Kemper et al., 2021; Otosaka et al., 2023). As a result, it is one of the primary sources of uncertainty regarding future global and regional sea level projections, according to the Intergovernmental Panel on Climate Change Sixth Assessment Report (Fox-Kemper et al., 2021). This can mainly be attributed to major remaining uncertainties regarding ice sheet–climate interactions and feedback mechanisms, that will determine the ice sheet's long-term mass loss. Although many of these interactions and feedbacks have been

35 identified and characterized for some time, quantifying their effects remains challenging (Fyke et al., 2018). Moreover, for some of them, it is still unclear whether they function as positive feedbacks (i.e., amplifying effects) or negative feedbacks (i.e., dampening effects) in the context of ice mass loss.

A comprehensive overview of ice sheet–climate interactions is presented by Fyke et al. (2018). In this study, we focus

40 specifically on interactions between the ice sheet and the atmosphere, including changes in precipitation, winds, and cloudiness. Among these, the most prominent and well-characterized interaction between the ice sheet and atmosphere is the melt–elevation feedback, that arises because of the elevation-dependence of temperature. When the ice sheet surface melts, its surface elevation is lowered and the air temperature increases following the adiabatic lapse rate, thereby inducing even more melt. This amplifies ice mass loss and incorporating this effect in model simulations thus leads to higher projected sea

45 level rise (e.g. Edwards et al., 2014; Vizcaíno et al., 2015; Aschwanden et al., 2019; Le clec'h et al., 2019a; Delhasse et al., 2024).

However, it is not straightforward to represent this melt–elevation feedback in ISMs, since the changing topography of the GrIS can in turn influence the (local) atmospheric circulation and induce changes in the precipitation pattern. This can result

50 in a negative feedback effect, when the altered precipitation leads to increased accumulation and snowfall over the ice sheet interior (Ridley et al., 2005; Hakuba et al., 2012; Gregory et al., 2020). Alternatively, the precipitation changes can constitute a positive feedback effect, when the rising temperatures over the lowered ice sheet surface lead to an increased rain fraction (Feenstra et al., 2025). In many cases, the impact is more nuanced and varies regionally, since the precipitation is advected further inland where it contributes to accumulation, but decreases near the margin (Fyke et al., 2018). This underscores the

55 complexity involved in modelling precipitation and the ongoing lack of consensus regarding changes in precipitation patterns and their subsequent impact on the shrinking ice sheet.

Other ice sheet–atmosphere interactions include changes in cloudiness, as clouds can alter the surface energy balance and surface melt over the ice sheet in several ways, such as through alteration of incoming shortwave radiation, increased

60 longwave warming, or reduced longwave cooling. It has been reported, for example, that clouds can reduce melt (in the ablation zone with low albedo) by blocking the incoming solar radiation, which is the main driver of melt here (Hofer et al., 2017). Vice versa however, it has been shown that clouds can enhance the meltwater runoff over the GrIS by one-third compared to clear skies, by reducing surface radiative cooling (mainly at night) and impeding the meltwater to refreeze (Van

Tricht et al., 2016). Besides, these radiational effects not only vary substantially in space and across seasons, but they also strongly depend on the cloud properties (Bennartz et al., 2013; Van Tricht et al., 2016; Hofer et al., 2017; Lenaerts et al., 2020). It is therefore crucial to accurately incorporate these effects when looking into future climate conditions and mass balance over the GrIS.

Near the ice sheet margin, two types of winds have been shown to impact melt rates. Katabatic winds transport cooled, dense air from the elevated ice sheet interior towards the lower-lying margins, whereas barrier winds, that develop because of the temperature difference between the tundra and ice sheet surface, can cause high melt rates as they advect warm air from the tundra towards the ice sheet margin (van den Broeke and Gallée, 1996). A respective strengthening and weakening of these winds near the margin as a result of ice sheet retreat and changing slopes can thus mitigate melt (Le clec'h et al., 2019a; Delhasse et al., 2024). It is therefore important to consider such changing wind patterns, at the high spatial resolution of regional climate models, when studying the future GrIS evolution.

Consequently, RCMs are needed to represent ice sheet–atmosphere interactions, or changes in (local) atmospheric circulation in response to the evolving ice sheet slopes and (local) topography at high resolution (Fettweis et al., 2020). Meanwhile, an ice sheet model (ISM) is needed to represent the ice dynamics and evolving ice sheet topography that are not considered by the RCM. As a result, efforts are currently emerging to couple ISMs to RCMs. Yet, to date only two such coupled ISM–RCM studies have been performed, on the centennial timescale (Le clec'h et al., 2019a; Delhasse et al., 2024). On millennial timescales several studies have been conducted, though with atmospheric, global climate or earth system models of lower resolution or complexity (e.g. Ridley et al., 2005, 2010; Charbit et al., 2008; Robinson et al., 2012; Aschwanden et al., 2019; Gregory et al., 2020; Van Breedam et al., 2020; Feenstra et al., 2025). They therefore provide rather limited insights regarding the local surface–atmosphere interactions and their impact on the atmospheric circulation and ice sheet mass loss, that generally become relevant on multi-centennial to millennial timescales (e.g. Ridley et al., 2005; Robinson et al., 2012; Feenstra et al., 2025; Goelzer et al.,2025). Therefore, we coupled the Greenland Ice Sheet Model (GISM) with a high-resolution RCM, the Modèle Atmosphérique Régional (MAR), and performed millennial-length simulations to obtain a better understanding of the ice sheet–atmosphere interactions and potential feedback mechanisms over Greenland. It is the first time ice sheet–atmosphere interactions are accounted for using an RCM on the millennial timescale.

Our main objectives are to identify the impact of the above-mentioned ice sheet–atmosphere interactions, to identify whether they act as positive or negative feedback mechanisms on ice mass loss, and to assess their relative importance over time. To do this, we conduct three MAR–GISM simulations of different coupling complexity, forced by six-hourly outputs from the IPSL-CM6A-LR global climate model (Boucher et al., 2020) for the high-warming SSP5-8.5 scenario. They allow to

compare the impact of representing all ice sheet–climate interactions, representing the melt–elevation feedback in a parametrized way, or not representing any interactions.

## 2. Methods

In the following section, the models as well as their initialization are briefly described. The initialization of the coupled models is one of the most crucial steps for the coupled simulations, as these need to start from a fully equilibrated state to be free from model drift. Besides, the initialized system should represent the observed state as closely as possible for an accurate representation of ice sheet–atmosphere interactions and ensuing estimates of the GrIS contribution to sea level. We rely on the established assumption that the GrIS was in steady state with surface mass balance (SMB) for the period 1960 to

1990 (e.g. Hanna et al. 2005; Sasgen et al., 2012; Khan et al., 2015; Mouginot et al., 2019). This way we do not need any assumptions regarding the state of the ice sheet between 1990 and present, or a historical run preceding the future simulations as in Rahlves et al. (2025). Lastly, we explain the different coupling strategies for the simulations until the end of this millennium.

### 2.1 Greenland Ice Sheet Model

The Greenland Ice Sheet Model (GISM) is a three-dimensional thermomechanically coupled ISM, that can also account for the isostatic bedrock adjustment resulting from ice mass changes. Ice temperature is computed using a prognostic equation for the conservation of heat, that includes vertical heat conduction, three-dimensional advection and internal frictional heating due to ice deformation (Huybrechts et al., 1991; Huybrechts, 2002; Fürst et al., 2015). The model has 30 non-equidistant vertical layers, with refined grid spacing towards the bottom where vertical plane shearing is concentrated.

Though different approximations to the force balance equations governing ice flow can be considered, the presented simulations are performed using the higher-order approximation, which includes multilayer longitudinal stresses and lateral horizontal shearing. It is classified as a LMLa higher-order model or Blatter-Pattyn model (Blatter, 1995; Pattyn, 2003; Hindmarsch et al., 2004) and is described in detail in Fürst et al. (2011). It is complemented by a simplified equation to describe the basal resistance (called SR HO in Fürst et al., 2013) in the basal sliding formulation.

The geometric input for the model consists of the BedMachine v3 dataset for bedrock topography and ice sheet surface elevation or ice thickness (Morlighem et al., 2017). The dataset is upscaled to suit the needs of GISM and extended with ocean bathymetry from the IBCAO Version 3 dataset (Jakobsson et al., 2012) to cover the area of ice sheet expansion during the Last Glacial Maximum. In the horizontal direction, GISM is run on a 5 km uniform grid. The selected resolution is

125 determined by several factors, with the primary consideration being the timescale of the simulations. Besides, ideally the GISM resolution should not be too high with respect to the MAR resolution, to maintain a reasonable level of discrepancy between both model resolutions throughout the coupled simulations, and to facilitate the efficient initialization of the ice

sheet and coupled model into an equilibrium state resembling recent observations, represented here by the BedMachine v3 dataset (Morlighem et al., 2017). Lastly, given the ice sheet model resolution and the research focus on long-term ice sheet – atmosphere interactions, dynamic outlet glacier retreat is not explicitly considered here.

Compared to the previous two ice sheet models that were coupled to MAR, the advantages of GISM are its (new) possibility to combine a glacial-interglacial spin-up and data assimilation for its initialization, as explained below, and the fact that the higher-order model version can still run at relatively high resolution for the envisioned millennial timescale. Additionally, by coupling MAR with another ice sheet model, we can evaluate the robustness of the results compared to those obtained with GRISLI and PISM (Le clec'h et al., 2019a; Delhasse et al., 2024).

## 2.2 Modèle Atmosphérique Régional

The Modèle Atmosphérique Régional (MAR) is a hydrostatic RCM, that was specifically designed for the polar regions and has been calibrated and exhaustively evaluated over the GrIS (Fettweis et al., 2020). It has been widely used to simulate and reconstruct the SMB over the Greenland (Delhasse et al., 2020, 2024; Fettweis et al., 2017, 2020; Hofer et al., 2020) and Antarctic (Agosta et al., 2019; Amory et al., 2021; Kittel et al., 2021) ice sheets, as well as over Arctic land ice (Maure et al., 2023) and smaller ice caps such as Svalbard (Lang et al., 2015). The atmospheric component of the MAR model is a mesoscale primitive equation model, discretized on a non-staggered grid by applying higher-order numerical schemes (Gallée and Schayes, 1994). The atmospheric part of MAR is coupled to the one-dimensional Soil Ice Snow Vegetation Atmosphere Transfer (SISVAT) scheme (De Ridder and Gallée, 1998). The snow-ice component of SISVAT is originally based on the CROCUS snow model (Brun et al., 1992) and represents the surface energy balance by simulating processes related to surface albedo, snow metamorphism, meltwater percolation, retention, and refreezing. In this study, we use the MAR version 3.13, simply called MAR hereafter. Compared to version 3.11 described in Kittel et al. (2021), the main differences include corrections of bugs in the clouds scheme, a water mass conservation in the soil and snowpack at each timestep, and a continuous snowfall-rainfall limit between -1°C (full snow) and +1°C (full rain) near-surface temperature. In our coupled model set-up MAR provides the SMB and runoff as forcing for GISM and is run at 30 km, a relatively coarse horizontal resolution owing to the envisioned timescale of the simulations. To downscale the SMB and runoff onto the 5 km GISM grid, we apply the method developed by Franco et al. (2012) that is further explained in Sect. 2.4.

As an RCM, MAR requires (six-hourly) lateral forcing fields to accurately simulate the SMB and atmospheric conditions within the selected domain. Given our focus on feedback effects driving ice sheet decline, we force the coupled simulations with a high-warming climate scenario. Specifically, we use IPSL-CM6A-LR (Boucher et al., 2020) six-hourly outputs as large-scale forcing fields as it is one of the only global climate or earth system models for which six-hourly output is available until 2300 under the SSP5-8.5 scenario and its extension (Meinshausen et al., 2020). SSP5-8.5 is the highest-

emission scenario considered by the IPCC and assumes that peak $CO_2$ emissions are only reached by the end of the 21[st] century before linearly decreasing to zero by 2250.

## 2.3 Ice sheet model initialization

For the ISM initialization, we combine a glacial–interglacial spin-up with a data assimilation technique, to capture the ice
sheet response to past climatic conditions and represent its recently observed geometry as closely as possible. The glacial–interglacial spin-up is performed once to provide a three-dimensional ice temperature field for the GrIS that captures its long-term thermal history, as well as an initial velocity field for the first step of the data assimilation procedure (Sect. 2.3.2) that is iteratively repeated during the coupled MAR–GISM initialization (Sect. 2.4). The former is indispensable for reliable ice sheet simulations, as ice deformation is non-linearly dependent on ice temperature and basal sliding only occurs for those
parts of the ice sheet that are at the pressure melting point (Goelzer et al., 2017). However, as is often the case, the ice sheet geometry obtained from the glacial–interglacial spin-up is slightly oversized and too thick near the margin (Huybrechts, 2002; Greve et al., 2011; Robinson et al., 2011; Graversen et al., 2010; Aschwanden et al., 2013; Fürst et al., 2015; Van Breedam et al., 2020). Using this geometry as input for the coupled ice sheet–atmosphere simulations would introduce biases in the ice dynamics and modelled SMB (Goelzer et al., 2013; Fürst et al., 2015; Delhasse et al., 2024). Therefore, the second
part of the ice sheet initialization consists of a data assimilation procedure for ice thickness, to accurately represent the observed ice thickness at the start and thereby the ice sheet–atmosphere interactions throughout the simulations.

### 2.3.1 Glacial–interglacial spin-up

The glacial–interglacial spin-up approach is described in detail in Huybrechts (2002) and Fürst et al. (2015). During this spin-up, GISM is run with a freely evolving geometry over the last two glacial cycles from 225 000 BP until the present-day
in response to past precipitation rates, and temperature and sea-level anomalies derived from ice and marine sediment cores. The isostatic bedrock adjustment resulting from ice mass changes is enabled in GISM (Huybrechts, 2002). Though as it would further complicate the MAR–GISM coupled model initialization (Sect. 2.4), it is disabled afterwards. Similarly, the built-in SMB model based on the positive degree-day approach for ablation (Janssens and Huybrechts, 2000; Gregory and Huybrechts, 2006) is only used throughout the glacial–interglacial spin-up. With this approach, the amount of (energy
available for) melt is determined based on the sum of mean daily temperatures above 0°C, i.e. the number of positive degree days. Accumulation is considered as the fraction of precipitation falling when the temperature is below a certain threshold, denoted as the snow fraction limit, here 1°C. During the data assimilation and coupled simulations, GISM is forced directly with the SMB produced by MAR.

### 2.3.2 Data assimilation

The applied data assimilation technique relies on the rapidly converging iterative method of Le clec'h et al. (2019b), that was slightly adapted for our ISM and purpose. It consists of an optimization step, during which the basal sliding coefficient

(BSC) and enhancement factor (EF) are updated to match the modelled ice thickness to observations. This is followed by a relaxation step, during which the ISM is run in free geometry mode with the inferred BSC and EF fields, to minimize any remaining model drift. Throughout both steps, GISM is forced by the MAR SMB for our reference period (1961-1990). Both
steps are repeated until the root mean square error (RMSE) between the modelled and observed ice thickness no longer improves significantly (Le clec'h et al., 2019b). We opted to combine a relatively short optimization period (50 years) with a slightly longer relaxation period (300 years), since the optimization period is computationally expensive, especially for the higher-order ISM version.

Yet, there are some essential differences compared to the original approach. First of all, during the optimization step we optimize the basal sliding rather than the basal drag coefficient, starting from the reference value of $0.83*10^{-10}$ $m^2\,Pa^{-3}\,y^{-1}$ in GISM (Fürst et al., 2015). Secondly, we also optimize the ice viscosity or EF in Glen's flow law periodically and iteratively to improve the method and facilitate ice thickness optimization in frozen regions without sliding, as suggested by the authors (Le clec'h et al., 2019b). In addition, this two-dimensional optimization of the EF facilitates the ice thickness adjustment for
areas where the minimum and maximum allowed values for the BSC are reached. As opposed to Le clec'h et al. (2019b), the constrained min-max range is increased stepwise for every iteration of the optimization step, as this restricts the magnitude of change for the BSC within one iteration and prevents a noisy pattern as well as extreme values, that are unfavourable for obtaining a stable ice sheet. Besides, for reasons of numerical stability, the three-dimensional temperature field obtained from the glacial–interglacial spin-up is held constant both throughout the data assimilation and coupled simulations. After
the data assimilation, in absence of a better approach, the optimized basal sliding coefficient and enhancement factor are held constant throughout the coupled simulations, as is the geothermal heat flux. In general, these fixed parameters are justifiable for short-term projections but inevitably become more contestable over the course of time (e.g. Goelzer et al., 2013; Le clec'h et al., 2019a, 2019b). For the detached peripheral glaciers and ice caps surrounding the ice sheet, identified based on the PROMICE aerophotogrammetric map of Greenland ice masses (Citterio and Ahlstrøm, 2013), the data assimilation was
not performed. For these areas the observed ice thickness was adopted and SMB anomalies were applied throughout the coupled simulations.

## 2.4 Coupled MAR–GISM initialization

To obtain an equilibrated ice sheet–climate system, the ice sheet provided by GISM should be in steady state with the MAR (1961 to 1990) SMB, which in turn should be computed on the ice sheet topography as simulated by GISM. The
initialization is thus an iterative process, during which MAR and GISM are initialized repeatedly for the period 1961 to 1990 and exchange information (SMB and ice sheet topography, respectively) at every turn, until the differences in SMB and ice sheet topography between two consecutive initializations become insignificant.

At the start of the MAR–GISM initialization, MAR is forced by IPSL-CM6A-LR (historical scenario) and once run
continuously for the period 1950 to 1990 on the recently observed topography (Morlighem et al., 2017) to stabilize the
snowpack in the model. The resulting annual SMB is used to compute the mean SMB for the 1961 to 1990 reference period,
which is passed to GISM to perform the data assimilation (Sect. 2.3.2). The obtained updated GISM ice sheet topography is
passed back to MAR for the next iteration, after being aggregated onto the coarser 30 km MAR grid by weighted averaging
of the four nearest neighbours. The fraction of tundra versus ice in every MAR grid cell thus depends on the number of
corresponding ice-covered GISM grid cells. Conversely, when the MAR SMB is passed to GISM, it needs to be downscaled
onto the finer GISM grid. This is done by first interpolating the MAR SMB onto the GISM grid, using a nearest-neighbour
distance-weighted method, and applying an additional correction to account for the topographic spatial variability on the
higher-resolution GISM grid. For this, we adopt the method developed by Franco et al. (2012) which consists of applying a
correction for every higher resolution (GISM) grid cell based on the vertical SMB–elevation gradient of the nine surrounding
lower resolution (MAR) grid cells. The procedure is illustrated step-by-step in Wyard (2015), and Delhasse et al. (2024)
where it was named offline correction. It will be referred to as offline extrapolation hereafter.

## 2.5 Coupled simulations

### 2.5.1 Two-way coupled simulation

As illustrated in Fig. 1, similar to Le clec'h et al. (2019a) and Delhasse et al. (2024), three different coupling types between
the ice sheet and the RCM were considered: a so-called two-way (2wC), a one-way (1wC), and a zero-way coupled (0wC)
simulation. The first year of these simulations is 1990. In the 2wC simulation, GISM is forced with SMB and runoff from
MAR, which in turn operates on the changing ice sheet geometry. This is the only way to explicitly consider ice sheet–
atmosphere interactions such as the melt–elevation feedback, the melt–albedo feedback, and changing patterns of
precipitation. The information exchange between both models occurs annually. More specifically, with the offline
extrapolation (Sect. 2.4) the 30 km MAR SMB and runoff are extrapolated onto the 5 km GISM grid and used as input to run
GISM for one year. The ice sheet topography is annually updated in GISM for all three coupled simulations, but the glacial
isostatic bedrock adjustment is not considered (Sect. 2.3.1). In the 2wC simulations, the updated GISM topography is then
aggregated on the MAR grid (Sect. 2.4) and serves as input to run MAR for one year together with the atmosphere and
snowpack states from the previous MAR year.

### 2.5.2 One- and zero-way coupled simulations

The 1wC simulation functions similarly, except that the changing GISM geometry is not communicated to MAR. In other
words, the ice sheet topography in MAR remains fixed throughout the entire simulation. However, the 30 km MAR SMB is
still annually corrected for the changes in the GISM topography every year using the offline extrapolation (Fig. 1). As such,
the melt–elevation feedback is considered implicitly.

In the 0wC simulation, no corrections are made for elevation changes, hence none of the feedbacks between the ISM and the RCM are considered and this simulation thus represents the system's response to climate forcing. Nevertheless, we could not entirely omit the offline extrapolation for this simulation, as it was applied throughout the iterative MAR–GISM initialization. Therefore, during the 0wC simulation the MAR SMB is always extrapolated onto the (fixed) initialized 5 km GISM topography, instead of onto the annually updated GISM topography.

### 2.5.3 Control simulation

Lastly, the control simulation consists of running GISM with the fixed MAR 1961 to 1990 mean SMB and runoff obtained at the end of the coupled initialization (Sect. 2.4). As such, atmospheric changes are excluded and this simulation quantifies the remaining model drift of GISM with respect to the MAR 1960 to 1990 mean SMB, i.e. the remaining drift of the equilibrated coupled models.

### 2.5.4 Prolongation until 3000

For all three coupled simulations, after 2300 the GCM forcing for MAR is held constant until the year 3000 by randomly sampling the yearly IPSL-CM6A-LR output from the period 2250 to 2300, during which the forcing temperature stabilizes (Fig. 3). In any case, as stressed by Delhasse et al. (2024), even when the temperature stabilizes it is important to repeat the GCM forcing fields for MAR to prolong the 1wC and 0wC simulations, rather than repeating the MAR output (SMB and runoff), since the meltwater retention capacity and thereby the ablation area and SMB still require several decades to stabilize once warming stops.

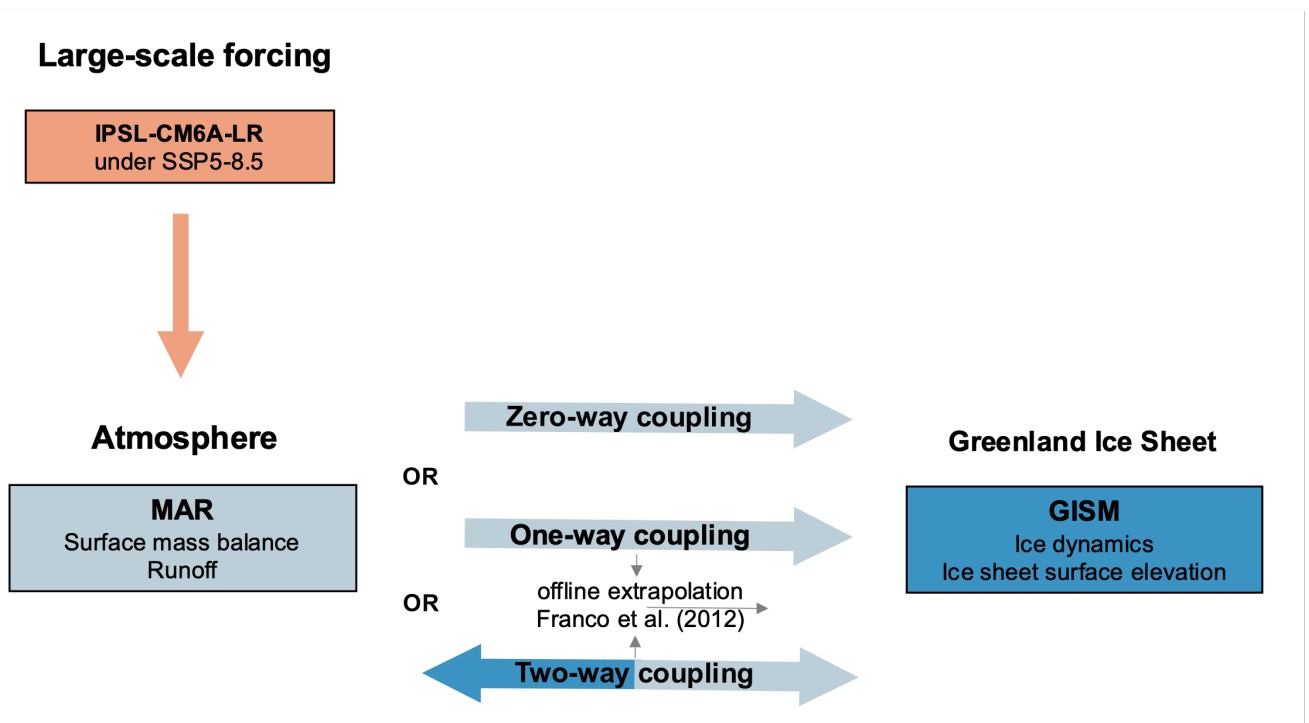

**Figure 1: Overview of the components of the coupled ice sheet–atmosphere model and their interactions. The model is self-consistent in that climatic fields are transferred directly and the forcing for the atmospheric component is also used during the ISM as well as the RCM initialization.**

## 2.6 Presentation of results

For the conversion of ice volume change to sea level contribution since the start of our simulations in 1990, we consider only the ice above floatation (Goelzer et al., 2020a), include volume changes from the peripheral glaciers and ice caps, assume an ice density of 916.7 kg m$^{-3}$ and apply a conversion factor of 361.8 Gt per millimetre of barystatic sea level rise (Morlighem et al., 2017). For all figures, it is indicated in the subscript on what grid and mask the variables are depicted. Where possible, the results over the 5 km GISM grid are shown, but since only SMB and runoff are extrapolated onto the 5 km GISM grid

throughout the simulations, most MAR variables are shown over their respective 30 km MAR grid. For most variables, as indicated in the figure subscript, the 30 year running mean is shown. Apart from the ice sheet's contribution to sea level, all values mentioned throughout the text refer to these running means. Besides, it should be noted that the MAR ice masks in the 1wC and 0wC simulations remain fixed over time, while the MAR ice mask in the 2wC simulation retreats. As such, it is always indicated whether the values refer to those over the fixed or retreating ice mask. Lastly, it should be kept in mind that

all climatic changes in the 2wC simulation after 2300 are due to local ice sheet–atmosphere feedbacks, as the large-scale forcing (i.e. the climate) remains constant after this time (Sect. 2.5.4).

## 3. Results

### 3.1 Initialized ice sheet topography

The ice sheet topography from the fifth MAR-GISM initialization iteration is used as the initial topography for the coupled simulations starting in 1990. As can be seen in Fig. 2, the differences between the initialized and observed topography (Morlighem et al., 2017) are generally less than 40 m and up to several hundred metres around the steep margin, where a minor misalignment of slope can cause large differences between the modelled and observed ice thickness. Besides, the observed topography is very irregular around the ice sheet margins and the central east in particular, which is marked by coastal mountain ranges with deep valleys in between, and an ensuing high spatial variability in ice thickness. As also reported by Le clec'h et al. (2019b), it is therefore most difficult to obtain a well-constrained ice thickness in these areas. The remaining drift of the coupled models as represented by the control simulation (Sect. 2.5.3) is -0.27 mm s.l.e. by 2100, 1.20 mm s.l.e. by 2300, 3.60 mm s.l.e. by 2500, and 12.78 mm s.l.e. by 3000, the end of the simulation, hence almost negligible (Fig. 3).

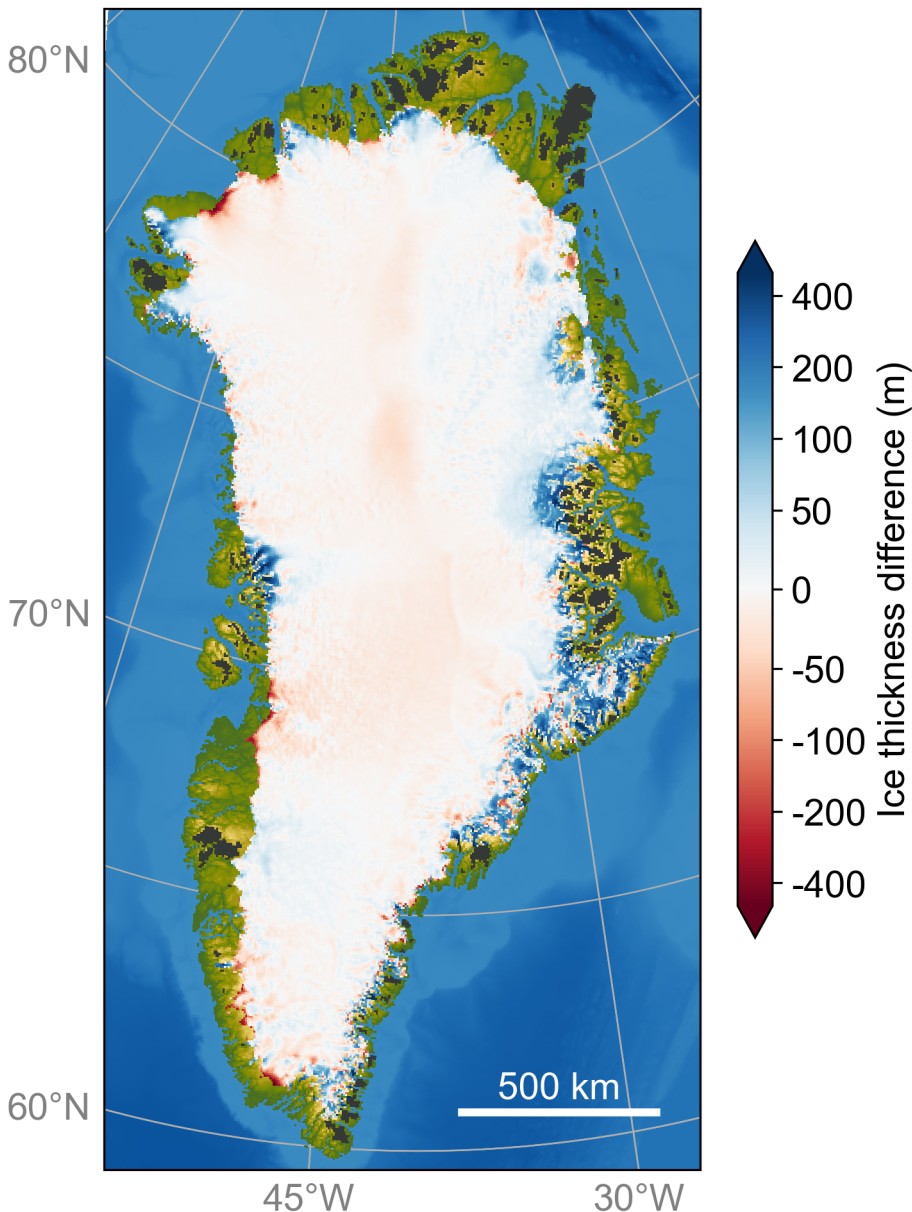


**Figure 2: Initialized minus observed ice thickness (Morlighem et al., 2017). The dark grey areas represent the detached peripheral glaciers and ice caps for which the data assimilation was not performed and the observed topography was adopted. Note that the applied colour scale is logarithmic.**

## 3.2 Sea level contribution under a high-warming scenario

As displayed in Fig. 3, the predicted mean warming by IPSL-CM6A-LR for the SSP5-8.5 scenario over the larger Greenland region (60-80° N, 20-80° W) at 700 hPa rises by +14.93°C, from -16.27°C to -1.34°C, between 1990 and 2300. This leads to an increase in 2 m air temperature of + 14.79°C by 2300 in MAR, over the retreating ice mask. The resulting ice sheet contribution to sea level by 2300 is 2.201 m s.l.e., 2.149 m s.l.e., and 1.732 m s.l.e. for the 2wC, 1wC, and 0wC simulations, respectively. Of this contribution 0.0258 m s.l.e. can be attributed to the peripheral glaciers and ice caps, that have disappeared entirely by 2200 in all simulations.

Regardless of the constant climate forcing after 2300 (Sect. 2.5.4), in MAR the 2 m air temperature over the remaining ice mask in 3000 rises by another +8.55°C as the ice sheet surface elevation further decreases (Fig. 3). By 3000, the mean annual temperature over the remaining ice sheet has thus risen to -1.73°C. Besides, the ice sheet contribution to sea level further increases for all simulations, indicating that even in the 1wC and 0wC simulations the ice sheet does not reach a new equilibrium with the MAR SMB under this high-warming scenario. By 2500, the ice sheet contribution to sea level is 4.330 m s.l.e. for the 2wC simulation, 3.787 m s.l.e. for the 1wC simulation, and 3.107 m s.l.e. for the 0wC simulation. By the year 3000, for the 1wC and 0wC simulations the contribution rises to 5.653 m s.l.e. and 5.122 m s.l.e., respectively. Yet in the 2wC simulation, the ice sheet has disappeared almost entirely with a contribution to sea level of 7.135 m s.l.e., as illustrated by Fig. 3 and Fig. 4. The most important processes explaining this evolution are described in the following subsections.

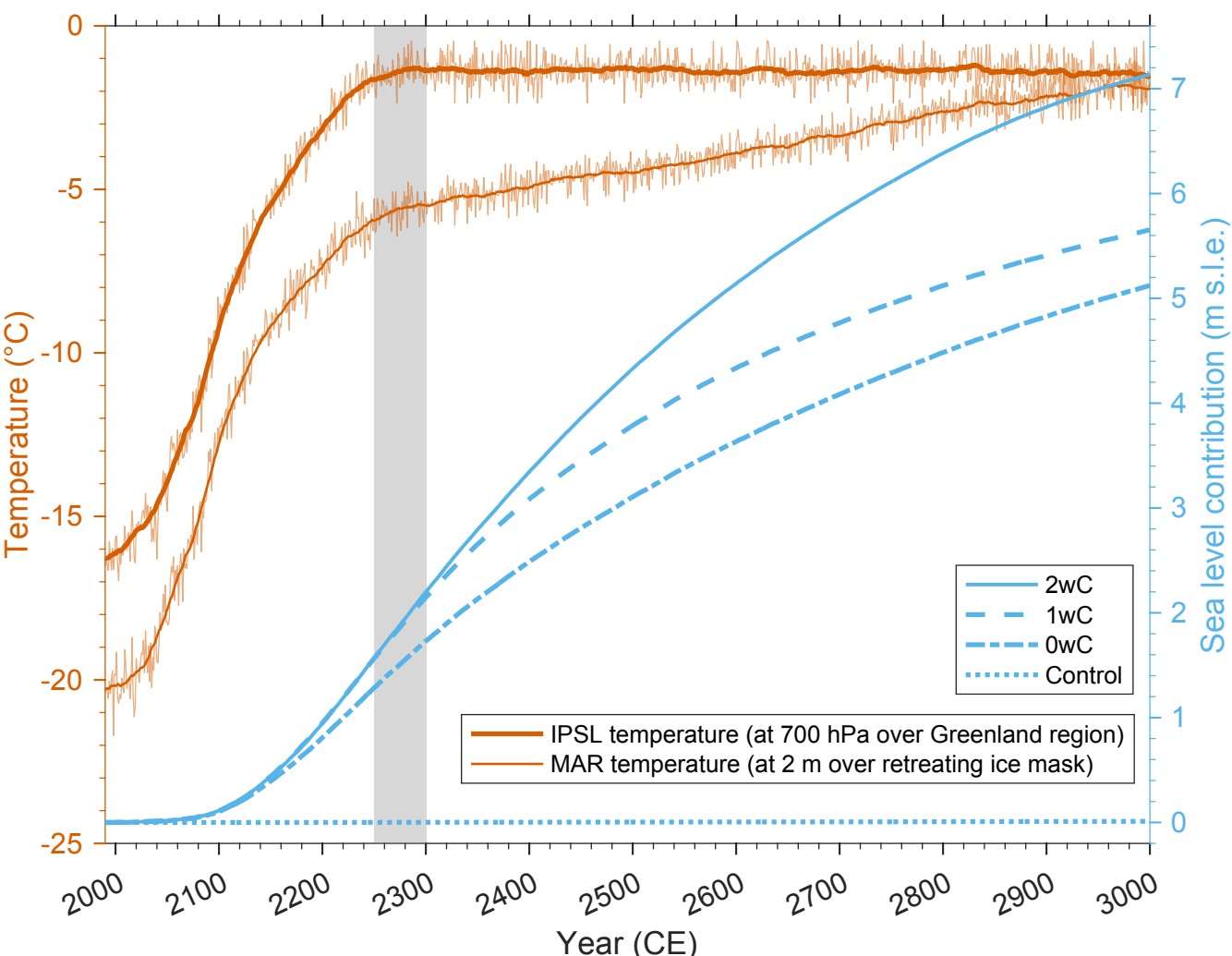

**Figure 3:** The IPSL-CM6A-LR mean annual air temperature at 700 hPa under the SSP5-8.5 scenario over the larger Greenland region (60-80° N, 20-80° W), as well as the MAR 2 m mean annual air temperature over the retreating ice mask (in °C), i.e. for the 2wC simulation, are shown in orange on the left axis. Thin lines depict the mean annual temperatures, thick lines their 30 year running mean. The grey area indicates the years that were randomly sampled (2250 to 2300) to prolong the climate forcing after 2300. The corresponding annual GrIS contributions to sea level (in m s.l.e.) for the two-way (2wC), one-way (1wC), zero-way coupled (0wC), and the control simulation are shown in blue on the right axis.

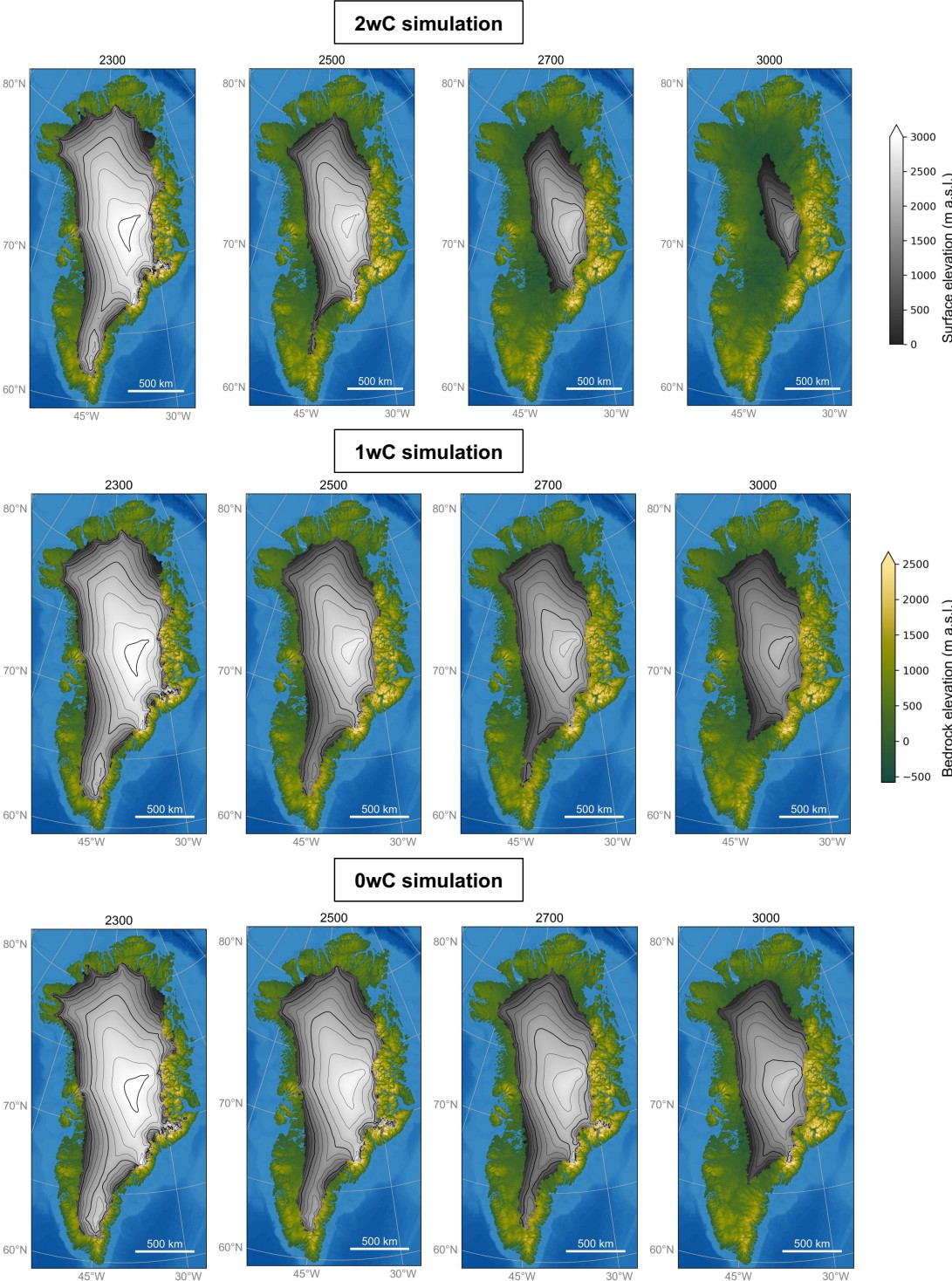

**Figure 4: Evolution of the GrIS topography for all three coupled simulations. The ice sheet surface elevation is displayed at several points in time on the 5 km GISM grid, in grey tones with additional contours plotted every 250 m (thin lines) and 1000 m (thick lines).**


### 3.3 Changing wind speeds and reduced ablation at the ice sheet margin

During the first three centuries, the ice sheet–climate evolution in the simulations is dominated by the strong warming predicted by IPSL-CM6A-LR under the SSP5-8.5 scenario (Fig. 3). The mean temperature, surface elevation, SMB, and hence the integrated ice mass loss over the ice sheet only start to diverge around the year 2150 between simulations and the

ice sheet contribution to sea level is thus quite similar for all simulations up to 2300. However, this does not imply an absence of feedback effects, as the SMB reveals clear spatial differences between the simulations, despite this similar integrated ice mass loss (Fig. 5a-c). Especially around 2200 it becomes clear that a spatial compensation of differences within the SMB fields is at play, with ablation in the 1wC simulation being overestimated (by 1 to >3 m w.e.) compared to the 2wC simulation within 60 km from the ice sheet margins (i.e. two MAR grid cells), referred to hereafter as the ice-

marginal zone, and slightly underestimated over the interior compared to the 2wC simulation (by generally 10 to 20 mm w.e.). However, as Fig. 5 (d-e) demonstrates, the magnitude and sign of the resulting SMB difference between the 1wC and 2wC on the 5 km GISM grid not only vary spatially but also vary over time. On the other hand, on the 30 km MAR grid, the SMB is always more negative for the 2wC simulation and the difference with the 1wC simulation is ever larger over time. In other words, the over- and (weaker but more widespread) underestimation of ablation on the 5 km GISM grid can be linked

to the offline extrapolation that falls short at the ice sheet margins over time. Notably, in the 0wC simulation, the ablation is always underestimated with respect to the 2wC simulation, as every year the SMB is extrapolated onto the (fixed) initialized GISM topography rather than the updated GISM topography.

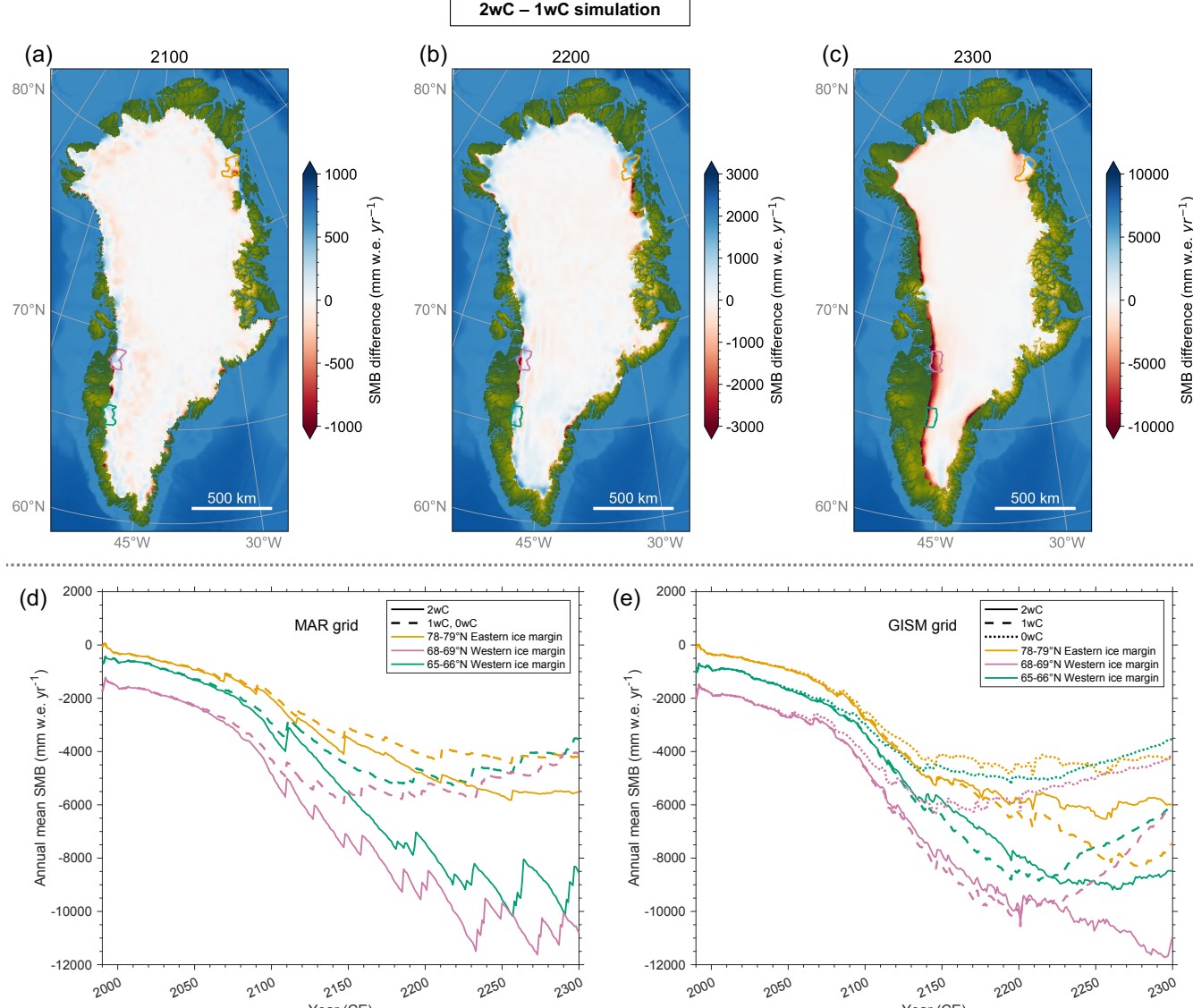

**Figure 5: Annual mean SMB differences between the 2wC and 1wC simulations in 2100, 2200 and 2300 on the 5 km GISM grid, over the remaining 2wC ice mask (a-c). Note the differing scale for the individual panels. For the three outlined regions along the 60 km broad retreating ice sheet margin, the annual mean SMB per region over time for the 2wC, 1wC, and 0wC simulations is shown in (d-e). All depicted values consist of the 30 year running means over the retreating ice mask. Note that the strong stepwise variations in SMB over the 30 km MAR grid are due to the retreating ice mask. As the MAR topography and ice mask remain fixed throughout the 1wC and 0wC simulations, the SMB over the 30 km MAR ice mask is equal for both simulations.**

This shortcoming of the offline extrapolation can mainly be attributed to changes in the boundary layer atmospheric circulation (wind) in the 2wC simulation with respect to the 1wC and 0wC simulations with fixed MAR topography. Similar to Delhasse et al. (2024), over time we observe higher wind speeds over the remaining ice sheet in the 2wC simulation

compared to the 1wC and 0wC simulations with fixed MAR topography (Fig. 6). Given the identical large-scale forcing for all simulations and the absence of barrier winds over the ice sheet interior, the higher wind speeds are of katabatic nature and are the result of a retreated ice sheet geometry with stronger slopes further inland compared to the 1wC and 0wC simulations with fixed MAR topography. Secondly, we observe decreased wind speeds within the (30 km to 60 km broad) ice-marginal zone and over the new tundra in the 2wC simulation compared to the fixed topography simulations. This is the result of a

decrease in katabatic winds, since the 2wC ice sheet has retreated here, and of a decrease in barrier winds as was also observed by Delhasse et al. (2024). By 2300, the maximal increase in (katabatic) wind speed is 0.8 m s$^{-1}$ and the maximal decrease in (katabatic and barrier) wind speed is -3.8 m s$^{-1}$. The strongest decrease over the remaining ice sheet occurs along the northeast margin between 78°N and 80°N, where north-westerly, westerly and south-westerly winds converge due to the locally concave ice sheet topography. This coincides largely with the outlined zone in Figure 5 for which the offline

extrapolated SMB on the GISM grid remains more negative by 2300 for the 1wC simulation compared to the 2wC simulation, and where the ice sheet on the GISM grid has therefore retreated faster in the 1wC simulation (Fig. 4). The changing wind speeds thus seem to impact the surface energy balance (sensible and latent heat fluxes) in a way that is not captured by the offline extrapolation.

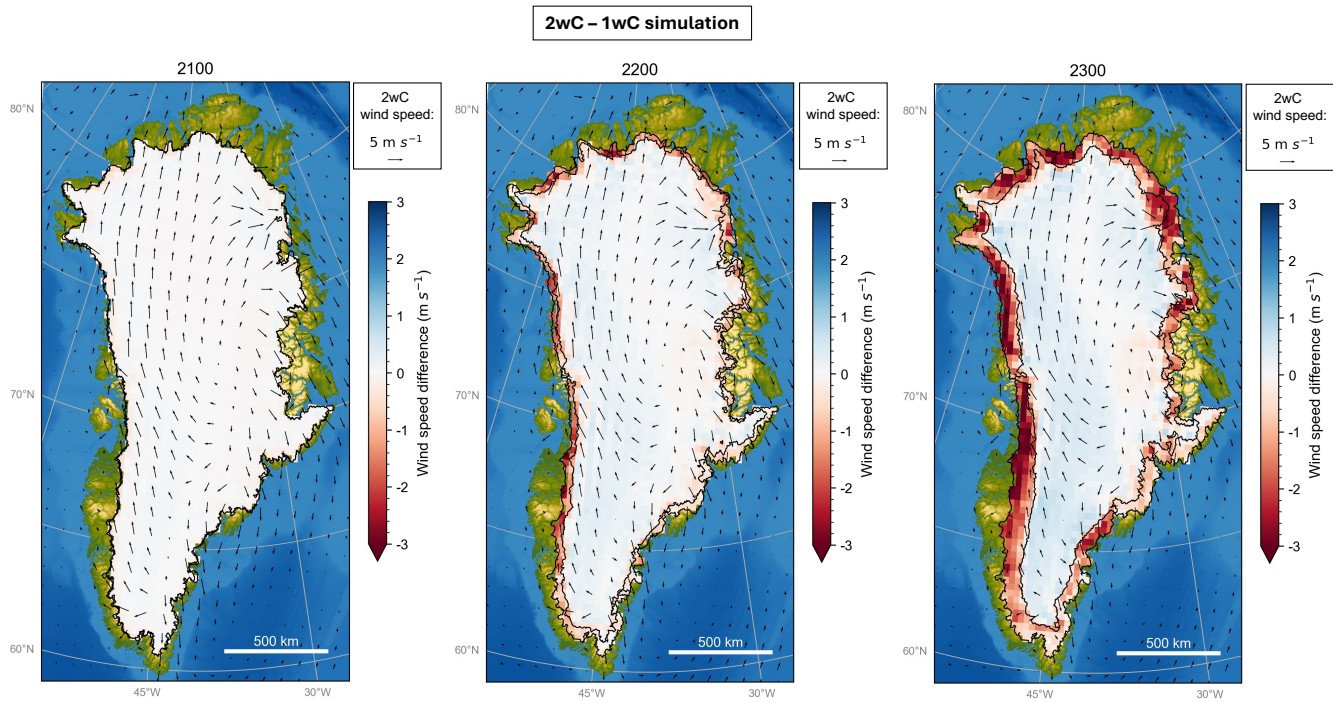


**Figure 6: Annual mean wind speed differences (m s$^{-1}$) between the 2wC and 1wC or 0wC simulations with fixed MAR topography in 2100, 2200 and 2300. The overlying arrows illustrate the wind speed magnitude and direction in the 2wC simulation for each time frame. All values are shown on the 30 km MAR grid and depict the 30 year running means. The black contours indicate the remaining, as well as the initial ice mask (i.e. identical to the fixed ice mask for the 1wC and 0wC simulations) on the 5 km GISM**
**grid.**

The impact of the changing wind speeds on the surface energy balance can be demonstrated by a transect from the ice sheet interior to the margin (Fig. 7). It shows that the changing wind speeds lead to lowered runoff and a reduced or even inversed SMB–elevation gradient in the ice-marginal zone in the 2wC simulation. Figure 8 disentangles the impact of these changing wind speeds on the different radiation balance components. Moreover, the increased katabatic winds lead to an accumulation of cold air along the ice-marginal zone (slightly decreasing temperature and strongly decreasing sensible heat flux) and potentially clouds (continuously increasing long- and decreasing shortwave downward radiation towards the margin), as well as deposition or condensation onto the ice sheet behind the ice-marginal zone (higher latent heat flux). Decreasing barrier winds on the other hand lead to reduced runoff in the ice-marginal zone. The strength of the SMB–elevation gradient inversion in the ice-marginal zone varies spatially, with stronger inversions along the western ice sheet margin, but a generally weaker or even absent inversion along the eastern margin as a result of the mountainous bedrock (not shown here). In addition, Fig. 7 demonstrates that even throughout the 2wC simulation, the extrapolated SMB on the 5 km grid is too negative compared to the SMB on the original 30 km grid when the grid cell is at the ice sheet margin (reduced colour saturation) and slightly overestimated when it is part of the ice sheet interior (full colour saturation). In other words, the offline extrapolation reproduces the effect of the inversed SMB–elevation gradient in the ice marginal-zone, but it does not fully capture the negative feedback effect of changing wind speeds on ablation along the margins, where the topographic differences between the MAR and GISM grid are largest. For the 1wC and 0wC simulations, this effect is further amplified as the differences in surface elevation between the (fixed) 30 km MAR topography and (changing) 5 km GISM topography keep increasing throughout the simulation. This explains the over- and (slight) underestimation of ablation with respect to the 2wC simulation and the ensuing similar ice sheet contribution up to 2300. After 2300, this negative wind feedback at the margins persists as Figures 7 and 8 show, but over time nevertheless the positive melt–elevation feedback and related changes in the atmospheric circulation amplify the ice mass loss everywhere in the 2wC simulation, as explained in the next sections.

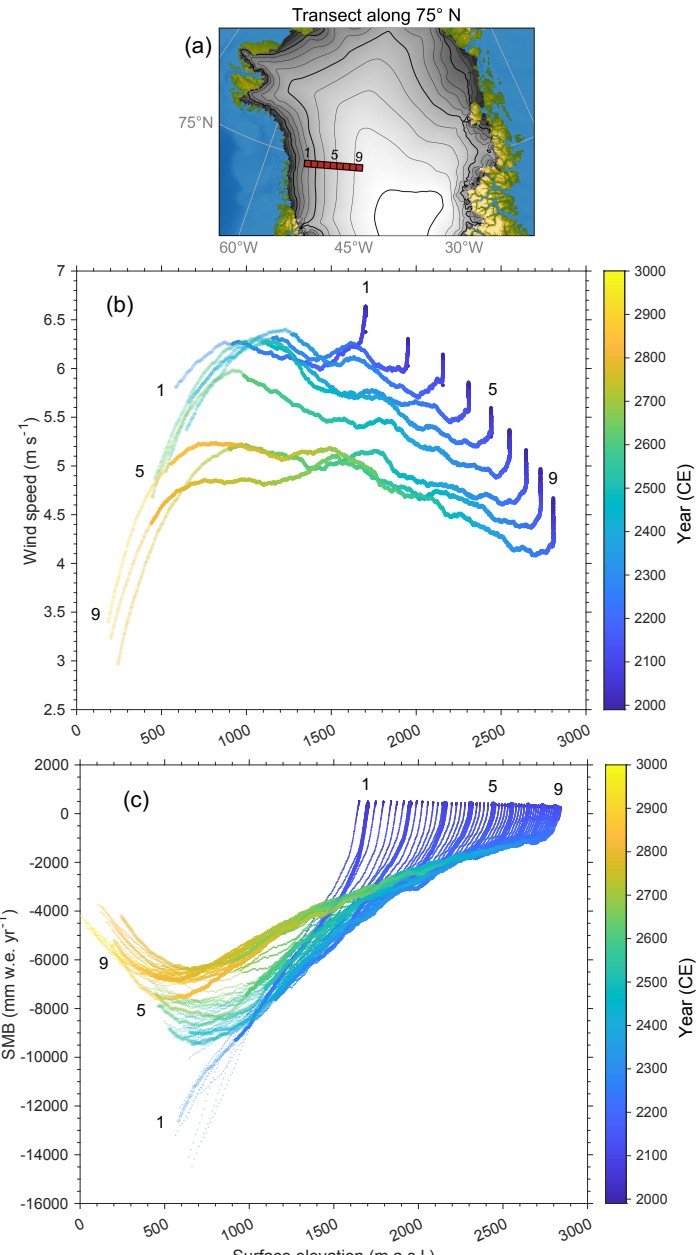

**Figure 7: Transect of nine adjacent grid cells at the western margin of the ice sheet along 75° N (a) for which scatter plots over time of annual mean wind speed (b) and annual surface mass balance (c) against surface elevation are shown. Depicted are the 30 year running means for the entire duration of the 2wC simulation (1990 to 3000). Note that up to 2300, the changes are due to the strong climatic warming. For the SMB (c), both the values for the 30 km MAR grid cells as well as the extrapolated values for the**

420 **corresponding 5 km GISM grid cells (small markers) are shown. Note that the number of corresponding GISM grid cells is higher than the number of MAR grid cells, as for each 30 km MAR grid cell roughly six adjacent corresponding 5 km GISM grid cells are depicted. For all variables, the reduced colour saturation indicates when the depicted grid cells are within 30 km from the retreating ice mask margin (i.e. adjacent to the first tundra grid cell).**

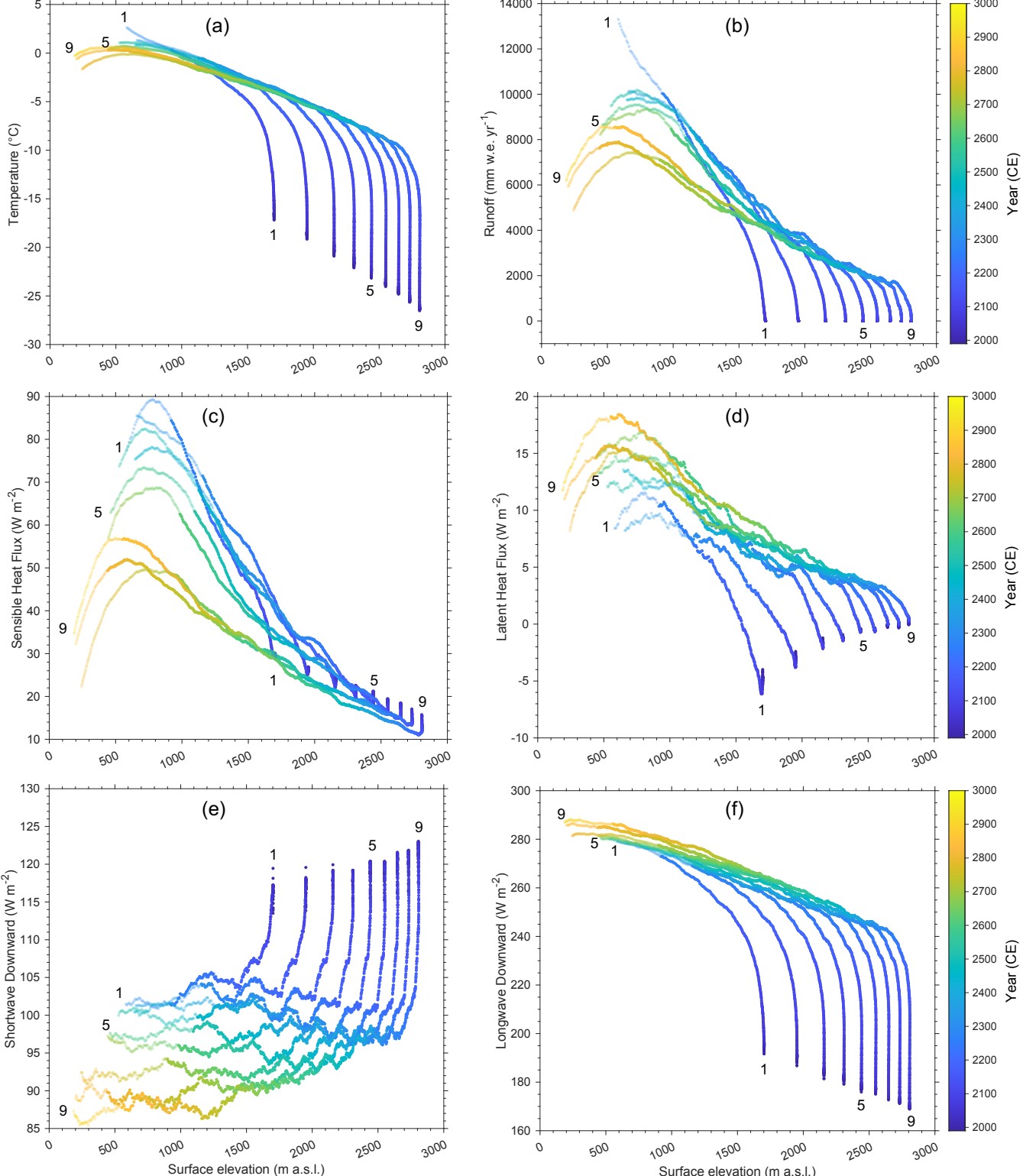

**Figure 8: Scatter plots of atmospheric variables over time against surface elevation for the transect of nine adjacent grid cells at the western margin of the ice sheet along 75° N (as in Fig. 7). Depicted are the 30 year running means for the entire duration of the 2wC simulation (1990 to 3000). The displayed variables are annual mean temperature (a), annual total runoff (b), annual mean sensible (c) and latent heat flux (d), as well as annual mean shortwave (e) and longwave downward radiation (f). As in Fig. 7, for all variables, the reduced colour saturation indicates when the depicted grid cells are within 30 km from the retreating ice mask margin (i.e. adjacent to the first tundra grid cell).**

### 3.4 Limited difference in melt–surface albedo feedback

Another reason for the very similar ice sheet contribution to sea level by 2300, is the similar mean SMB evolution for all simulations up to 2200 on both the MAR and GISM grids (Fig. 9). Besides, as Figure 10 shows, by that time the ablation area already covers 100 % of the ice sheet area in all simulations. This is also reflected in the density of the upper snow and ice layers up to 10 m depth in MAR that exceeds 910 kg m$^{-3}$ by 2200, indicating that most of the snowfall melts before densifying to firn in all simulations (not shown here). This implies that there is practically no difference in the positive melt–surface albedo feedback between all three simulations, as most of the ice sheet surface consists of bare ice during the Arctic summer in all three simulations (Fig. 11). The differences in albedo (that varies between 0.50 and 0.55 for bare ice in MAR depending on the presence of surface meltwater) and absorbed incoming solar radiation at the ice sheet surface between simulations are therefore negligible (Fig. 12). In winter the difference in snow cover or bare ice between the simulations is larger, but since there is hardly any incoming solar radiation due to the low solar elevation angle, this does not distinctly impact the absorbed energy at the surface.

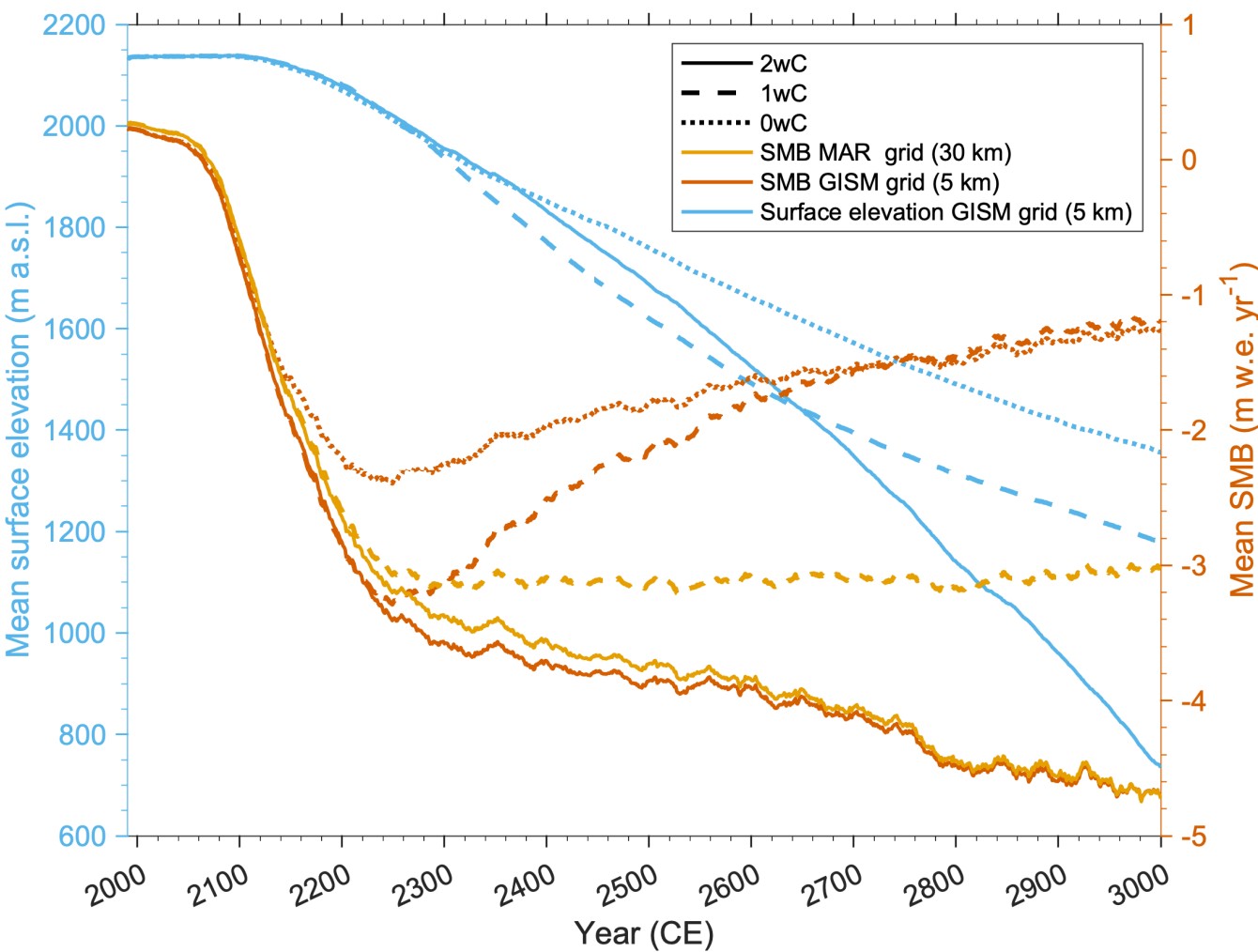

**Figure 9: The melt–elevation feedback components: annual mean ice sheet surface elevation over the 5 km GISM ice mask (left axis), and 30 year running mean SMB on the 30 km MAR grid as well as the on the 5 km GISM grid (i.e. after offline extrapolation) (right axis). Note that as the MAR topography and ice mask remain fixed throughout the 1wC and 0wC simulations, the SMB over the 30 km MAR ice mask is equal for both simulations. Yet, the corresponding extrapolated mean SMB over the 5 km GISM ice mask increases for these simulations, as the GISM topography and ice mask retreat over time (Fig. 4).**


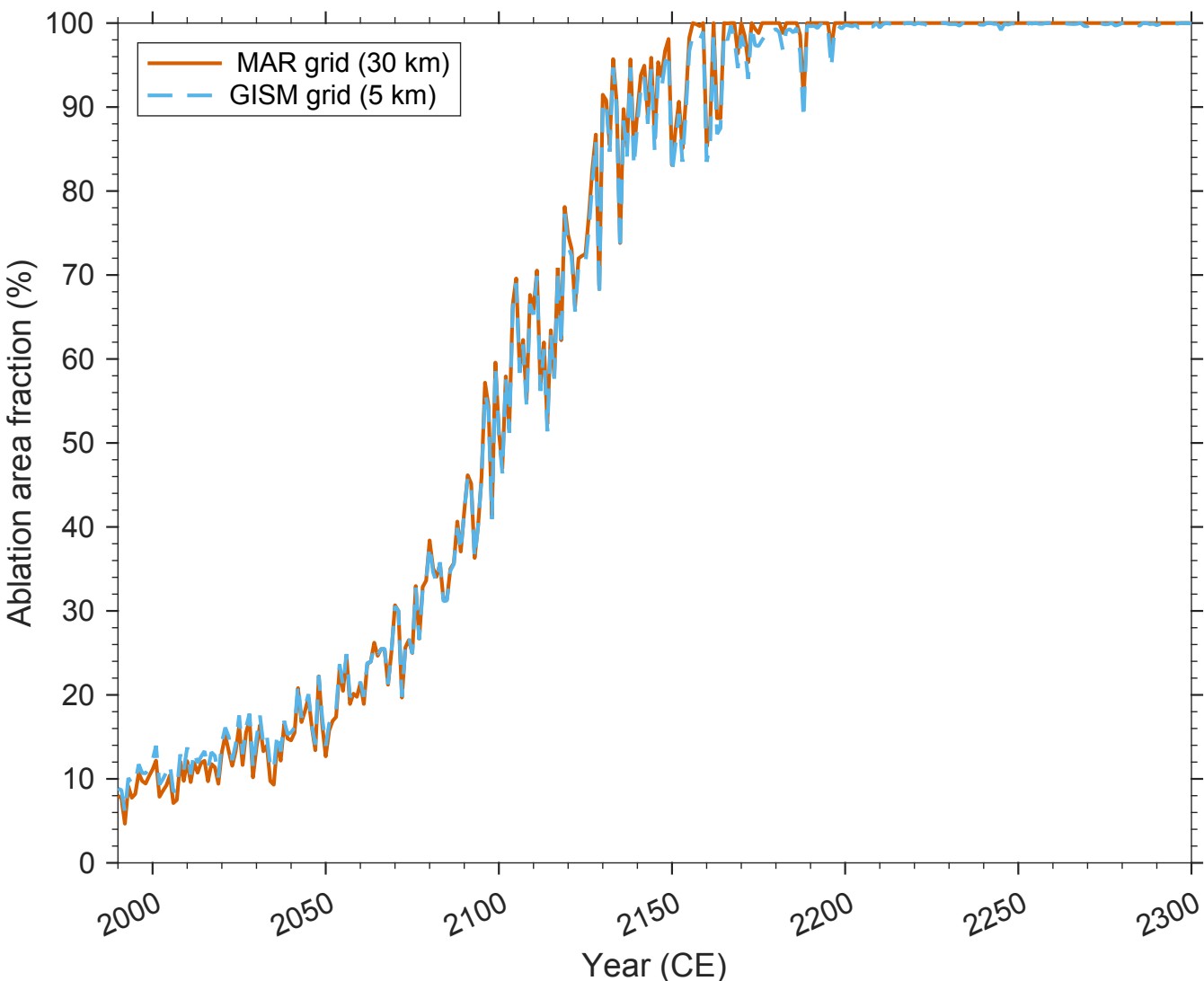

**Figure 10: Annual percentage of the ice sheet ablation area with respect to the total (retreating) ice sheet area. Specifically, on both the 30 km MAR and the 5 km GISM grid, every grid cell with annual mean SMB (original or extrapolated values, respectively) below -10 mm w.e. y⁻¹ was regarded as ablation area. Note that as the graphs for all three coupled simulations plot virtually on top of each other for both grids, only the ones for the 2wC simulation are shown here.**

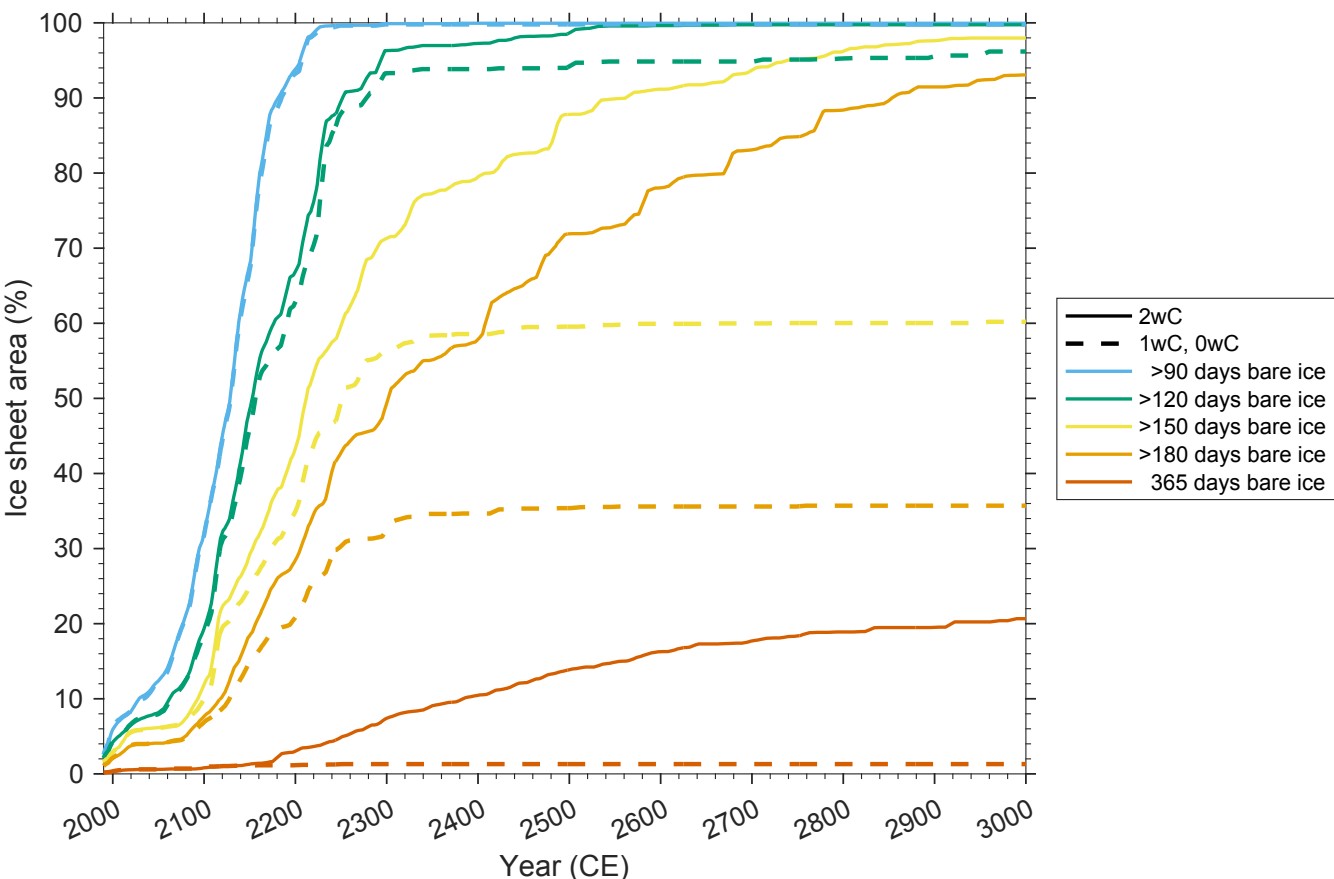

Figure 11: Area of the GrIS for which the MAR surface consists of bare ice (i.e. less than 5 cm of snow cover) during a certain number of days per year, as indicated by the colours. Depicted is the 30 year running mean over the 30 km MAR ice mask corresponding to each simulation. Note that the graphs are equal for the 1wC and 0wC simulations as the MAR topography and ice mask remain fixed throughout both simulations.

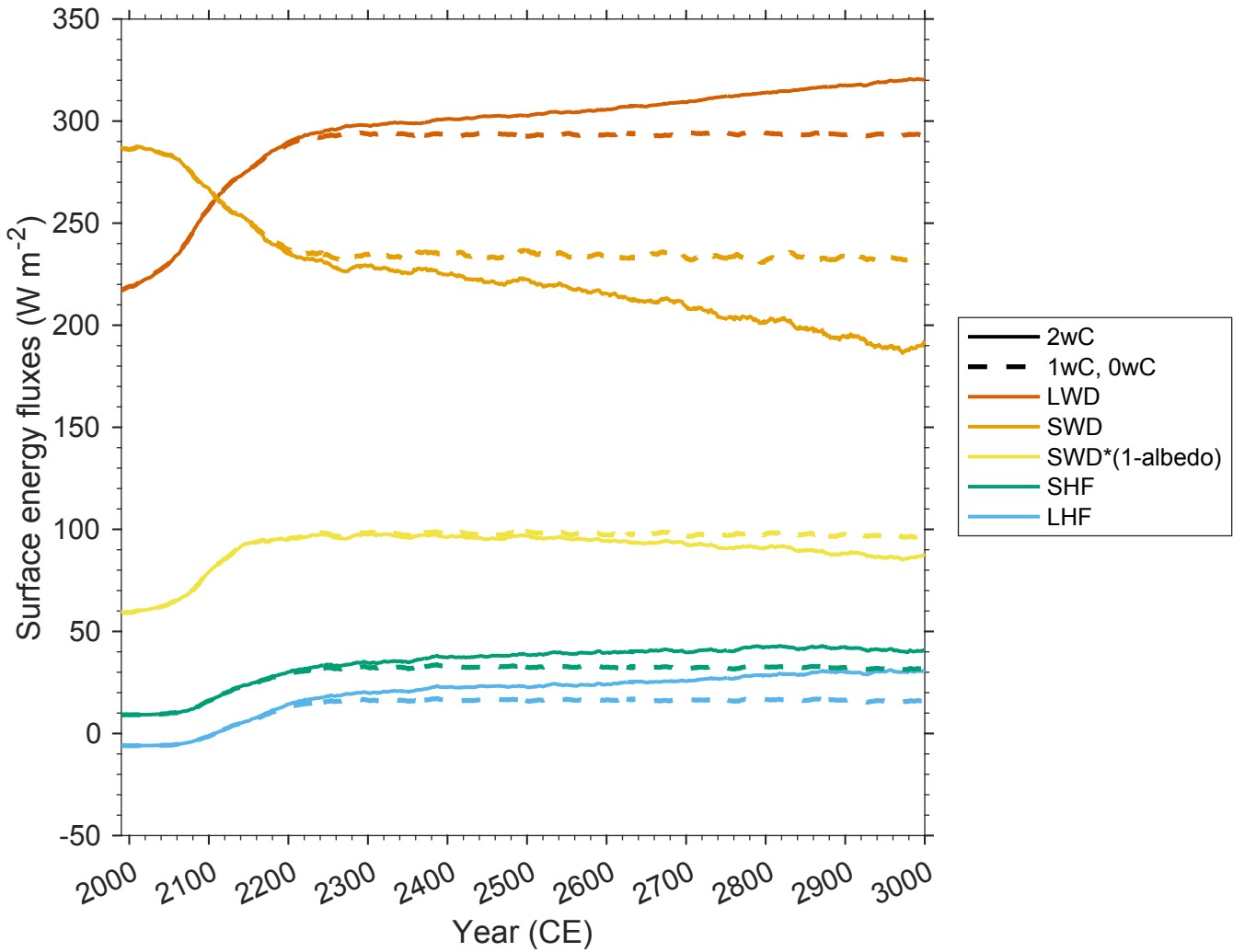


**Figure 12: Evolution of the mean summer (June-August) surface energy fluxes over the ice sheet: longwave downward radiation (LWD), shortwave downward radiation (SWD), absorbed proportion of the shortwave downward radiation (SWD*(1-albedo)), sensible heat flux (SHF), and latent heat flux (LHF). Specifically, the 30 year running means over the 30 km MAR ice mask corresponding to each simulation are shown. Note that the fluxes are equal for the 1wC and 0wC simulations as the MAR**
**topography and ice mask remain fixed throughout both simulations.**

### 3.5 The positive melt–elevation feedback

As illustrated by Figure 9, the ice sheet responds rather slowly to the rising temperatures and decreasing SMB at first, but after 2100 its mean surface elevation decreases almost linearly. Between 2250 and 2650 the mean surface elevation is lower
for the 1wC simulation compared to the 2wC simulation, because the low-lying ice margin retreats less rapidly in this simulation after 2250. The same applies to the 0wC simulation between 2150 and 2250, yet after this time the higher parts of

the ice sheet dwindle less rapidly than in the other two simulations. Besides, for the 1wC and 0wC simulations, SMB after 2300 is constant on the fixed MAR topography but becomes more positive on the 5 km GISM grid, as the GISM ice mask retreats. For the 2wC simulation, regardless of the constant climate forcing, around 2500 the decrease in mean surface
elevation accelerates and coincides with a continued rise in mean temperature and longwave downward (LWD) radiation over the ice sheet (Fig. 3 and Fig. 12). The latter is not only a direct effect of the decreasing surface elevation but can also be attributed to the increasing cloudiness over the ice sheet (Sect. 3.6.1). Besides, the acceleration in mean surface elevation decrease coincides with a strong reduction in snowfall and total precipitation for the 2wC simulation (Sect. 3.6.3). In essence, the melt–elevation feedback intensifies. At the time of this intensification, around 2500, the mean annual 2 m air
temperature over the remaining ice sheet area is -4.50°C (Fig. 3) and its mean surface elevation is 1687 m a.s.l (Fig. 9).

An in-depth analysis of the daily resolution MAR data reveals that after 2300 also the number of runoff days per year over the ice sheet (continues to) increase. Around 2500, an accelerated expansion can be observed in the ice sheet area exhibiting runoff during at least 120 days per year (Fig. 13). This explains the intensification of the melt–elevation feedback, as from
that time onwards, runoff is no longer restricted to the warmest three to four months over more than 40 % of the remaining ice sheet area. It is also consistent with the diminishing snow cover over the ice sheet, as around 2500, 100 % of the ice sheet surface consists of bare ice during at least 120 days each year. Yet, the areal expansion is more gradual than for the number of runoff days and does not exhibit an acceleration (Fig. 11).

Nevertheless, the proportion of June–August runoff remains 68 % or more of the annual total, compared to 94 % at the start of the simulations and 76 % by 2300 in the 2wC simulation. This is because even after 2300 the available energy during the Arctic winter months is hardly enough to melt the fresh snow layer over most of the ice sheet area. In other words, though the melt season keeps expanding, the number of bare ice and runoff days per year (Fig. 11 and 13) and therefore the strength of the melt–elevation feedback remain physically constrained by the low solar elevation angle.

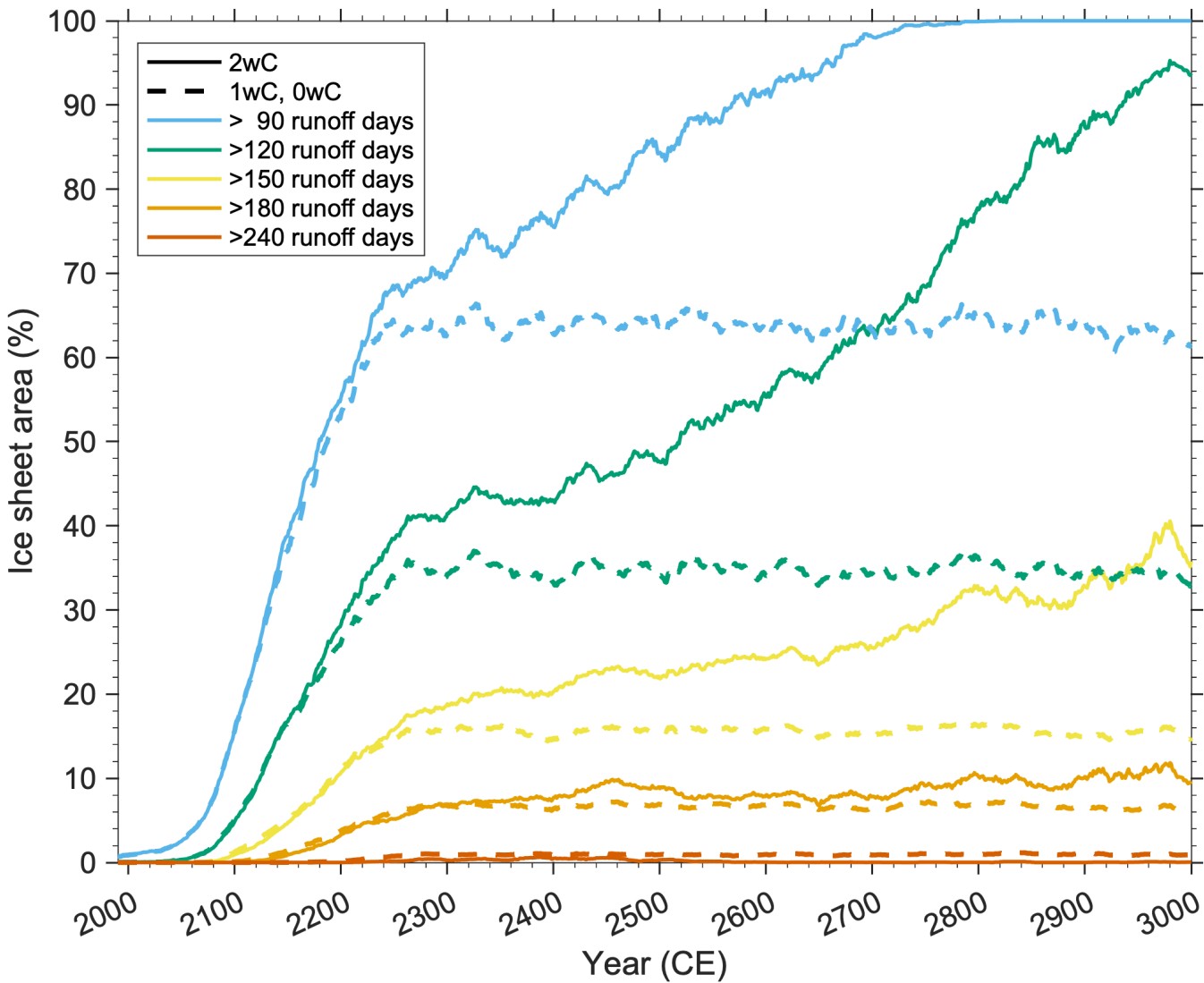

**Figure 13: Area of the GrIS for which runoff (more than 10 mm w.e. day$^{-1}$) in MAR occurs during a certain number of days per year, as indicated by the colours. Depicted is the 30 year running mean over the 30 km MAR ice mask corresponding to each simulation. Note that the graphs are equal for the 1wC and 0wC simulations as the MAR topography and ice mask remain fixed throughout both simulations.**

## 3.6 Additional positive feedbacks triggered by the decreasing surface elevation

Apart from an amplification of melt and runoff, the decreasing surface elevation also triggers self-enforcing changes in cloudiness and precipitation. Hence, though their impact in the 2wC simulation cannot be disentangled from the melt–elevation feedback, these changes in cloudiness and precipitation represent additional positive feedback effects over the ice sheet.

### 3.6.1 Increased cloudiness and the importance of cloud phase change

Regarding the increase in cloudiness, and the changing ratio of solid and liquid clouds with differing longwave emissivity, several feedback effects can be observed over time, that are stronger for the 2wC compared to the 1wC and 0wC simulations. Moreover, following the Clausius – Clapeyron relation, specifying that a warmer atmosphere can hold more water, the rising air temperatures lead to an increase in water vapour and clouds in the Arctic. Together with the decreasing ice sheet surface elevation and subsequent thicker atmospheric column, this leads to a negative feedback on or decrease in incoming shortwave downward (SWD) radiation. But, due to the increasing number of bare ice days per year and the reduced surface albedo, the absorbed SWD at the surface remains relatively stable (Fig. 12). Nevertheless, concurrently, the increase in cloudiness leads to further warming through increased longwave downward (LWD) radiation. By 2300 the increase relative to the 1990-2019 reference period in both solid and liquid clouds together is 73 % higher and the LWD increase is 5.8 % higher for the 2wC simulation than for the 1wC and 0wC simulations, compared over the same retreating ice mask (Fig. 14). As can be seen in Fig. 14, after 2300, the increase in longwave downward radiation continues. By 3000 it is 26 % higher in the 2wC simulation compared to the 1wC and 0wC simulations over the same retreating ice mask. Again, this is due to both the increase in cloudiness and the reduced surface elevation that results in a thicker atmospheric column above the surface. In addition, this enhanced longwave radiation can also be attributed to the Stefan-Boltzmann relation, stating that the total radiated energy by a body or matter is directly proportional to its temperature to the fourth power.

Secondly, as Figure 14 demonstrates, the increase in liquid clouds is much more pronounced than the increase in solid clouds and this increase is stronger for the 2wC simulation. By 2300, as a result of the climate forcing, the amount of liquid clouds over the retreating ice sheet increases by +283 to 313 % for the 1wC or 0wC and 2wC simulation, respectively, while the amount of solid clouds increases by only +55 to 58 % in all simulations. After 2300, regardless of the constant climate forcing, these trends continue in the 2wC simulation as a result of the thickening atmospheric column following the melt– elevation feedback. By 3000 the amount of liquid and solid clouds increase by +529 % and +85 %, respectively, compared to the start of the simulations. Though the increments after 2300 are thus smaller than those induced by the climate forcing until 2300, they are still substantial. Since the longwave emissivity of these two types of clouds differ, with the longwave emissivity of solid clouds being lower than that of liquid clouds, their changing ratio impacts the available energy for melt. As a result, the mean annual runoff over the retreating ice sheet increases. By 2300 it increases by a factor ~30 in all simulations, due to both climatic forcing and feedback effects. After 2300, regardless of the constant climatic forcing, it continues to increase and attains a factor 41.8 for the 2wC simulation (Fig. 14), which can thus be attributed to the melt– elevation feedback and related changes in cloudiness and cloud phase change.

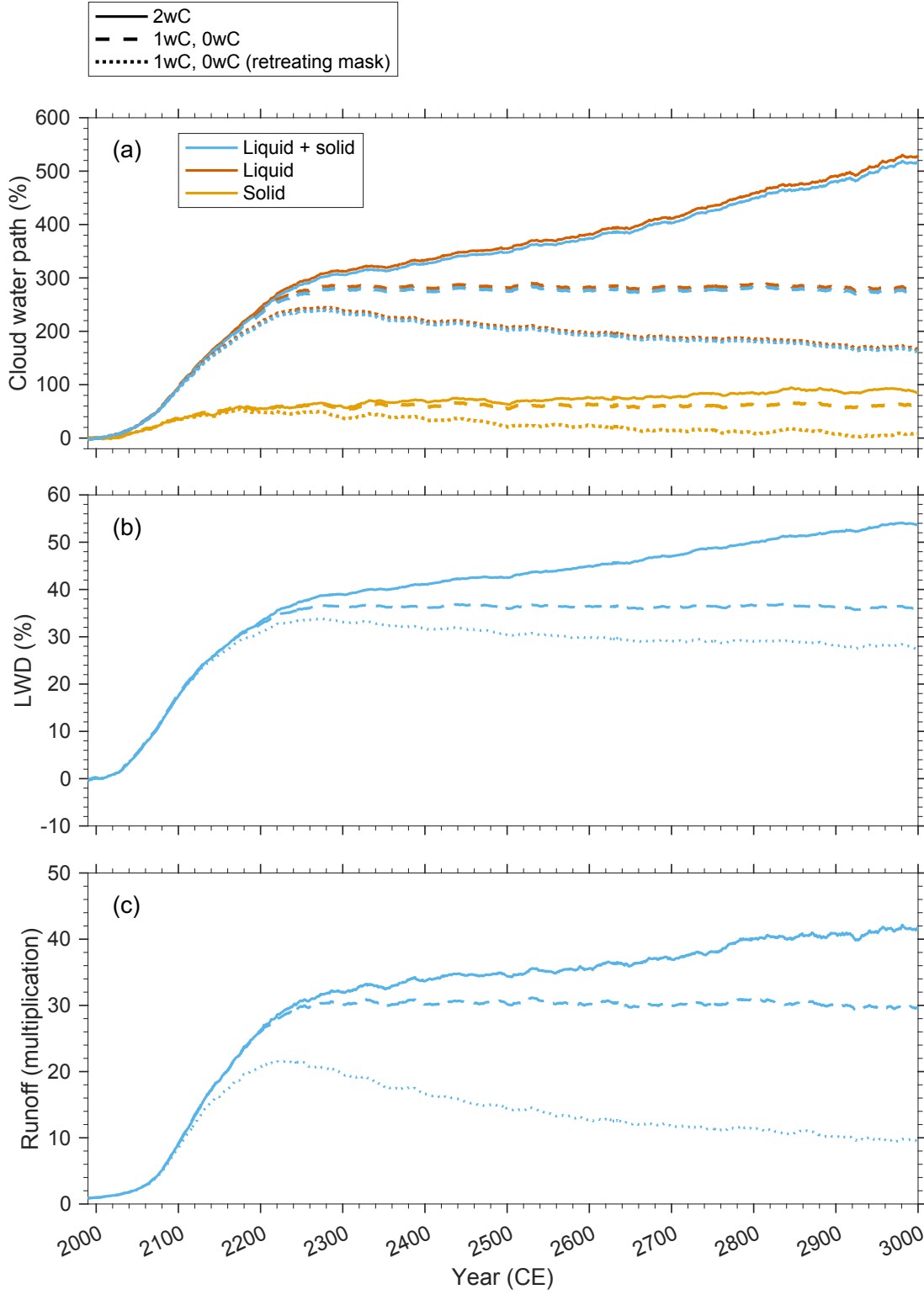

**Figure 14: Percentual increase in solid clouds (ice water path), liquid clouds (water vapour path) and the sum of both (a), percentual increase in longwave downward (LWD) radiation (b), and multiplication factor of runoff (c) with respect to the 30 year mean at the start of the simulations (1990–2019). Depicted is the 30 year running mean over the retreating ice mask for the 2wC simulation, and over the fixed as well as the same retreating ice mask for the 1wC and 0wC simulations.**

### 3.6.2 Changing precipitation phase and seasonality

Due to the applied high-warming climate scenario, a significant and similar increase in precipitation can be observed for all simulations up to 2300, as well as a phase change from predominantly solid to predominantly liquid precipitation (Fig. 15 and 16). More specifically, the mean rainfall over the ice sheet increases significantly from only 16 mm w.e. $y^{-1}$ at the start of the simulations to 360 mm w.e. $y^{-1}$ by 2300. Whereas in contrast, the annual mean snowfall peaks around 2100 in all simulations as it increases from 402 mm w.e. $y^{-1}$ to 475 mm w.e. $y^{-1}$ but decreases again towards its initial value by 2300 (Fig. 15). Hence, in terms of total precipitation over the ice sheet, the proportion of snowfall diminishes from 96 % at the start of the simulations to only 58 % by 2300. In addition, the proportion of summer snowfall diminishes from 26 % of the annual total snowfall at the start of the simulations to at most 4.5 % by 2200 and 2.4 % by 2300 in all simulations. This low summer snowfall proportion and the small difference between simulations (at most 0.7 %) further explains the reduction in accumulation area and the increase in bare ice exposure during most of the Arctic summer for all simulations (Sect. 3.4). In the 2wC simulation, this precipitation phase change and reduction in summer snowfall continue after 2300 due to the decreasing surface elevation.

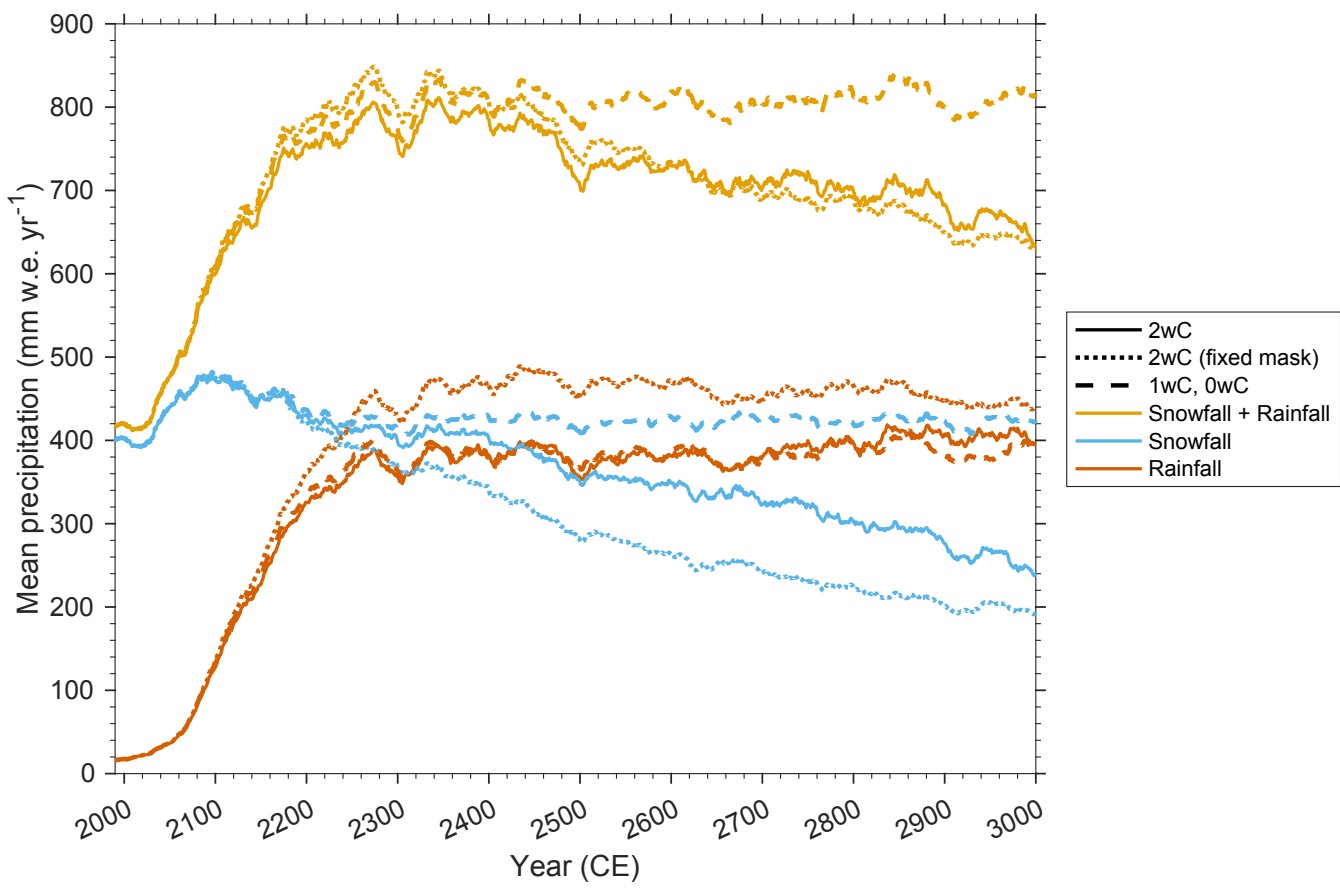

**Figure 15:** Mean precipitation (mm w.e. yr$^{-1}$) over time for all simulations on the 30 km MAR grid. Depicted is the 30 year running mean. For the 2wC simulation, the mean over the retreating ice mask is shown, as well as over the fixed mask at the start of the simulations (1990) that is identical to the one in the 1wC and 0wC simulations. Note that the large variability after 2300 results from the large variability in precipitation during the years that were randomly sampled (2250 to 2300) to prolong the IPSL-CM6A-LR forcing for MAR.

### 3.6.3 Inland displacement and diminishing orographic barrier

Besides, beyond 2300, in the 2wC simulation the precipitation also continues to move further inland following the retreating ice sheet margin, as often reported before (Toniazzo et al., 2004; Ridley et al., 2005; Vizcaíno et al., 2010; Solgaard and Langen, 2012; Gregory et al., 2020; Andernach et al., 2025; Feenstra et al., 2025). Though this may initially seem to positively impact the SMB, conversely, the precipitation increasingly falls as rain further inland, with a substantial increase in rain fraction over the retreating as well as the initial ice mask (Fig. 16). However, as Fig. 15 demonstrates, over time not all snowfall is persistently transformed into rainfall. Instead, the decrease in snowfall also leads to a reduction in total precipitation. The decline is strongest after 2500, as the mean snowfall drops from 398 mm w.e. y$^{-1}$ by 2300 to 349 mm w.e. y$^{-1}$ by 2500 and to 241 mm w.e. y$^{-1}$ by 3000 over the retreating ice sheet. This corresponds to a reduction of 42 % in total

snowfall and 38 % in total precipitation over the ice sheet between 2300 and 2500, with further decreases reaching 90 % and
86 %, respectively, by 3000. Since the decline in snowfall also occurs over the fixed (initial) ice mask identical to the mask
in the 1wC and 0wC simulations (Fig. 15 and 16), this snowfall decline cannot merely be attributed to the retreating ice sheet
area and the fact that it no longer extends as far as the principal south-eastern accumulation zone (Fig. 16). A comparison
with the precipitation over the fixed mask indicates that only about a third of the reduction in snowfall and a quarter of the
reduction in total precipitation can be attributed to the retreating mask. Hence, this indicates that the ice sheet no longer acts
as a strong orographic barrier for (solid) precipitation, as part of the air masses that used to precipitate (snow) onto the ice
sheet now pass over it without precipitating. This negatively impacts the accumulation and SMB over the ice sheet. In short,
the continued precipitation phase change and decline in solid and therefore total precipitation after 2300 in the 2wC
simulation are thus the result of the positive melt–elevation feedback, that intensifies around 2500 (Sect. 3.5).

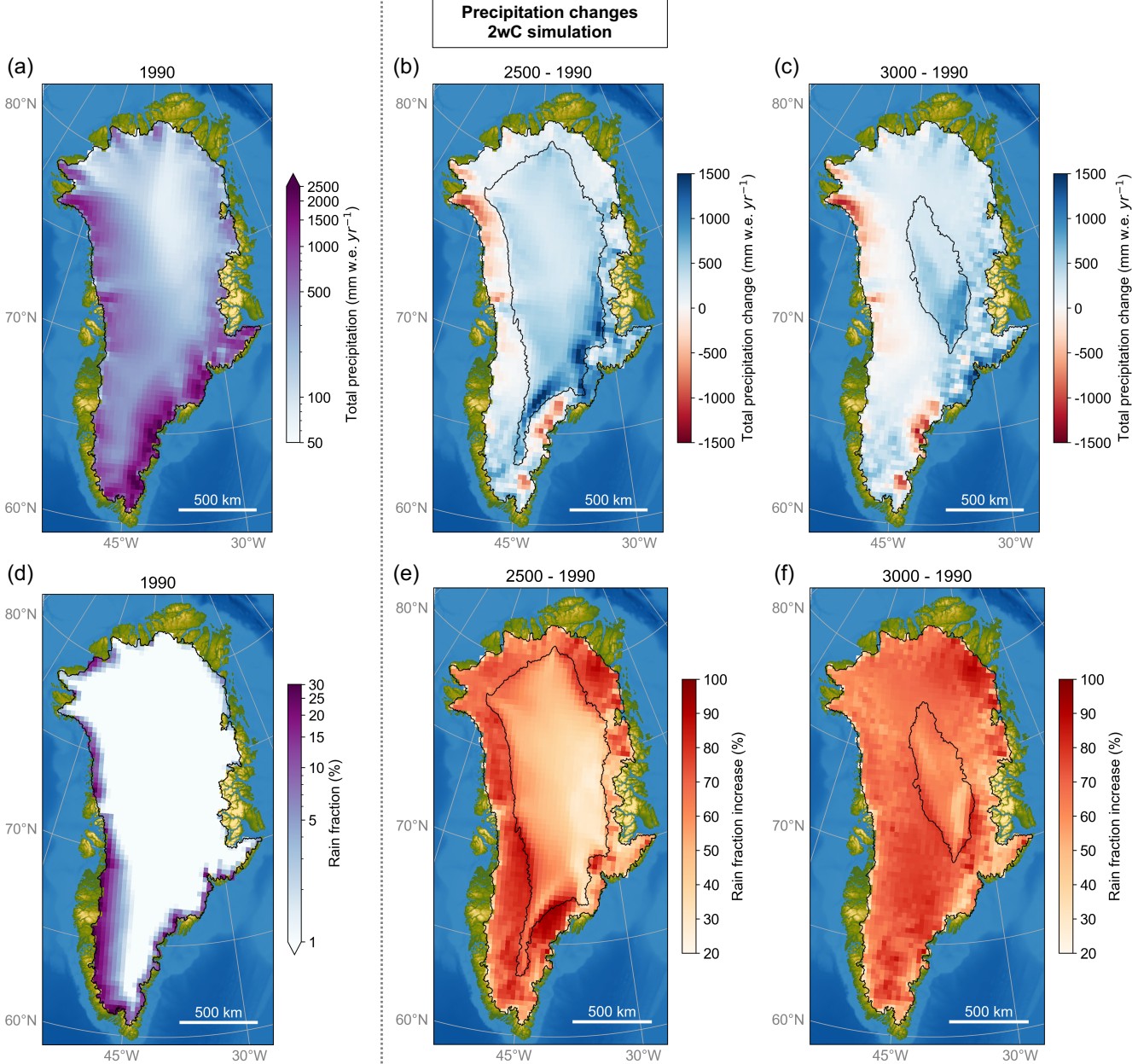

**Figure 16: Changes in total precipitation (sum of snow- and rainfall) (b-c) and rain fraction (e-f) for the 2wC simulation in 2500 and 3000, with respect to the initial conditions (a, d) at the start of the simulation. All values are shown on the 30 km MAR grid and depict the 30 year running means. The black contours indicate the initial as well as the remaining ice mask over time on the 5 km GISM grid.**

## 4. Discussion

### 4.1 Relative importance of negative and positive feedbacks

During the first three centuries of our simulations, the changes in the near surface wind speed have an observable negative feedback effect on ablation at the ice sheet margins, an effect that was also observed by Delhasse et al. (2024). This results in a similar ice sheet contribution to sea level across our three coupled simulations up to 2300. Nevertheless, though the effect persists, over time this negative wind feedback becomes subordinate to the positive feedback effects in the ice sheet–climate system.

The most important positive feedback effect is undoubtedly the melt–elevation feedback, that in turn also leads to a decline in snow fraction and total precipitation over the course of the 2wC simulation. Precipitation therefore acts as a positive feedback over longer timescales. This is opposed to former findings whereby the net impact of increased rain- and snowfall is almost negligible (Feenstra et al., 2025), or whereby precipitation is identified as a negative feedback because winter snowfall increases and slows melt (Ridley et al., 2005), snowfall decreases less than ablation (Gregory et al., 2020) or even increases (Hakuba et al., 2012). This highlights the importance of future research into the sign of the precipitation feedback and the need to carry out simulations using more (regional climate) models and climate forcing scenarios.

Around 2500, though its strength remains constrained by the low solar elevation angle during the Arctic winter months, the melt–elevation feedback intensifies. This coincides with an accelerated decrease in snowfall and an accelerated expansion of the melt season and runoff in both space and time. In addition, important changes are observed in the radiative budget, whereby a distinction can be made between the effects on the SWD and LWD radiation. Firstly, our results show that the negative feedback of increased clouds and water vapour (through the Clausius-Clapeyron relation and thickening atmospheric column) on the SWD radiation balances the positive melt–albedo feedback by reducing incoming shortwave radiation. In other words, the net absorbed shortwave radiation remains stable as the albedo decreases. This cooling effect from the reduction in SWD radiation is outweighed by the warming effect of the stronger increase in LWD radiation. The latter is the combined result of the increasing atmospheric temperatures (Stefan-Boltzmann relation) and the increase in water vapour and mainly liquid clouds with a higher longwave emissivity than solid clouds. Hence, though it is difficult to quantify all the effects separately, altogether the changes in clouds and the positive melt–elevation feedback amplify one another in our 2wC simulation. Though this seems to contrast with other long-term modelling studies that do not regard the impact of cloud changes on the longwave radiation (Gregory et al., 2020; Feenstra et al., 2025), it aligns with earlier findings regarding the impact of clouds on the Greenland near-surface climate and surface energy balance (Franco et al., 2013; Vizcaíno et al., 2014; Van Tricht et al., 2016; Hofer et al., 2019; Lenaerts et al., 2020).

In contrast, the factor that has a smaller impact on the coupled ice sheet–climate evolution than initially expected is the (summer) surface albedo and ensuing absorbed energy at the surface, that hardly differs between the simulations of different coupling complexity. This is likely due to the applied high-warming scenario, as already by 2200 the entire ice sheet has become ablation area in all simulations and reaches the minimal ice albedo, reducing the relative importance of the melt–albedo feedback (Zeitz et al., 2021).

## 4.2 Importance of coupling complexity

In our two-way coupled simulation, the GrIS has almost entirely disappeared within the next millennium. This is similar to earlier findings for a high-warming climate forcing scenario though with a different experimental set-up and without full coupling, using a general circulation model (Gregory et al., 2020) or corrected climatologies from an RCM (Aschwanden et al., 2019). Nevertheless, our results demonstrate that the contribution to sea level rise is severely underestimated over time when the ice sheet–atmosphere interactions are considered merely through the application of the offline extrapolation (1wC simulation) or entirely omitted (0wC simulation). This contrasts with previous long-term simulations from coarser resolution models, that identify a more significant role for negative feedbacks (e.g., Ridley et al., 2005; Gregory et al., 2020; Feenstra et al., 2025), leading to an overestimated sea level contribution from one-way compared to two-way coupled simulations. In our simulations, however, positive feedbacks dominate the ice sheet–climate system over time, amplifying ice mass loss, particularly beyond the centennial timescale and/or once the climate stabilizes. This aligns with results from the two centennial-scale ISM–RCM coupling studies to date (Le clec'h et al., 2019a; Delhasse et al., 2024). We find an underestimation of the sea level contribution of 10.4 % by 2150 or 14.4 % by 2200 when not including the melt–elevation feedback (i.e. 0wC simulation). This is somewhat higher than the 9.3 % by 2150 reported in Le clec'h et al. (2019a), and the 10.5 % by 2200 reported in Delhasse et al. (2024), since in these studies the simulations were extended with a stabilized climate beyond 2100. By 2300 we find an underestimation of 21.3 %, which is slightly lower than the 24 % reported by Vizcaíno et al. (2015) using a coarse resolution AOGCM. By 3000, the underestimation with respect to the 2wC is 20.8 % and 28.2 % for our 1wC and 0wC simulations, respectively.

In addition, our results illustrate that the Franco et al. (2012) method for extrapolating the SMB from the lower resolution RCM grid to the higher resolution ISM grid by means of annually and locally derived SMB–elevation gradients does not fully capture indirect effects on the SMB, like the effect of changing winds that act as a negative feedback in our 2wC simulation. This is consistent with observations by Delhasse et al. (2024) for their MAR-PISM simulations with different large-scale forcing (CESM2). On the one hand their horizontal model resolutions are similar to ours, namely 25 km for MAR and 4.5 km for PISM, such that it remains unclear to what extent these resolutions affect the strength of the observed negative wind feedback. Yet, on the other hand, the occurrence of the feedback and its poor reproduction by the offline extrapolation can thus be said to be independent of the coupled ISM and large-scale forcing. Therefore, as already suggested

by for example Le clec'h et al. (2019a), common downscaling procedures for SMB that rely heavily on the temperature–elevation gradient may not remain valid for large elevation differences, such as at the ice sheet margin or over longer timescales. Consequently, the added value of long-term one- and zero-way coupled simulations that rely on these procedures is questionable.

## 4.3 Remaining limitations

The applied 30 km resolution for MAR is still relatively coarse, as for example the 30 to 60 km broad ice-marginal zone wherein the near-surface wind speed changes are observed spans only two MAR grid cells. However, even at this resolution it is clear from the presented simulations that the location, type and amount of precipitation are very strongly topographically
controlled, highlighting the need for an RCM to accurately represent local atmospheric dynamics. In fact, the most accurate way to represent all ice sheet–atmosphere interactions would be to run both the ISM and RCM at the same horizontal resolution. However, as with all modelling research, the trade-off between the required computational resources, and the spatial as well as temporal resolution of the model (output) is inevitable. Running the RCM at the same resolution as the ISM thus currently remains unreasonable for millennial-scale simulations, like the ones presented here.

Regarding GISM, even though it was not run at the highest spatial resolution, several arguments justify the use of the 5 km grid. Foremost, the timescale and initialization procedure were among the most decisive factors in this respect. In addition, our results demonstrate that over time, the extrapolated SMB on the 5 km GISM grid increasingly deviates from the original SMB on the 30 km MAR grid, and it is reasonable to assume that this effect would become more pronounced as the
difference in model resolution increases. Therefore, considering the exploratory nature of this study, the timescale and the accurately represented observed ice sheet geometry with low remaining model drift, we do not regard the horizontal model resolution among the most prominent limitations of this study.

Applying a data assimilation procedure for the ice sheet initialization propagates model inaccuracies into the calibrated
parameters, making their effects difficult to trace (Berends et al., 2023). In addition, it is unlikely that the optimized two-dimensional fields for the BSC and EF in the ISM remain valid over a period exceeding several hundred years. Over such timescales their values will likely be impacted by the changing overlying ice thickness and basal hydrological conditions (Leclec'h et al., 2019a, 2019b). On the other hand, the fixed ice temperature is not expected to substantially affect the presented results, as the rate of ice melt in our simulations exceeds the rate at which ice temperature is altered through
advection or the propagation of atmospheric temperature perturbations into the ice.

Regarding the coupled model set-up, incorporating the glacial isostatic adjustment could mitigate the (rate of) ice mass loss and the strength of the melt–elevation feedback. As this would complicate the initialization procedure for equilibrating the

coupled MAR–GISM model and since it does not represent an ice sheet–atmosphere interaction in itself, this process was omitted here. Besides, the impact of this negative feedback is likely limited for the presented simulations, as the observed rapid ice sheet collapse is expected to outpace the slow glacio-isostatic rebound. This was also suggested by Aschwanden et al. (2019) who reported a mass loss reduction of only 2 % within the next millennium due to this feedback in their study.

Although ice–ocean interactions are not the focus here, it should be noted that ice discharge at outlet glacier fronts is expected to accelerate due to warming ocean temperatures and increased surface runoff (Slater et al., 2019), at least as long as the ice sheet remains in contact with the ocean, which is up to the 2340s in our 2wC simulation. Since dynamic outlet glacier retreat was not included here, the (rate of) ice mass loss in our coupled simulations might thus be somewhat too low during the first centuries. On the other hand, part of the ice at the ocean boundary might be removed by SMB-driven surface melting before it reaches the calving front, such that the present-day observed discharge rates cannot merely be extrapolated over time (Fürst et al., 2015) and the impact is thus likely more limited than such extrapolations would suggest.

Lastly, our coupled model set-up does not represent the impact of the GrIS decline on the large-scale atmospheric and ocean circulation over the northern hemisphere (e.g. Davini et al., 2015; Madsen et al., 2022; Andernach et al., 2025; Haubner et al., 2025) but this is far beyond the scope of the study. Nevertheless, as MAR simulates its own boundary layer over the changing topography and land cover independently of the large-scale forcing fields, we are confident that the modelled ice sheet–atmosphere interactions are successfully represented.

## 5. Conclusions

To obtain a better understanding of the ice sheet–atmosphere interactions and potential feedback mechanisms over Greenland, we have coupled an ISM and RCM and performed the first long-term simulations under a high-warming scenario extending over the millennial timescale. These millennial-length MAR–GISM simulations consist of a zero-way, one-way, and two-way coupled simulation that were forced with six-hourly outputs from the IPSL-CM6A-LR model under the extended SSP5-8.5 scenario until 2300. They were prolonged until 3000 by randomly sampling the last 51 years of forcing. Thanks to the rigorous iterative initialization procedure, the remaining coupled model drift is minimal and the differences between the zero-way, one-way, and fully two-way coupled simulations represent the growing significance of the ice sheet–atmosphere interactions over time.

Compared to the 2wC simulation, we find that the ice sheet contribution by 2300 is underestimated by 46.9 cm s.l.e. or 21.3 % when omitting all feedbacks between the ice sheet and atmosphere, and by only 5.2 cm s.l.e. or 2.4 % when accounting for the melt–elevation feedback by means of an offline correction. The latter small difference can in part be attributed to the

similar bare ice cover during the melt season, and the ensuing lack of difference in the positive melt–albedo feedback between the simulations. In addition, distinct spatial differences in SMB were observed between the 1wC and 2wC simulations, that largely compensate one another and thus lead to a similar integrated ice mass loss. These SMB differences can be attributed to changing near-surface wind speeds that reduce ablation along the ice sheet margin in the 2wC
simulation, and the fact that this negative feedback effect is not adequately represented by the offline extrapolation. This shortcoming becomes more pronounced for the 1wC and 0wC simulations due to the growing topographic differences over time between the (fixed) MAR and (retreating) GISM ice sheet topography in these simulations.

Beyond the 2300 timescale however, and under constant climate forcing, the contribution to sea level rapidly differs between
simulations, especially between the 2wC and 1wC simulation, indicating that positive feedback effects dominate the ice sheet–climate system and ice mass loss in the 2wC simulation. As a result, by the year 3000, the ice sheet has (almost) entirely disappeared with a sea level contribution of 7.135 m s.l.e. for the 2wC simulation, compared to only 5.635 m s.l.e. and 5.122 m s.l.e. for the 1wC and 0wC simulations, respectively. For long-term simulations, both the omission and implicit representation of the melt–elevation feedback thus lead to a severe underestimation of the ice sheet contribution to sea level.
Around 2500, the positive melt–elevation feedback intensifies. We observe a decrease in both solid and total (orographic) precipitation and the summer runoff expands more rapidly in both space and time. In addition, the rising atmospheric temperatures, and increasing water vapour coincide with an increase in mainly liquid clouds that further increase the runoff through amplified LWD radiation. Though at the same time the increased clouds and water vapour act as a negative feedback effect on the SWD radiation, this effect is balanced by the positive melt–albedo feedback. Hence, altogether the positive
feedback effects on the radiative budget prevail.

However, better constraining the importance of each feedback separately would require more simulations wherein each feedback is switched on and off, as well as similar investigations under a range of future warming scenarios. We would like to emphasize that the presented millennial-length coupled ISM–RCM simulations would not have been possible without the
6-hourly large-scale forcing up to 2300 from the IPSL-CM6A-LR model under the SSP5-8.5 scenario. The availability of extended global climate model output is therefore crucial for future ice sheet–climate research.

**Code and data availability**

The main output data from this study, as well as information regarding the MAR and GISM source code used to generate the data will be made publicly available upon publication.

## Author contributions

CMP, XF and PH conceived the study. CMP conducted the simulations and prepared the manuscript with contributions from all authors.

## Competing interests

Some authors are members of the editorial board of The Cryosphere.

## Acknowledgements

The resources and services used in this work were provided by the VSC (Flemish Supercomputer Center), funded by the Research Foundation - Flanders (FWO) and the Flemish Government.

We would like to thank Christoph Kittel for an insightful discussion and ensuing improvement of the manuscript.

## Financial support

Chloë Marie Paice holds a PhD fellowship with number 1147824N of the Research Foundation-Flanders (FWO-Vlaanderen).

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
