# Peer review of "Positive feedbacks drive the Greenland ice sheet evolution in millennial-length MAR–GISM simulations under a high-end warming scenario"

_EGUsphere, 2025_

## Referee Comment (RC1)

**Review of the manuscript "Positive feedbacks drive the Greenland ice sheet evolution in millennial-length MAR–GISM simulations under a high-end warming scenario ", by Chloë Marie Paice et al.**

Paice et al. present the first coupled simulations of an ice sheet model and a regional climate model on a millennial timescale. To this end, they coupled the ice sheet model GISM to the regional climate model MAR, forcing it with IPSL-CM6A-LR global climate model output under the SSP5-8.5 scenario. They compare a zero-way (elevation changes are not taken into account), one-way (offline correction to consider elevation changes), and two-way (elevation changes are communicated between GISM and MAR) simulation, spanning the period 1990-3000. The analysis focuses on feedbacks arising from Greenland ice sheet elevation changes. The authors discuss several positive and negative feedbacks that play a role over different time scales and relate this to the surface mass balance and ice mass loss.

This study builds on the work done by Le clec'h et al. (2019) and Delhasse et al. (2024), in which MAR was coupled to the ice sheet models GRISLI and PISM. This study brings something new by presenting simulations spanning over a millennium. Over longer timescales, the effects of coupling complexity on ice mass loss become more pronounced, and feedbacks that play a role on longer timescales can be identified. This study presents a sound methodology and brings new and interesting results that will be valuable to the ice sheet modelling community. The paper is well written, and the results are presented in clear figures. My comments mainly concern the attribution of the differences between the simulations to the different feedbacks and the interpretation of climate interactions. After addressing these, I would recommend this manuscript for publication in The Cryosphere.

**Major comments**

**The choice of the ice sheet model:** To me, it is unclear why you couple MAR to another ice sheet model, rather than using the existing MAR-GRISLI or MAR-PISM. Is this for computational reasons, or to explore the sensitivity to the choice of ISM? Can you add something about the choice for this model at the beginning of section 2 or in section 2.1?

**Section 3.3 on wind speed changes:** In this section, the reason for a different pattern in SMB changes between the simulations is attributed to changes in katabatic and barrier winds. Although I find these results very interesting, I do not think you can attribute all these differences to differences in wind patterns. I would also suggest looking into the effect of your offline correction on the differences in the near-surface air temperature changes at the margin. In your offline correction, you use an SMB gradient based on a climate computed at a higher elevation (since the MAR topography does not change). Using these SMB-elevation gradients, you implicitly assume some sort of relationship between temperature and elevation. When a grid cell becomes part of the ablation area

in GISM, but not in MAR, the relationship you find based on the MAR SMB will not hold for GISM, because the energy required for melting reduces temperature increases. This (combined with the decreased winds) might lead to a too strong SMB-elevation gradient around the margins. Hence, your corrected SMB might correspond to a higher temperature than you would find in two-way coupled. You can also see this behaviour in Fig. 7a, where the strong increase in temperature at the high elevations is due to the initial global warming until 2300, the slower, but rather linear increase thereafter results from the temperature-elevation feedback, and the constant temperatures around zero for the lower elevations are because of the thermodynamic limit of melting ice. The timing of changing towards a new type of temperature-elevation relationship would then likely not be the same between MAR and GISM, leading to differences in SMB. I understand that you do not downscale temperature separately for your offline correction, but I would suggest thinking about this at least in a conceptual way. Regarding your statement on the increase of cloudiness derived from the increased long- and decreased shortwave radiation, I agree that the radiation changes are likely because of an increase in cloudiness, but you should consider a few other processes as well. The increase in water vapour in the column due to both the increase in temperature (Clausius-Clapeyron) and the elevation decrease leading to a larger atmospheric column will also lead to a decrease in shortwave and an increase in longwave radiation. Additionally, for the longwave radiation, increasing air temperatures affect the Stefan-Boltzmann relationship as well. Then, you attribute this increase in cloudiness to decreasing wind speeds, resulting in "cloud accumulation". Although this is a likely explanation for at least a part of this increased cloudiness, you should not forget about the increase in cloudiness due to a warming atmosphere (Clausius-Clapeyron), which you do mention in section 3.6.1, but should be mentioned here as well.
For this section, it is also not completely clear to me whether you are only considering the period up to 2300 or the whole period. Initially, I was thinking about the whole period, since the figures 6 and 7 show this as well, but then lines 353-354 contradict this. So please state more clearly which period you are considering and if this is only a shorter period (e.g., till 2300), where to look in the figures.

**Section 3.6.1 on clouds:** This section discusses the changes in cloudiness and cloud phase and their effect on the shortwave and longwave radiative effect. I am happy to see that you separate solid and liquid clouds and consider their distinct effects on radiation. I do think you are missing a few things regarding the cloud effect on radiation and the claim that clouds act here as a positive feedback. That there is no net effect on shortwave radiation does not mean there is no feedback effect of clouds on shortwave radiation. In fact, I would say that the negative feedback of clouds on shortwave radiation counters the positive melt-albedo feedback by reducing incoming shortwave radiation. Additionally, the observed changes in the radiative fluxes are the sum of changes in more components than just clouds. To make a definite statement towards the sign of the cloud radiative effect and feedbacks, you would need to disentangle the effects of clouds, water vapor, temperature and albedo, which would be very complex using the set of simulations you did and involve a lot of assumptions. Therefore, I suggest toning down and mentioning that there are more components influencing the radiation budget. Also consider this in lines 542-548 in the discussion. Specifically, "long

wave warming outweighing cooling associated to the reduced incoming short wave radiation" would be a too strong statement since you cannot separately compute and compare the longwave and shortwave radiative effects from your results.

**Minor/technical comments**

p1, line 9: "predicting" -> "projecting"

p2, line 56: remove "and winds", since you only discuss this in the next paragraph.

p3, line 84-85: "It is the first time ice sheet–atmosphere interactions are accurately accounted for on the millennial timescale." I would tone down here. As you mention, there are a number of studies that presented two-way coupled simulations on millennial timescales before, but with lower resolution or complexity. Therefore, these MAR-GISM simulations with higher resolution and/or complexity than used in these other studies will add a lot to our understanding of feedbacks, but 30 km is still a rather coarse resolution (as you mention in the discussion), and both MAR and GISM have their limitations as well. I would suggest highlighting that your simulations are run on the highest resolution or that your model has the highest complexity, rather than stating that they "are accurately accounted for".

p4, line 109: "using the higher-order approximation", what kind of approximation is this? Could you mention it briefly?

p4, line 116-117: What do you mean by "acceptable level of discrepancy between the model resolutions"? Do you mean that the resolutions should not be too different, and it would not be ideal to run GISM on a much higher resolution?

p6, line 170-174: Can you briefly explain what these optimization and relaxation steps are?

p10, line 266: several hundred metres around the margins, where the ice thickness is small, is quite a large relative difference. Do you know what causes these differences (particularly in the central east)? How does it affect your results?

p12, line 275-276: I am a bit confused about the definition of the GBI region. It makes sense to show the temperature evolution over this region, but since you are not considering the GBI in your analysis, I think it only adds confusion. You could just refer to it as "Greenland region" or something similar, with the definition of the coordinates.

p13, fig 3: same for GBI region, and add that the MAR 2m mean annual air temperature is for the two-way simulation.

P15, line 310-313: I do not see a clear overestimation of SMB in one-way over the interior.

p15, line 313-315: I had to read this multiple times to understand what you are referring to. Can you rephrase in such a way that it is clear that you mean the change of sign of the difference between the GISM SMB between one-way and two-way, which does not occur for the MAR SMB?

p16, line 329: decreased barrier winds, compared to what? I assume this is a decrease over time and only observed in two-way?

p16-17, line 326-335: How do you define barrier and katabatic winds here? You describe the maximum changes in wind speed; are these the locations with the largest change? I think it could be very interesting to add some plots showing the (changes in) katabatic and barrier wind speeds over the ice sheet. Or otherwise, at least describe where these changes are most pronounced, or whether the behaviour is very similar along the whole Greenland margin.

p17, line 339-340: I do not see this sudden decrease in temperature in Fig. 7a.

p18, Fig. 6c: It is not clear to me what all these lines are. If you plot one line for the MAR SMB and one for the GISM SMB for 9 locations, you would end up with fewer lines than you have now. Also, the reduced colour saturation makes the lines with small markers quite hard to see on the left side.

p19, Fig.7: longwave and shortwave instead of long wave and short wave. There are also multiple instances of this in the text (e.g., line 340 and section 3.6.1).

p20, line 373: Please state whether you consider the MAR or GISM SMB and refer to Fig. 11.

p20, line 376-379: Is the bare ice albedo in MAR a fixed value?

p23, Fig. 10: I would suggest showing the summer fluxes instead of the annual mean, since the melt mainly happens in summer.

p24, line 411: mean surface elevation -> mean surface elevation decrease

p24, line 413: "The time of this intensification", I assume this is 2500?

p25, line 425: What is meant by "entire Arctic summer"? From the context, I would understand this is >150 days. Please define Arctic summer or mention that you refer to the yellow line in the plot.

p25, line 428-429: Do you know why the areal expansion for runoff days is so different from the number of bare ice days?

p25, line 431-436: I do not think this is very new. With no incoming shortwave radiation for a large part of the ice sheet during winter, it is no surprise that there is no energy available for melt. Therefore, I would focus more on the extension of the melt season, which you did in the previous paragraph, and consider shortening or removing this paragraph.

p22, Fig. 9 and p26, Fig. 12: Please make the colours consistent between the plots. Now, in Fig. 9, the shortest number of days is in red, and the longest in blue, while for Fig. 12, it is the other way around.

p27, line 459-460: A thicker atmospheric column also results in changes in shortwave radiation. Additionally, you should consider the effect of the Clausius-Clapeyron relationship here as well. It does not only result in an increase in cloudiness, but in water vapour as well, influencing radiation. For longwave, temperature changes can have an influence as well through the Stefan-Boltzmann relationship.

p28, Fig. 13: Do you show the increase in cloud cover or cloud water path here? Please mention this. In case you used cloud cover, I would suggest looking into the water path, since this likely tells you more about the amount of clouds and their radiative effect than just cloud cover. An increase in cloud water path might not be reflected in the cloud cover, but it still has large effects on radiation.

p30, Fig. 14: Why do you show the grey area in this figure specifically, but not in many of the others? I would either show this in all timeseries or only in Fig. 3.

p30, line 507: "This corresponds to a reduction of -42 % in total snowfall and -38.4 %" I think you can get rid of the minus signs, since a reduction already implies it is negative.

p33, line 531-537: Are there other studies that looked into the inland displacement of precipitation and the diminishing orographic barrier? I particularly find your results on the diminishing orographic barrier very interesting, and I would be interested in knowing whether this was found before.

p33, line 539-540: I do not think that the fact that the melt-elevation feedback is constrained by the solar elevation angle is a new finding of your study. The same comment for the conclusion, p36, line 650-652. There, you also state that this is even the case under high-warming scenarios, but the solar elevation angle is independent of the warming scenario.

p33, line 544: associated to -> associated with

p34, line 567-570: "not including the melt-elevation feedback" Are you referring to the zero-way or one-way simulation? Also, you find a larger underestimation than the three other studies. Is this because you have stronger warming by 2200 in your simulations? Or could this be attributed to the choice of ISM? If the warming is very different, you could consider comparing SLR contributions linked to a certain amount of warming instead of timing?

p34, line 574-575: I suggest rephrasing this. Since you do not downscale winds, it makes sense that your offline correction does not capture this feedback. I would rather say something about the fact that you need two-way coupling to include this effect.

p34, line 585-586: How well can you resolve wind speeds at 30 km resolution? Is this resolution high enough to capture the katabatic and barrier winds realistically?

p35, line 593-598: This paragraph is a bit vague, especially the last sentence. I would consider removing this paragraph entirely and maybe moving the first sentence to the paragraph before.

p35, line 601-603: I would not consider the increasing difference between the MAR and GISM SMB when using a higher GISM resolution a big problem. I think the computational constraint justifies the choice for the 5 km resolution already well enough.

p35, line 609: remove "though"

p35-36, line 620-626: I would consider moving this to line 606, since this paragraph discusses limitations of your ISM.

p36, line 623: remove the comma here.

p36, line 636-642: Are winds the only reason for this difference?

p40, line 741-743: Feenstra, et al. (2024) is not in discussion anymore: https://doi.org/10.5194/tc-19-2289-2025

**References**

Le clec'h, S., Charbit, S., Quiquet, A., Fettweis, X., Dumas, C., Kageyama, M., Wyard, C., and Ritz, C.: Assessment of the Greenland ice sheet–atmosphere feedbacks for the next century with a regional atmospheric model coupled to an ice sheet model, The Cryosphere, 13, 373–395, https://doi.org/10.5194/tc-13-373-2019, 2019.

Delhasse, A., Beckmann, J., Kittel, C., and Fettweis, X.: Coupling MAR (Modèle Atmosphérique Régional) with PISM (Parallel Ice Sheet Model) mitigates the positive melt–elevation feedback, The Cryosphere, 18, 633–651, https://doi.org/10.5194/tc-18-633-2024, 2024.

---

## Referee Comment (RC2)

Review of egusphere-2025-2465: "Positive feedbacks drive the Greenland ice sheet evolution in millennial-length MAR–GISM simulations under a high-end warming scenario" (Chloë Marie Paice et al.)

**Summary:**

Paice et al. present a suite of coupled ice-sheet–atmosphere simulations of the Greenland Ice Sheet (GrIS), spanning the period from the present to the year 3000. The regional climate model MAR is coupled to the ice-sheet model GISM. The simulations are initialized through a glacial–interglacial spin-up of the ice sheet model (ISM), followed by a data assimilation procedure to obtain a realistic present-day ice-sheet geometry. Three forward simulations with varying coupling degrees are then performed under a high-emission scenario: (i) a zero-way coupled run, in which MAR-derived surface mass balance (SMB) is applied directly to the ISM without accounting for evolving surface height; (ii) a one-way coupled run, in which the SMB–elevation feedback is parameterized; and (iii) a two-way coupled run, in which annual changes in ice-sheet geometry are passed from GISM to MAR. Results from these experiments are used to systematically assess key atmospheric feedback processes affecting ice-sheet evolution, including the SMB–elevation feedback, wind, cloud, and albedo effects. The authors find that positive feedbacks (e.g. the SMB-elevation feedback) dominate on the millennial timescale and substantially enhance future ice mass loss.

This study represents the first millennial-scale two-way coupling of an ISM with a regional climate model (RCM), making the manuscript both novel and highly relevant for advancing our understanding of Greenland's long-term atmospheric feedback processes.

**General Assessment:**

This manuscript is highly relevant to the field of ice-sheet modelling and makes an original contribution by advancing our understanding of long-term atmosphere–ice-sheet interactions over Greenland. The paper addresses important scientific questions that fall well within the scope of *The Cryosphere*. The authors present novel methods and results, in particular the two-way coupling of an ice-sheet model with a regional climate model over millennial timescales, which, to my knowledge, has not previously been demonstrated.

The title accurately reflects the content of the paper. The scientific methods and assumptions are valid and are generally well described in the methods section, though I suggest some refinements and clarifications to improve clarity, as well as the addition of

visual aids. The conclusions are substantial and well supported by the presented results. I particularly appreciated the systematic approach to disentangling and presenting the various feedback processes that contribute to future mass change of the ice sheet. The manuscript is well structured and generally clearly written, though in some instances sentences are awkwardly formulated and the logical connections within paragraphs are not always transparent.

I recommend publication after minor revisions, mainly to improve language, grammar, and precision, and to enhance clarity in the methods section.

**General Comments:**

1. Although the methods are generally well described, some details remain unclear. I recommend providing additional precision and detail where necessary (see specific comments). In addition, a visual aid would be valuable to guide the reader through the initialization steps and the interactions between MAR and the ISM (see also comment on Sect. 2.3).

2. While the manuscript focuses on atmosphere–ice interactions, I suggest explicitly addressing the treatment (or non-treatment) of the ice–ocean boundary in both the methods and discussion sections. In particular: How are outlet glaciers and their potential retreat represented in the simulations? If dynamic retreat is not included, how do the authors justify this omission, and how might they expect dynamic retreat to interact with SMB-driven retreat? What limitations does this introduce to the interpretation of results? I believe this point is important, since dynamic retreat of outlet glaciers contributes substantially to present-day mass loss of the GrIS and is expected to remain significant in the future, at least as long as the ice sheet is in contact with the ocean.

3. The manuscript is generally well written and well structured, but some sentences are awkwardly phrased, and the logical connections are not always clear. I believe a careful revision of the language, sentence structure, and causal links within sentences and paragraphs would improve clarity and linguistic precision.

**Specific Comments:**

**l. 22:** To avoid misinterpretation, I suggest to already clearly state in the abstract that glacial isostatic adjustment is not considered, when mentioning numbers of sea-level contribution.

**l. 33:** The expression "… *the major remaining uncertainties…"* is a bit unclear. Please specify what "*remaining*" is referring to.

**l. 43:** Explain why the air temperature increases (changes in air density, adiabatic lapse rate).

**l. 45:** I suggest adding a sentence or two explaining why it is not straightforward to include the SMB-elevation effect in ISMs (see also comment l. 72).

**l. 52:** I suggest replacing "*landward*" with "*inland*" and "*contributes*" with "*increases*".

**l. 58:** I suggest to explicitly explain how clouds affect the meltwater refreezing capacity (via surface energy balance/longwave radiation).

**l. 72:** You could mention why it is not straightforward to represent ice-atmosphere interaction in standalone ISMs and why coupling is needed (see also comment l. 45).

**Sect. 2.3:** If I understand correctly, during the initialization procedure you apply 1961-1990 SMB, but you also assume the ice sheet to resemble present-day conditions at the start of your forward simulations. How do you treat the period between 1991 and present-day? Is this period explicitly simulated? If so, which forcing is used? If not, how do you handle this gap and the observed mass-loss trend? Please clarify.

**l. 96:** Mention that initialization should ideally reflect not only the present-day state but also current mass loss trends in response to recent 'historical' forcing (see also comment l. 118). See for example Rahlves et al. (2025) for including historical trends of ice mass loss in projections.

**l. 99:** To me it is not clear what you mean by "*additional assumptions*". Please clarify.

**l. 110:** Consider explicitly showing the sliding law equation.

**l. 111:** Please specify what is meant by "*geometric input*" (bedrock topography and/or ice sheet geometry).

**l. 116:** Define "*acceptable*" here. How is the acceptable discrepancy level determined, and what happens if it is exceeded?

**l. 118:** You describe the system as initialized *"into an equilibrium state resembling present-day observations."* In reality, the ice sheet is not in equilibrium at present. How do you account for this (see also comment l. 96)?

**Sect. 2.3:** Since the initialization is a complex but central part of your setup, I recommend a schematic figure/flowchart showing the main steps (spin-up, data assimilation, coupled initialization), inputs/forcings, parameters adjusted vs. fixed, and the MAR–GISM information exchange (including frequency).

**l. 163:** Although you mention here that the isostatic bedrock adjustment is disabled after the spin up, I suggest reiterating this in Sect. 2.5 when describing the future simulations.

**l. 165:** Add a brief explanation of how the positive degree-day approach works.

**Sect. 2.3.2**: Reiterate here that GISM is forced with MAR SMB during data assimilation (currently mentioned earlier).

**l. 184**: Regarding peripheral glaciers: if data assimilation is not applied there, does this mean they are initially too large? How are they treated in sea-level contribution calculations? Do they make a difference? Please clarify.

**l. 190:** The phrase *"unvarying parameters"* is misleading. Suggest rephrasing as: *"Holding these parameters fixed is justified over short-term periods …"* Also specify what "short-term" vs. "long-term" means, and where your simulation timeframe fits.

**l. 195:** Be more precise: *"… computed on the ice-sheet topography as simulated by GISM."*

**Sect. 2.5:** What is the first year of the forward simulation (after the initialization procedure is complete)? I suppose it is 1991, but I think it would be good to mention it explicitly.

**l. 216**: Does *"similar to …"* refer only to the zero-way run, or to all three? If the latter, consider: *"Similar to …, we consider three coupling types …"*

**l. 234:** Fig. 2 shows differences in ice thickness, not the initialized topography itself. Please rephrase.

**l. 241 ff.:** I find this sentence difficult to understand. I suggest rewriting for clarity.

**Sect. 2:** In your methods section I am missing a description of how you are treating the ice-ocean boundary. Please include a brief explanation.

**l. 252:** *"…since only SMB was extrapolated…"* What about runoff?

**l. 256:** Clarify how sea-level contribution is calculated and what the reference period is.

**l. 264:** Specify *"fourth iteration"* of what.

**l. 305:** Instead of *"the first part of the simulation,"* write *"During the first X years ..."*

**l. 310:** I find the expression "compensation of differences" unclear. Differences of what? Suggest rephrasing to *"spatial compensation within SMB fields."* Also, I think the expressions "over- and underestimation" only work if there is a reference. Consider rewriting to "overestimated compared to..." and "underestimated compared to...".

**l. 335:** Refer to a figure if this is illustrated.

**l. 336:** Clarify causality: e.g. *"Changes in wind speed impact the surface energy balance, reducing runoff and lowering SMB (Fig. 6)."*

**l. 375:** The causal link in this sentence is unclear. Please clarify how the density of the upper ice layers is used as an indicator of snowfall melt. In addition, it would be helpful to specify the actual density values of these layers to support the statement.

**l. 403:** Remind the reader why low-lying margins retreat less rapidly in the 1wC simulation.

**l. 411:** Insert "loss" after mean surface elevation and change "stronger" to "strong".

**l. 413:** Specify the year of *"At the time of this intensification ..."*

**Sect. 3.6:** Consider merging 3.6.1 and 3.6.2 instead of using sub-subsections.

**l. 455:** Clarify that *"increase"* is relative to the 1990–2019 reference period.

**l. 466:** Explain why this is specifically a result of the melt–elevation feedback.

**l. 467:** I could not fully follow how this is shown in the text. Please clarify.

**l. 481:** Omit "important".

**l. 489:** Rephrase: *"... and the increase in bare ice exposure."*

**l. 491:** Add: *"... and the associated increase in near-surface air temperature."*

**l. 503:** Replace "snowfall" with "precipitation".

**l. 504:** The two clauses are not logically connected. Consider rewriting to: *"However, as Fig. 14 demonstrates, not all snowfall is transformed into rainfall. Instead, decreasing snowfall also reduces total precipitation."*

**l. 526:** Rephrase: *"Similar to Delhasse et al. (2024), during the first three centuries ..."*

**l. 528:** Rephrase: *"This results in a similar ice-sheet contribution to sea-level rise across our three coupled simulations up to 2300."*

**l. 542:** Add "per year" to "120 runoff days".

**l. 557:** When comparing to Aschwanden (2019) and Gregory (2020), stress that setups differ (coupled vs. uncoupled, forcing strategies). This distinction should be made clear throughout the discussion.

**l. 569:** The phrasing *"... in which the climate did not continue to warm up to 2200 ..."* is not entirely clear. It would help to clarify whether the intended meaning is that previous studies assumed climate stabilization before 2200, or simply that warming was not extended in their scenarios. Consider rephrasing for precision.

**l. 576:** The sentence *"Though their applied horizontal model resolutions ..."* is difficult to follow. I suggest rephrasing or splitting into shorter sentences to make the causality clearer.

**Sect. 4.3:** Briefly discuss missing representation of ice–ocean interactions as a limitation. Outlet-glacier dynamics currently contribute significantly to GrIS mass loss and will likely remain important over several centuries. Also note feedbacks between SMB and dynamic processes (e.g. thinning reduces flux vs. ocean warming accelerates retreat). Even if not in the scope of your study, this deserves a short mention.

**l. 592:** The argumentative line of this paragraph is not clear to me. Can you rewrite to make your thought clearer?

**l. 615:** Consider briefly mentioning expected effects of GrIS on large-scale circulation, and cite relevant studies (e.g. Haubner et al., 2025).

**l. 644:** Since you start a new paragraph here, I suggest opening with *"Beyond the 2300 timescale ..."*.

**Figures:**

**Fig. 2:** Omit the title (information is already in the caption) and add a description to the color bar (e.g. *"difference in ice thickness (m)"*). If possible, adjust the color scheme for detached peripheral glaciers to avoid overlap with the palette, where white indicates zero difference.

**Fig. 4:** Consider using a color bar for ice thickness (grey tones) as well as for the bedrock topography.

**Fig. 5:** Omit the title (repeated in the caption) and label all subpanels (a–e). In the upper panels, outline the selected regions in the same colors used for the corresponding SMB values in the lower panels, to improve readability. Add descriptions to the color bars (e.g. *"difference in SMB"*). In the caption, clarify whether the SMB evolution refers to areamean SMB or integrated SMB, and specify whether *"mean SMB"* on the y-axis denotes areal mean or annual mean.

**Fig. 6 & 7:** Consider reducing the number of points along the transects to make the plots less cluttered. As a matter of style, I suggest placing subpanel labels outside the plots.

**Fig. 9, 10, 11, 12:** Consider merging these into one figure, since they share the same timescale.

**Fig. 13:** Remove the title; this information is already given in the caption.

**Fig. 15:** Add descriptions to the color bars (e.g. *"total precipitation," "precipitation change,"* and *"rain fraction (increase)"*).

**Technical Corrections:**

**l. 56:** Include commas: *"It has, for example, been reported..."*

**l. 130:** Change to: *"The main difference to ..."*

**l. 197:** Avoid *"... until the differences ... no longer change ..."*. Instead, I suggest rephrasing as: *"until the differences become small/insignificant/approach zero"*

**l. 230:** (and other occasions): Replace *"on Fig. 2"* with *"in Fig. 2"*

**l. 427:** Replace *"a clear link with"* with *"a clear link to"*.

**l. 722:** Add missing *"d"* in *"Land surface induced regional climate change..."*.

**References:**

Rahlves, C., Goelzer, H., Born, A., and Langebroek, P. M.: Historically consistent mass loss projections of the Greenland ice sheet, The Cryosphere, 19, 1205-1220, https://doi.org/10.5194/tc-19-1205-2025, 2025.

Haubner, K., Goelzer, H., and Born, A.: Limited global effect of climate-Greenland ice sheet coupling in NorESM2 under a high-emission scenario, EGUsphere [preprint], https://doi.org/10.5194/egusphere-2024-3785, 2025.

---

## Author Response (AR1)

Dear Editor,

We would like to thank you and both reviewers for taking the time to review our work and for the reviewers' valuable comments that have helped us to improve our manuscript.
We have attached detailed, point-by-point response letters addressing each reviewer's comments below. Compared to the first version of the manuscript, we have mainly:

- Addressed the necessary clarifications in the methodology section and throughout the manuscript overall as requested by the reviewers, and added additional references to the methodology and discussion sections.

- Addressed the fact that outlet glaciers and their dynamic retreat are not explicitly represented in our simulations in both the methodology and discussion sections (Sect. 2.1 and 4.3).

- Nuanced our findings and revised our text in Sections 3.6.1, 4.1 and Conclusions to better disentangle the impact on the radiation balance resulting from the (balancing) effects of clouds and other indirect effects associated with the melt-elevation feedback (Stefan-Boltzmann relation, thickening atmospheric column, increased water vapour).

- Rephrased our text in Sections 3.5, 4.1 and Conclusions to emphasize the expansion of the runoff season rather than the constraint imposed by the low solar elevation angle.

- Added a new Figure 6 and – after uploading our response letters – also a new paragraph discussing wind speed changes over the entire ice sheet and surrounding area in Section 3.3

- Adjusted Figure 10 depicting the ablation zone expansion to display the results on both the MAR and GISM grid; and updated Figure 12 to show the summer (June-August) mean radiative fluxes over the ice sheet instead of the annual means.

- Incorporated most of the reviewers' suggested minor adjustments to labels, titles and legends for the remaining figures.

Best regards,

Chloë, on behalf of all the co-authors.

**Review of the manuscript "Positive feedbacks drive the Greenland ice sheet evolution in millennial-length MAR–GISM simulations under a high-end warming scenario ", by Chloë Marie Paice et al.**

Paice et al. present the first coupled simulations of an ice sheet model and a regional climate model on a millennial timescale. To this end, they coupled the ice sheet model GISM to the regional climate model MAR, forcing it with IPSL-CM6A-LR global climate model output under the SSP5-8.5 scenario. They compare a zero-way (elevation changes are not taken into account), one-way (offline correction to consider elevation changes), and two-way (elevation changes are communicated between GISM and MAR) simulation, spanning the period 1990-3000. The analysis focuses on feedbacks arising from Greenland ice sheet elevation changes. The authors discuss several positive and negative feedbacks that play a role over different time scales and relate this to the surface mass balance and ice mass loss.

This study builds on the work done by Le clec'h et al. (2019) and Delhasse et al. (2024), in which MAR was coupled to the ice sheet models GRISLI and PISM. This study brings something new by presenting simulations spanning over a millennium. Over longer timescales, the effects of coupling complexity on ice mass loss become more pronounced, and feedbacks that play a role on longer timescales can be identified. This study presents a sound methodology and brings new and interesting results that will be valuable to the ice sheet modelling community. The paper is well written, and the results are presented in clear figures. My comments mainly concern the attribution of the differences between the simulations to the different feedbacks and the interpretation of climate interactions. After addressing these, I would recommend this manuscript for publication in The Cryosphere.

References:

Le clec'h, S., Charbit, S., Quiquet, A., Fettweis, X., Dumas, C., Kageyama, M., Wyard, C., and Ritz, C.: Assessment of the Greenland ice sheet–atmosphere feedbacks for the next century with a regional atmospheric model coupled to an ice sheet model, The Cryosphere, 13, 373–395, https://doi.org/10.5194/tc-13-373-2019, 2019.

Delhasse, A., Beckmann, J., Kittel, C., and Fettweis, X.: Coupling MAR (Modèle Atmosphérique Régional) with PISM (Parallel Ice Sheet Model) mitigates the positive melt–elevation feedback, The Cryosphere, 18, 633–651, https://doi.org/10.5194/tc-18- 633-2024, 2024.

We would like to thank the reviewer for their dedicated time thoroughly reviewing the manuscript and for their useful and constructive suggestions. We have carefully addressed all comments by the reviewer (in blue, with track changes below when applicable), and the manuscript will strongly benefit from the proposed changes.

**Major comments**

**The choice of the ice sheet model:** To me, it is unclear why you couple MAR to another ice sheet model, rather than using the existing MAR-GRISLI or MAR-PISM. Is this for computational reasons, or to explore the sensitivity to the choice of ISM? Can you add something about the choice for this model at the beginning of section 2 or in section 2.1?

Thank you for notifying us that this was indeed not addressed. GISM is the ice sheet model developed at the Vrije Universiteit Brussel (VUB) to which the 1[st] and 3[rd] author are affiliated. Besides, there are several other reasons why we opted for GISM as the ice sheet model in the coupling framework. We will clarify this by adding the following paragraph at the end of section 2.1:

*Compared to the previous two ice sheet models that were coupled to MAR, the advantages of GISM are its (new) possibility to combine a glacial-interglacial spin-up and data assimilation for its initialization, as explained below, and the fact that the higher-order model version can still run at relatively high resolution for the envisioned millennial timescale. Additionally, by coupling MAR with another ice sheet model, we can evaluate the robustness of the results compared to those obtained with GRISLI and PISM (Le clec'h et al., 2019b; Delhasse et al., 2024).*

**Section 3.3 on wind speed changes:** In this section, the reason for a different pattern in SMB changes between the simulations is attributed to changes in katabatic and barrier winds. Although I find these results very interesting, I do not think you can attribute all these differences to differences in wind patterns. I would also suggest looking into the effect of your offline correction on the differences in the near-surface air temperature changes at the margin. In your offline correction, you use an SMB gradient based on a climate computed at a higher elevation (since the MAR topography does not change). Using these SMB-elevation gradients, you implicitly assume some sort of relationship between temperature and elevation. When a grid cell becomes part of the ablation area in GISM, but not in MAR, the relationship you find based on the MAR SMB will not hold for GISM, because the energy required for melting reduces temperature increases. This (combined with the decreased winds) might lead to a too strong SMB-elevation gradient around the margins. Hence, your corrected SMB might correspond to a higher temperature than you would find in two-way coupled. You can also see this behaviour in Fig. 7a, where the strong increase in temperature at the high elevations is due to the initial global warming until 2300, the slower, but rather linear increase thereafter results from the temperature-elevation feedback, and the constant temperatures around zero for the lower elevations are because of the thermodynamic limit of melting ice. The timing of changing towards a new type of temperature-elevation relationship would then likely not be the same between MAR and GISM, leading to differences in SMB. I understand that you do not downscale temperature separately for your offline correction, but I would suggest thinking about this at least in a conceptual way.

Thank you for this interesting question and explanation. However, we do not think that the explanation regarding the temperature–elevation gradients applies here, as the largest SMB differences take place at the (current) ice sheet margin where the vertical SMB gradients are fully based on SMB values over the (present-day) ablation zone. Moreover, the ablation zone rapidly expands over the whole ice sheet after

2100, even in the offline simulation, and follows the same evolution on both the MAR and GISM grid for all three simulations as a result of the high-warming GCM forcing. We will update Figure 8 to show the ablation zone expansion on both grids. Therefore, the SMB–elevation gradients remain based on SMB values computed in the (offline) ablation zone where energy is used to melt the snow in MAR, even after a significant retreat of the ice sheet.

Hence, we believe that what we explain and show in this section and Figure 7 is sound, namely that the offline extrapolation does not remain valid at the ice sheet margins, even before 2300, and that ultimately this is due to wind changes in the 2wC simulation, that in turn lead to changes in other variables. After 2300, this effect or feedback effect persists, and additionally, the extrapolation starts to fail overall for the 1wC simulation because of the increasing differences between the MAR and GISM topographies in this simulation.

Regarding your statement on the increase of cloudiness derived from the increased long- and decreased shortwave radiation, I agree that the radiation changes are likely because of an increase in cloudiness, but you should consider a few other processes as well. The increase in water vapour in the column due to both the increase in temperature (Clausius-Clapeyron) and the elevation decrease leading to a larger atmospheric column will also lead to a decrease in shortwave and an increase in longwave radiation. Additionally, for the longwave radiation, increasing air temperatures affect the Stefan-Boltzmann relationship as well. Then, you attribute this increase in cloudiness to decreasing wind speeds, resulting in "cloud accumulation". Although this is a likely explanation for at least a part of this increased cloudiness, you should not forget about the increase in cloudiness due to a warming atmosphere (Clausius-Clapeyron), which you do mention in section 3.6.1, but should be mentioned here as well.

We fully agree with the reviewer regarding these other processes influencing the radiation balance and observed changes. However, the changes that we describe in this paragraph only concern the changes or differences that we observe at the ice-marginal zone compared to the ice sheet interior or area(s) further away from the margin. We are explaining the inversion of the SMB–elevation gradient at the margin, not the changes in cloudiness itself. What we meant here is that changes in cloudiness at the margin are a potential side-effect of the observed changes in wind speed. We do not observe abrupt changes in the LWD and SWD radiation (and therefore clouds) at the margin, as for wind speed. Therefore, we are confident that the changes in the radiative components are caused by changing wind speeds. We will rephrase our text to reduce ambiguity:

*As demonstrated by a transect from the ice sheet interior to the margin (Fig. 6), these changes in wind speed impact the surface energy balance and result in lowered runoff and a reduced or even inversed SMB–elevation gradient in the ice-marginal zone in the 2wC simulation. Figure 7 disentangles the impact of these changing wind speeds on the different radiation balance components. This is because the changing wind speedsMoreover, the increased katabatic winds lead to an accumulation of cold air along the ice-marginal zone (sudden slightly decreasing temperature and strongly decreasing sensible heat flux) and potentially clouds (continuously increasinged long- and decreasinged short waveshortwave downward radiation towards the margin) along the ice-marginal zone, as well as deposition or condensation onto the ice sheet behind the ice-marginal zone (increased latent heat flux). Decreasing barrier winds on the other hand reduce runoff in the ice-marginal zone (Fig. 7).*

For this section, it is also not completely clear to me whether you are only considering the period up to 2300 or the whole period. Initially, I was thinking about the whole period, since the figures 6 and 7 show this as well, but then lines 353-354 contradict this. So please state more clearly which period you are considering and if this is only a shorter period (e.g., till 2300), where to look in the figures.

We are indeed considering the whole period. The Figures show that the observed patterns and negative wind feedback at the margins persist for the entire period, i.e. 1990-3000. Yet, what we meant is that over time, throughout the 2wC simulation, the positive feedbacks become more important in terms of ice mass loss (i.e. the negative wind feedback does not counteract the observed positive feedback effects). We will clarify this in the text:

*After 2300, this negative wind feedback at the margins persists as Figures 6 and 7 show, but over time nevertheless  is overruled by the positive melt–elevation feedback and related changes in the atmospheric circulation amplify the ice mass loss everywhere in the 2wC simulation, as explained in the next sections.*

**Section 3.6.1 on clouds:** This section discusses the changes in cloudiness and cloud phase and their effect on the shortwave and longwave radiative effect. I am happy to see that you separate solid and liquid clouds and consider their distinct effects on radiation. I do think you are missing a few things regarding the cloud effect on radiation and the claim that clouds act here as a positive feedback. That there is no net effect on shortwave radiation does not mean there is no feedback effect of clouds on shortwave radiation. In fact, I would say that the negative feedback of clouds on shortwave radiation counters the positive melt-albedo feedback by reducing incoming shortwave radiation.

We agree with the reviewer regarding the (balancing) effects of clouds and albedo on shortwave radiation, and we feel like we had addressed this in sections 3.4, 3.6.1, and 4.1 albeit less explicitly than stated here. We will update the text to better disentangle and address the effects of these different processes (see also comment hereafter).

Section 3.6.1:

*Regarding the increase in cloudiness, and the changing ratio of solid and liquid clouds with differing longwave emissivity, severaltwo positive feedback effects can be observed over time, that are stronger for the 2wC compared to the 1wC and 0wC simulations. Moreover, following the Clausius – Clapeyron relation, specifying that a warmer atmosphere can hold more water, the rising air temperatures lead to an increase in water vapour and clouds in the Arctic. Together with the decreasing ice sheet surface elevation and subsequent thicker atmospheric column, this leads to a negative feedback on or decrease in incoming shortwave downward (SWD) radiation. bBut, due to the increasing number of bare ice days per year and the reduced surface albedo, the absorbed SWD at the surface remains relatively stable (Fig. 110).*

Additionally, the observed changes in the radiative fluxes are the sum of changes in more components than just clouds. To make a definite statement towards the sign of the cloud radiative effect and feedbacks, you would need to disentangle the effects of clouds, water vapor, temperature and albedo, which would be very complex using the set of simulations you did and involve a lot of assumptions. Therefore, I suggest toning down and mentioning that there are more components influencing the radiation budget. Also consider this in lines 542-548 in the discussion. Specifically, "longwave warming outweighing cooling associated to the reduced incoming shortwave radiation" would be

a too strong statement since you cannot separately compute and compare the longwave and shortwave radiative effects from your results.

We would like to thank the reviewer for this useful comment, that serves to disentangle different feedback effects and better explain them in the manuscript. We agree that the changes in the radiative fluxes are the result of more components than just clouds and that it is very difficult to quantify all the separate effects. We have therefore better addressed this in our discussion. Nevertheless, we believe that it can still be stated based on our results that the warming effect of the increased LWD radiation outweighs the cooling effect of the decreased SWD radiation in our 2wC simulation, though it is right that these changes cannot only be attributed to the changes in clouds. In total however, when considering all the above-mentioned effects on the radiative budget, the changes in clouds and the melt–elevation feedback do amplify one another here. Hence, we suggest to rewrite this paragraph in the discussion to nuance our previous statements.

Section 4.1:

*In addition,  important changes are observed in the radiative budget, whereby a distinction can be made between the effects on the SWD and LWD radiation. Firstly, our results show that the negative feedback of increased clouds and water vapour (through the Clausius-Clapeyron relation and thickening atmospheric column) on the SWD radiation balances the positive melt-albedo feedback by reducing incoming shortwave radiation. In other words, the net absorbed shortwave radiation remains stable as the albedo decreases. Thus, the cooling effect from the reduction in SWD radiation is outweighed by the warming effect of the stronger increase in LWD radiation. The latter is the combined result of the increasing atmospheric temperatures (Stefan-Boltzmann relation) and the increase in water vapour and mainly liquid clouds with a higher longwave emissivity than solid clouds.  Hence, though it is difficult to quantify all the effects separately, altogether the changes in clouds and the positive melt–elevation feedback amplify one another in our 2wC simulation. Though this seems to contrast with other long-term modelling studies that do not regard the impact of cloud changes on the longwave radiation (Gregory et al., 2020; Feenstra et al., 2024), it aligns with earlier findings regarding the impact of clouds on the Greenland near-surface climate and surface energy balance (Franco et al., 2013; Vizcaíno et al., 2014; Van Tricht et al., 2016; Hofer et al., 2019; Lenaerts et al., 2020).*

And additionally, we will rewrite the conclusions accordingly:

*In addition, the rising atmospheric temperatures, and increasing water vapour coincide with an increase in mainly liquid clouds that further increase the   runoff through amplified LWD radiation. Though at the same time the increased clouds and water vapour act as a negative feedback effect on the SWD radiation, this effect is balanced by the positive melt–albedo feedback. Hence, altogether the positive feedback effects on the radiative budget prevail.*

**Minor/technical comments**

p1, line 9: "predicting" -> "projecting"
Ok.

p2, line 56: remove "and winds", since you only discuss this in the next paragraph.
We will.

p3, line 84-85: "It is the first time ice sheet–atmosphere interactions are accurately accounted for on the millennial timescale." I would tone down here. As you mention, there are a number of studies that presented two-way coupled simulations on millennial timescales before, but with lower resolution or complexity. Therefore, these MAR-GISM simulations with higher resolution and/or complexity than used in these other studies will add a lot to our understanding of feedbacks, but 30 km is still a rather coarse resolution (as you mention in the discussion), and both MAR and GISM have their limitations as well. I would suggest highlighting that your simulations are run on the highest resolution or that your model has the highest complexity, rather than stating that they "are accurately accounted for".
We thank the reviewer for their rightful comment and will update the text accordingly:
*It is the first time ice sheet–atmosphere interactions are  accounted for using an RCM on the millennial timescale.*

p4, line 109: "using the higher-order approximation", what kind of approximation is this? Could you mention it briefly?
Thank you for stipulating that this was not described in the text. We will therefore clarify it further (and will update the reference list):
*Though  different approximations to the force balance equations governing ice flow can be considered, the presented simulations are performed using the higher-order approximation, which includes multilayer longitudinal stresses and lateral horizontal shearing. It is classified as a LMLa higher-order model or Blatter-Pattyn model (Blatter et al., 1995; Pattyn, 2003; Hindmarsch, 2004) and is described in detail in Fürst et al. (2011). It is complemented by a simplified equation to describe the basal resistance (called SR HO in Fürst et al., 2013) in the basal sliding formulation.*

p4, line 116-117: What do you mean by "acceptable level of discrepancy between the model resolutions"? Do you mean that the resolutions should not be too different, and it would not be ideal to run GISM on a much higher resolution?
That is indeed what we mean. We will make this clearer as follows:
*Besides, ideally the GISM resolution should not be too high with respect to the MAR resolution,  to maintain a reasonable level of discrepancy between both model resolutions  throughout the coupled simulations, and to facilitate the efficient initialization of the ice sheet and coupled model into an equilibrium state resembling present-day observations.*

p6, line 170-174: Can you briefly explain what these optimization and relaxation steps are?
Thank you for pointing out that we did not explicitly mention the purpose of both steps. We will add the following brief explanations:
*It consists of an optimization step, during which the basal sliding coefficient (BSC) and*

*enhancement factor (EF) are updated to match the modelled ice thickness to observations. This is followed by a relaxation step, during which the ISM is run in free geometry mode with the inferred BSC and EF fields, to minimize any remaining model drift. that Both steps are repeated until the root mean square error (RMSE) between the modelled and observed ice thickness no longer improves significantly (Le clec'h et al., 2019a).*

p10, line 266: several hundred metres around the margins, where the ice thickness is small, is quite a large relative difference. Do you know what causes these differences (particularly in the central east)? How does it affect your results?

We agree that several hundred metres is indeed a relatively large difference. However, this is in line with expectations and can be explained as follows:

*As can be seen on Fig. 2, the differences between the initialized and observed present-day topography (Morlighem et al., 2017) are generally less than 40 m and up to several hundred metres around the steep margin, where a minor misalignment of slope can cause large differences between the modelled and observed ice thickness. Besides, the observed topography is very irregular around the ice sheet margins and the central east in particular, which is marked by coastal mountain ranges with deep valleys in between, and an ensuing high spatial variability in ice thickness. As also reported by Le clec'h et al. (2019a), it is therefore most difficult to obtain a well-constrained ice thickness in these areas.*

This does not significantly influence our results, since the overall present-day ice sheet geometry is well-represented (without the need for flux corrections or forced ice removal at the ice sheet boundaries) and the MAR SMB at the start of our simulations is the one over this initial GISM topography thanks to our iterative MAR-GISM initialization.

p12, line 275-276: I am a bit confused about the definition of the GBI region. It makes sense to show the temperature evolution over this region, but since you are not considering the GBI in your analysis, I think it only adds confusion. You could just refer to it as "Greenland region" or something similar, with the definition of the coordinates.

Thank you for indicating this confusion. We will change it with the following replacement in the text and in the subscript of Figure 3:

*Greenland Blocking Index (GBI) region → larger Greenland region*

with indication of the coordinates in-text.

p13, fig 3: same for GBI region, and add that the MAR 2m mean annual air temperature is for the two-way simulation.

Ok. The latter was implied by "the retreating ice mask" but we will explicitly mention it for clarity:

*… the MAR 2 m mean annual air temperature over the retreating ice mask (in °C), i.e. for the 2wC simulation, …*

P15, line 310-313: I do not see a clear overestimation of SMB in one-way over the interior.

We agree that this overestimation is not very apparent in the Figure. This is because the order of magnitude is much smaller (~10 to 20 mm w.e.) than for the underestimation at the edges (order of 1 to several metres w.e.). We will address this in the text as follows:

*Especially around 2200 it becomes clear that a spatial compensation of differences within the SMB fields is at play, with ablation in the 1wC simulation being overestimated (by 1 to >3 m w.e.) compared to the 2wC simulation within 60 km from the ice sheet margins (i.e. two MAR grid cells), referred to hereafter as the ice-marginal zone, and slightly underestimated*

*over the interior compared to the 2wC simulation (by generally 10 to 20 mm w.e.).*
*(…)*
*In other words, theis over- and (weaker but more widespread) underestimation of SMB on the 5 km GISM grid can be linked to the offline extrapolation that falls short at the ice sheet margins over time.*
And further down:
*This explains the over- and (slight) underestimation of ablation with respect to the 2wC simulation and the ensuing similar ice sheet contribution up to 2300.*
In addition, note that we have also referred to it in our description of Figure 6, where it is easier to see:
*In addition, Fig. 6 demonstrates that even throughout the 2wC simulation, the extrapolated SMB on the 5 km grid is too negative compared to the SMB on the original 30 km grid when the grid cell is at the ice sheet margin (reduced colour saturation) and slightly overestimated when it is part of the ice sheet interior (full colour saturation).*

p15, line 313-315: I had to read this multiple times to understand what you are referring to. Can you rephrase in such a way that it is clear that you mean the change of sign of the difference between the GISM SMB between one-way and two-way, which does not occur for the MAR SMB?
Thank you for indicating that this part was difficult to understand. We will reformulate it more straightforwardly:
*However, as Fig. 5b demonstrates, the magnitude and sign of this ablation difference between the 1wC and 2wC SMB on the 5 km GISM grid over- and underestimation of ablation not only varyies spatially but also varyies over time. Besides, it clarifies that this is only the case for the SMB extrapolated onto the 5 km GISM grid, not for the original SMB on the 30 km MAR grid. On the other hand, on the 30 km MAR grid, the ablation is always more negative for the 2wC simulation and the difference with the 1wC simulation is ever larger over time. In other words, theis over- and (weaker but more widespread) underestimation of SMB on the 5 km GISM grid can be linked to the offline extrapolation that falls short at the ice sheet margins over time.*

p16, line 329: decreased barrier winds, compared to what? I assume this is a decrease over time and only observed in two-way?
Indeed, that is exactly what we mean. We will specify it as follows:
*Similar to Delhasse et al. (2024), over time we observe decreased barrier wind speeds in the 2wC simulation within the (30 km to 60 km broad) ice-marginal zone and increased katabatic wind speeds behind this zone further inland because of the retreating margin and steepening slopes.*

p16-17, line 326-335: How do you define barrier and katabatic winds here? You describe the maximum changes in wind speed; are these the locations with the largest change? I think it could be very interesting to add some plots showing the (changes in) katabatic and barrier wind speeds over the ice sheet. Or otherwise, at least describe where these changes are most pronounced, or whether the behaviour is very similar along the whole Greenland margin.
Katabatic winds consist of cooled dense air masses above the ice sheet surface that accelerate down the ice sheet slopes. Barrier winds arise when the geostrophic wind at the top of the atmospheric boundary layer is directed from the warmer tundra towards the colder ice sheet surface and is confined by the ice sheet acting as a topographic barrier, resulting in a jet blowing (clockwise) along the ice sheet edge. The katabatic and barrier winds can thus be distinguished based on their direction

relative to the ice sheet.

The described maximum changes in wind speed indeed refer to the locations with the largest change. We will add a map showing these overall wind speed changes over the entire ice sheet for the 2wC versus the 1wC and 0wC simulations and refer to it in the text.

p17, line 339-340: I do not see this sudden decrease in temperature in Fig. 7a.

We agree that the decrease in temperature is not very large for most depicted grid cells and that the "sudden" or substantial decrease mainly applies to the sensible heat flux. We will therefore remove the word "sudden" here.

p18, Fig. 6c: It is not clear to me what all these lines are. If you plot one line for the MAR SMB and one for the GISM SMB for 9 locations, you would end up with fewer lines than you have now. Also, the reduced colour saturation makes the lines with small markers quite hard to see on the left side.

These lines depict the values for all the corresponding GISM grid cells, which are more numerous than the number of MAR grid cells, as one 30 km grid cell corresponds to roughly six GISM grid cells at 5 km resolution. To avoid confusion, we will explicitly mention this in the figure description:

*For the SMB (c), both the values for the 30 km MAR grid cells as well as the extrapolated values for the corresponding 5 km GISM grid cells (small markers) are shown. Note that the number of corresponding GISM grid cells is higher than the number of MAR grid cells, as one 30 km grid cell corresponds to roughly six GISM grid cells at 5 km resolution.*

To improve the readability of the figure, we will increase the size of panels b and c with respect to a, and we will slightly increase the size and saturation of the small markers.

p19, Fig.7: longwave and shortwave instead of long wave and short wave. There are also multiple instances of this in the text (e.g., line 340 and section 3.6.1).

We will update all instances.

p20, line 373: Please state whether you consider the MAR or GISM SMB and refer to Fig. 11.

We would like to thank the reviewer for their observant comment. The SMB on both the MAR and GISM grid is indeed displayed in Figure 11. We will move it upwards in the text to refer to it here, and will change the figure numbering of Figures 8-11 accordingly.

p20, line 376-379: Is the bare ice albedo in MAR a fixed value?

The bare ice albedo in MAR is not exactly fixed, though the variations are small:

*This implies that there is practically no difference in the positive melt–surface albedo feedback between all three simulations, as most of the ice sheet surface consists of bare ice during the Arctic summer in all three simulations (Fig. 10). The difference in albedo (that varies between 0.50 and 0.55 for bare ice in MAR depending on the presence of surface meltwater) and absorbed incoming solar radiation at the ice sheet surface between simulations is therefore negligible (Fig. 11).*

p23, Fig. 10: I would suggest showing the summer fluxes instead of the annual mean, since the melt mainly happens in summer.

We agree to show the summer (June-August) fluxes and will update the figure.

p24, line 411: mean surface elevation -> mean surface elevation decrease

Ok.

p24, line 413: "The time of this intensification", I assume this is 2500?

Indeed. We will add this to the sentence for clarity:

*At the time of this intensification, around 2500, the mean annual 2 m air temperature…*

p25, line 425: What is meant by "entire Arctic summer"? From the context, I would understand this is >150 days. Please define Arctic summer or mention that you refer to the yellow line in the plot.

We were referring to the (extended) summer period of 120 runoff days earlier in the sentence, but agree that it may sound ambiguous and will therefore remove it:

*Around 2500, an accelerated expansion can be observed in the ice sheet area exhibiting runoff during at least 120 days per year , or, during the entire Arctic summer (Fig. 12).*

p25, line 428-429: Do you know why the areal expansion for runoff days is so different from the number of bare ice days?

Thank you for this interesting question. Based on the observed changes in precipitation and density of the upper ice layers we can conclude that the rapid areal expansion of bare ice days is linked to changes in snowfall. Already by 2200 most of the snowfall melts before densifying to firn in all simulations (section 3.4), and there is hardly any (summer) snowfall remaining (at most 4.5 % by 2200 and 2.4 % by 2300 in all simulations) (section 3.6.2). Nevertheless, the slower expansion of runoff days is linked to the available energy for melt, that remains too low over most of the ice sheet area to generate melt and runoff throughout a large part of the year due to the low solar elevation angle, especially during the winter months (section 3.5).

p25, line 431-436: I do not think this is very new. With no incoming shortwave radiation for a large part of the ice sheet during winter, it is no surprise that there is no energy available for melt. Therefore, I would focus more on the extension of the melt season, which you did in the previous paragraph, and consider shortening or removing this paragraph.

A low amount of incoming shortwave radiation will indeed naturally lead to a low amount (of energy available for) melt. However, what we wanted to stress is that though the melt season keeps expanding in the 2wC simulation, this expansion is physically constrained by the low solar elevation angle in winter. We suggest to rephrase the paragraph accordingly:

*Nevertheless, the proportion of June–August runoff remains 68 % or more of the annual total, compared to 94 % at the start of the simulations and 76 % by 2300 in the 2wC simulation. This is because even after 2300 the available energy during the Arctic winter months is hardly enough to melt the fresh snow layer over most of the ice sheet area. In other words, though the melt season keeps expanding, the number of runoff and bare ice days per year and therefore the strength of the melt–elevation feedback remain physically constrained by the low solar elevation angle (Fig. 10 and 12).*

p22, Fig. 9 and p26, Fig. 12: Please make the colours consistent between the plots. Now, in Fig. 9, the shortest number of days is in red, and the longest in blue, while for Fig. 12, it is the other way around.

Thank you for pointing out that the colour scale of both figures was accidentally still each other's inverse. We will rectify this.

p27, line 459-460: A thicker atmospheric column also results in changes in shortwave radiation. Additionally, you should consider the effect of the Clausius-Clapeyron relationship here as well. It does not only result in an increase in cloudiness, but in water vapour as well, influencing radiation. For longwave, temperature changes can have an influence as well through the Stefan-Boltzmann relationship.

We agree with the reviewer that we could have been more specific here and should have mentioned the impact of the Stefan-Botzmann relation on the longwave radiation. We will update the paragraph accordingly (same as above):

*Regarding the increase in cloudiness, and the changing ratio of solid and liquid clouds with differing longwave emissivity, several two positive feedback effects can be observed over time, that are stronger for the 2wC compared to the 1wC and 0wC simulations. Moreover, following the Clausius – Clapeyron relation, specifying that a warmer atmosphere can hold more water, the rising air temperatures lead to an increase in water vapour and clouds in the Arctic. Together with the decreasing ice sheet surface elevation and subsequent thicker atmospheric column, this leads to a negative feedback on or decrease in incoming shortwave downward (SWD) radiation. bBut, due to the increasing number of bare ice days per year and the reduced surface albedo, the absorbed SWD at the surface remains relatively stable (Fig. 1_10).*

*(…)*

*As can be seen on Fig. 13, after 2300, the increase in longwave downward radiation continues and by 3000 it has increased 26 % more compared to the 1wC and 0wC simulations over the same retreating ice mask. Again, Tthis is due to both the increase in cloudiness and the reduced surface elevation that results in a thicker atmospheric column above the surface. In addition, the and the consequently enhanced longwave radiation can also be attributed to the Stefan-Boltzmann relation, stating that the total radiated energy by a body or matter is directly proportional to its temperature to the fourth power.*

p28, Fig. 13: Do you show the increase in cloud cover or cloud water path here? Please mention this. In case you used cloud cover, I would suggest looking into the water path, since this likely tells you more about the amount of clouds and their radiative effect than just cloud cover. An increase in cloud water path might not be reflected in the cloud cover, but it still has large effects on radiation.

Thank you for mentioning that this was not specified. We are showing the changes in cloud water path and will update the axis label and figure description:

*Figure 13: Percentual iIncrease in solid clouds (ice water path), liquid clouds (water vapour path) and the sum of both (a) (solid, liquid and the sum of both), percentual increase in long wavelongwave downward (LWD) radiation (b), and multiplication factor of runoff (c) with respect to the 30 year mean at the start of the simulations (1990–2019). Depicted is the 30 year running mean over the retreating ice mask for the 2wC simulation, and over the fixed as well as the same retreating ice mask for the 1wC and 0wC simulations.*

p30, Fig. 14: Why do you show the grey area in this figure specifically, but not in many of the others? I would either show this in all timeseries or only in Fig. 3.

We had included the grey area in this figure specifically to highlight that the large

variability in precipitation stems from the large variability during the randomly sampled period to prolong the simulations (2250-2300). We will remove it from the figure and will only mention this in the figure description:

*Note that the large variability after 2300 results from the large variability in precipitation during the*  *years that were randomly sampled (2250 to 2300) to prolong the IPSL-CM6A-LR forcing for MAR* .

p30, line 507: "This corresponds to a reduction of -42 % in total snowfall and -38.4 %" I think you can get rid of the minus signs, since a reduction already implies it is negative.

We agree that the minus sign is in fact redundant and will rephrase this:

*This corresponds to a reduction of* *42 % in total snowfall and* *38.4 % in total precipitation over the ice sheet between 2300 and 2500,*  *with further decreases reaching*  *90 % and* *86 %, respectively, by 3000.*

p33, line 531-537: Are there other studies that looked into the inland displacement of precipitation and the diminishing orographic barrier? I particularly find your results on the diminishing orographic barrier very interesting, and I would be interested in knowing whether this was found before.

We sincerely thank the reviewer for their question and interest. Thus far, we are not aware of other studies that have reported a decrease in total precipitation due to the diminishing orographic barrier effect of the GrIS on the continent-wide scale. The inland displacement of precipitation on the other hand has often been reported before. We should indeed have mentioned this.

*Besides, beyond 2300, in the 2wC simulation the precipitation*  *also continues to move further inland following the retreating ice sheet margin, as often reported before (Toniazzo et al., 2004; Ridley et al., 2005; Vizcaíno et al., 2010; Solgaard and Langen, 2012; Gregory et al., 2020; Andernach et al., 2025; Feenstra et al., 2025).*

p33, line 539-540: I do not think that the fact that the melt-elevation feedback is constrained by the solar elevation angle is a new finding of your study. The same comment for the conclusion, p36, line 650-652. There, you also state that this is even the case under high-warming scenarios, but the solar elevation angle is independent of the warming scenario.

Of course, the solar elevation angle is independent of our applied warming scenario. What we mean however, is that even for this high-warming scenario, wherein the annual mean temperature rises by 15°C already by 2300, less than 30 to 40% of the (remaining) ice sheet experiences (more than) 150 runoff days. And that this is due to the low solar elevation angle, which is thus a physical constraint on the strength of the melt–elevation feedback. We will rephrase the respective paragraphs:

*A*round 2500, *though its strength remains constrained by the low solar elevation angle during the Arctic winter months, the melt–elevation*  *feedback intensifies.*

*Around 2500, the positive melt–elevation feedback intensifies.* *W*e observe a decrease in (solid) orographic precipitation and the summer runoff expands more rapidly in both space and time once (more than) 40 % of the remaining ice

*sheet area experiences runoff during at least 120 days, or a prolonged summer period.*

p33, line 544: associated to -> associated with
Ok.

p34, line 567-570: "not including the melt-elevation feedback" Are you referring to the zero-way or one-way simulation? Also, you find a larger underestimation than the three other studies. Is this because you have stronger warming by 2200 in your simulations? Or could this be attributed to the choice of ISM? If the warming is very different, you could consider comparing SLR contributions linked to a certain amount of warming instead of timing?

We are referring to the 0wC simulation and have now specified this. Our underestimation is indeed larger than for the other three studies because in our simulations the warming continues up to 2200 as opposed to these other studies, wherein the warming only continues up to the year 2100 with a constant climate after that time. Given the continued strong warming in our simulations after 2100, we consider this as the main reason for the difference rather than the ISM uncertainty. We will reformulate this part in the text for unambiguous interpretation and will leave out the reference to Edwards et al. (2014), since in hindsight not only the applied warming but also the different methodology makes it difficult to compare the numbers as is.

*We find an underestimation of the sea level contribution of 10.41 % by 2150 or 14.41 % by 2200 when not including the melt–elevation feedback (i.e. 0wC simulation). This is somewhat higher than the ~9.3 % by 2150 reported in Le clec'h et al. (2019b), and the 10.5 % by 2200 reported in Delhasse et al. (2024), since in these  studies  a stabilized climate was assumed to extend the simulations  beyond 2100.*

The suggestion of linking the SLR to a certain amount of warming would be very interesting. However, this is not straightforward either, since the sea level contributions for the different experiments are not reported for or linked to different temperature forcings in the different studies, and neither are the forcing temperatures reported in a consistent fashion, over a comparable mask or at the same atmospheric level for example.

p34, line 574-575: I suggest rephrasing this. Since you do not downscale winds, it makes sense that your offline correction does not capture this feedback. I would rather say something about the fact that you need two-way coupling to include this effect.

Thank you for mentioning this. In a way this indeed makes sense. Yet, when annually recalculating local SMB–elevation gradients, to some extent one would expect to capture indirect effects on the melt or SMB during the downscaling. But this was shown not to be the case, already for the first part of the simulation. We thus suggest to rephrase our sentence such that it better reflects what we mean:

*In addition, our results illustrate that the Franco et al. (2012) method for extrapolating the SMB from the lower resolution RCM grid to the higher resolution ISM grid by means of locally derived SMB–elevation gradients does not fully capture indirect effects on the SMB, like the effect of changing winds, that act as a negative feedback in our 2wC simulation.*

p34, line 585-586: How well can you resolve wind speeds at 30 km resolution? Is this resolution high enough to capture the katabatic and barrier winds realistically?

Yes, the 30 km resolution is high enough for this. As for example shown in Delhasse et al. (2020), winds are well resolved by ERA5 (~31 km) and ERA-Interim (~41 km) reanalysis products, with similar spatial resolutions as our MAR simulations. Furthermore, the katabatic and barrier winds were already resolved using a simplified 2D version of MAR by van den Broeke and Gallée (1996).

Delhasse, A., Kittel, C., Amory, C., Hofer, S., van As, D., S. Fausto, R., and Fettweis, X.: Brief communication: Evaluation of the near-surface climate in ERA5 over the Greenland Ice Sheet, The Cryosphere, 14, 957–965, https://doi.org/10.5194/tc-14-957-2020, 2020.

van den Broeke, M.R. and Gallée, H.: Observation and simulation of barrier winds at the western margin of the Greenland ice sheet. Q.J.R. Meteorol. Soc., 122, 1365–1383, https://doi.org/10.1002/qj.49712253407, 1996.

p35, line 593-598: This paragraph is a bit vague, especially the last sentence. I would consider removing this paragraph entirely and maybe moving the first sentence to the paragraph before.

Thank you for this suggestion. We will remove this paragraph and incorporate the first sentence with the previous paragraph.

p35, line 601-603: I would not consider the increasing difference between the MAR and GISM SMB when using a higher GISM resolution a big problem. I think the computational constraint justifies the choice for the 5 km resolution already well enough.

We agree that the computational constraint is the main reason for the 5 km GISM resolution and justifies it. However, based on our results it seems reasonable to assume that (with the applied or similar methods for extrapolation) the discrepancies between the original and extrapolated SMB will increase with increasing difference in model resolutions. It is thus an additional rationale for our and potential future coupled model set-ups that we would like to mention.

p35, line 609: remove "though"
Ok.

p35-36, line 620-626: I would consider moving this to line 606, since this paragraph discusses limitations of your ISM.

Thank you for this suggestion. We will move this paragraph upwards below the former paragraph that discusses the ISM limitations.

p36, line 622: remove the comma here.
Ok.

p36, line 636-642: Are winds the only reason for this difference?

Assuming that the reviewer refers to the spatial differences in SMB, we feel like we have addressed this in detail in section 3.3, where we argue that despite the similar integrated ice mass loss between simulations, clear spatial differences in SMB can be observed, that we could indeed relate to the changing wind speeds (and the ensuing changes in surface energy fluxes).

Regarding the similar integrated ice mass loss, in section 3.4 we presented the lack of difference in the positive melt–surface albedo feedback between simulations as another cause for the small difference in integrated ice mass loss. We will add this to the paragraph, as we did not repeat it here:

*We find that the ice sheet contribution by 2300 differs by only 46.9 cm s.l.e. or 2.36 % between the simulations with different coupling complexity.* This small difference can in part be attributed to the similar bare ice cover during the melt season, and the ensuing lack of difference in the positive melt–albedo feedback between the simulations. *Nevertheless,* in addition, *distinct spatial differences in SMB were observed* between the 1wC and 2wC simulations, that largely compensate one another and thus lead to a similar integrated ice mass loss.  These SMB differences *can be attributed to changing near-surface wind speeds that reduce ablation along the ice sheet margin in the 2wC simulation, and the fact that this negative feedback effect is not adequately represented by the offline extrapolation.*

p40, line 741-743: Feenstra, et al. (2024) is not in discussion anymore: https://doi.org/10.5194/tc-19-2289-2025

Thank you for pointing this out, we will update the reference list accordingly.

**References**
Below, we have inserted the references that we will add to our reference list based on the updates above.

Blatter, H.: Velocity and stress fields in grounded glaciers: a simple algorithm for including deviatoric stress gradients, J. Glaciol., 41, 333–344, https://doi:10.3189/S002214300001621X, 1995.

Fürst, J. J., Rybak, O., Goelzer, H., De Smedt, B., de Groen, P., and Huybrechts, P.: Improved convergence and stability properties in a three-dimensional higher-order ice sheet model, Geosci. Model Dev., 4, 1133–1149, doi:10.5194/gmd-4-1133-2011, 2011.

Hindmarsch, R. C. A.: A numerical comparison of approximations to the Stokes equations used in ice sheet and glacier modelling, J. Geophys. Res.-Earth, 109, F01012, https://doi.org/10.1029/2003JF000065, 2004.

Le clec'h, S., Quiquet, A., Charbit, S., Dumas, C., Kageyama, M., and Ritz, C.: A rapidly converging initialization method to simulate the present-day Greenland ice sheet using the GRISLI ice sheet model (version 1.3), Geosci. Model Dev., 12, 2481–2499, https://doi.org/10.5194/gmd-12-2481-2019, 2019a.

Pattyn, F.: A new three-dimensional higher-order thermomechanical ice sheet model: Basic sensitivity, ice stream development, and ice flow across subglacial lakes, J. Geophys. Res.-Sol. Earth, 108, 2382, https://doi:10.1029/2002JB002329, 2003.

Solgaard, A.M.,, and Langen, P.L.: Multistability of the Greenland ice sheet and the effects of an adaptive mass balance formulation, Clim. Dyn., 39, 1599–1612, https://doi.org/10.1007/s00382-012-1305-4, 2012.

Toniazzo, T., Gregory, J., and Huybrechts, P.: Climatic impact of a Greenland deglaciation and its possible irreversibility, J. Climate, 17, 21–33, https://doi.org/10.1175/1520-0442(2004)017<0021:CIOAGD>2.0.CO;2, 2004.

Vizcaíno, M., Mikolajewicz, U., Jungclaus, J., and Schurgers, G.: Climate modification by future ice sheet changes and consequences for ice sheet mass balance, Clim. Dyn., 34, 301–324, https://doi.org/10.1007/s00382-009-0591-y, 2010.

**Review of egusphere-2025-2465: "Positive feedbacks drive the Greenland ice sheet evolution in millennial-length MAR–GISM simulations under a high-end warming scenario" (Chloë Marie Paice et al.)**

**Summary**

Paice et al. present a suite of coupled ice-sheet–atmosphere simulations of the Greenland Ice Sheet (GrIS), spanning the period from the present to the year 3000. The regional climate model MAR is coupled to the ice-sheet model GISM. The simulations are initialized through a glacial–interglacial spin-up of the ice sheet model (ISM), followed by a data assimilation procedure to obtain a realistic present-day ice-sheet geometry. Three forward simulations with varying coupling degrees are then performed under a high-emission scenario: (i) a zero-way coupled run, in which MAR-derived surface mass balance (SMB) is applied directly to the ISM without accounting for evolving surface height; (ii) a one-way coupled run, in which the SMB–elevation feedback is parameterized; and (iii) a two-way coupled run, in which annual changes in ice-sheet geometry are passed from GISM to MAR. Results from these experiments are used to systematically assess key atmospheric feedback processes affecting ice-sheet evolution, including the SMB–elevation feedback, wind, cloud, and albedo effects. The authors find that positive feedbacks (e.g. the SMB-elevation feedback) dominate on the millennial timescale and substantially enhance future ice mass loss.

This study represents the first millennial-scale two-way coupling of an ISM with a regional climate model (RCM), making the manuscript both novel and highly relevant for advancing our understanding of Greenland's long-term atmospheric feedback processes.

**General Assessment**

This manuscript is highly relevant to the field of ice-sheet modelling and makes an original contribution by advancing our understanding of long-term atmosphere–ice-sheet interactions over Greenland. The paper addresses important scientific questions that fall well within the scope of *The Cryosphere*. The authors present novel methods and results, in particular the two-way coupling of an ice-sheet model with a regional climate model over millennial timescales, which, to my knowledge, has not previously been demonstrated.

The title accurately reflects the content of the paper. The scientific methods and assumptions are valid and are generally well described in the methods section, though I suggest some refinements and clarifications to improve clarity, as well as the addition of

visual aids. The conclusions are substantial and well supported by the presented results. I particularly appreciated the systematic approach to disentangling and presenting the various feedback processes that contribute to future mass change of the ice sheet. The manuscript is well structured and generally clearly written, though in some instances sentences are awkwardly formulated and the logical connections within paragraphs are not always transparent.

I recommend publication after minor revisions, mainly to improve language, grammar, and precision, and to enhance clarity in the methods section.

We would like to thank the reviewer for taking the time to evaluate our work and for providing helpful and constructive feedback. We have carefully addressed all comments (in blue, with track changes below where relevant) and believe the suggested revisions will significantly improve the quality of the manuscript.

**General Comments**

We thank the reviewer for their thorough assessment and general comments. Given that (especially) the first and third general comments are accompanied by or further divided into specific comments, we have carefully addressed these in the next section.

1. Although the methods are generally well described, some details remain unclear. I recommend providing additional precision and detail where necessary (see specific comments). In addition, a visual aid would be valuable to guide the reader through the initialization steps and the interactions between MAR and the ISM (see also comment on Sect. 2.3).

2. While the manuscript focuses on atmosphere–ice interactions, I suggest explicitly addressing the treatment (or non-treatment) of the ice–ocean boundary in both the methods and discussion sections. In particular: How are outlet glaciers and their potential retreat represented in the simulations? If dynamic retreat is not included, how do the authors justify this omission, and how might they expect dynamic retreat to interact with SMB-driven retreat? What limitations does this introduce to the interpretation of results? I believe this point is important, since dynamic retreat of outlet glaciers contributes substantially to present-day mass loss of the GrIS and is expected to remain significant in the future, at least as long as the ice sheet is in contact with the ocean.

   Thank you for this interesting question, which raises an important point.
   The outlet glaciers and their potential or dynamic retreat are not explicitly considered in our presented simulations. We acknowledge that dynamic outlet glacier retreat is an important part of the Greenland ice sheet contribution at present and in the coming centuries as long as the ice sheet is in contact with the ocean. Yet, given our focus on long-term ice sheet – atmosphere interactions and feedbacks (rather than specific sea level contribution in the coming centuries), the ice – ocean interactions are not of interest for the current study.
   We agree to mention and address this in the methods and discussion sections to improve them (see specific comments below).

3. The manuscript is generally well written and well structured, but some sentences are awkwardly phrased, and the logical connections are not always clear. I believe a careful revision of the language, sentence structure, and causal links within sentences and paragraphs would improve clarity and linguistic precision.

**Specific Comments**

**l. 22:** To avoid misinterpretation, I suggest to already clearly state in the abstract that glacial isostatic adjustment is not considered, when mentioning numbers of sea-level contribution.

Thank you for this suggestion. Since we have addressed this matter both in the Methodology and Discussion sections and have specified in the latter why we think the glacial isostatic adjustment will not substantially impact the rapid ice sheet decline and sea level contribution by the end of the millennium, we believe that this is not a major issue to be addressed in the abstract specifically. Nevertheless, we agree with the suggestion regarding l. 163 to reiterate the disabling of the glacial isostatic adjustment in Section 2.5.

**l. 33:** The expression "*… the major remaining uncertainties…*" is a bit unclear. Please specify what "*remaining*" is referring to.

"Remaining" uncertainties refers to the uncertainties that persist or are unknown up to present. We feel that we have explained these in the following two sentences, i.e. that it remains difficult to quantify the effects of ice sheet-climate interactions, or identify the positive or negative nature of several feedback effects on ice mass loss.

*This can mainly be attributed to  major remaining uncertainties regarding ice sheet–climate interactions and feedback mechanisms, that will determine the ice sheet's long-term mass loss. Although many of these interactions and feedbacks have been identified and characterized for some time, quantifying their effects remains challenging (Fyke et al., 2018). Moreover, for some of them, it is still unclear whether they function as positive feedbacks (i.e., amplifying effects) or negative feedbacks (i.e., dampening effects) in the context of ice mass loss.*

**l. 43:** Explain why the air temperature increases (changes in air density, adiabatic lapse rate).

We will include the following specification:

*When the ice sheet surface melts, its surface elevation is lowered and the air temperature increases following the adiabatic lapse rate, thereby inducing even more melt.*

**l. 45:** I suggest adding a sentence or two explaining why it is not straightforward to include the SMB-elevation effect in ISMs (see also comment l. 72).

Thank you for this suggestion. We will mention this more explicitly here (and also around line 72, see below).

*However, it is not straightforward to represent this melt–elevation feedback in ISMs, since  the changing topography of the GrIS can in turn influence the (local) atmospheric circulation and induce changes in the precipitation pattern.*

**l. 52:** I suggest replacing "*landward*" with "*inland*" and "*contributes*" with "*increases*".

We will update "landward" with "inland", but prefer to keep the formulation "contribute to accumulation" to avoid confusion with the remaining part of the sentence:

*In many cases, the impact is more nuanced and varies regionally, since the precipitation is advected further  inland where it contributes to accumulation, but decreases near the margin (Fyke et al., 2018).*

**l. 58:** I suggest to explicitly explain how clouds affect the meltwater refreezing capacity (via surface energy balance/longwave radiation).

Thank you for this suggestion. We will update the text here, to explain this more explicitly:

*Other ice sheet–atmosphere interactions include changes in cloudiness,  as clouds can alter the surface energy balance and surface melt over the ice sheet in several ways, such as through alteration of incoming shortwave radiation,  increased longwave warming, or reduction of longwave cooling. It has been reported, for example, that clouds can reduce melt (in the ablation zone with low albedo) by blocking the incoming solar radiation, which is the main driver of melt here (Hofer et al., 2017). Vice versa however, it has been shown that clouds can enhance the meltwater runoff over the GrIS by one-third compared to clear skies, by reducing surface radiative cooling (mainly at night) and impeding the meltwater to refreeze (Van Tricht et al., 2016).*

**l. 72:** You could mention why it is not straightforward to represent ice-atmosphere interaction in standalone ISMs and why coupling is needed (see also comment l. 45).

Thank you for this suggestion. We agree to mention this more explicitly:

*Consequently, RCMs are needed to  represent such ice sheet–atmosphere interactions, or changes in (local) atmospheric circulation in response to the evolving ice sheet slopes and (local) topography at high resolution (Fettweis et al., 2020). Meanwhile, an ice sheet model (ISM) is needed to represent the ice dynamics and evolving ice sheet topography that are not considered by the RCM. As a result, efforts are currently emerging to couple  ISMs to RCMs. Yet, to date only two such coupled ISM–RCM simulations have been performed, on the centennial timescale (Le clec'h et al., 2019b; Delhasse et al., 2024).*

**Sect. 2.3:** If I understand correctly, during the initialization procedure you apply 1961-1990 SMB, but you also assume the ice sheet to resemble present-day conditions at the start of your forward simulations. How do you treat the period between 1991 and present-day? Is this period explicitly simulated? If so, which forcing is used? If not, how do you handle this gap and the observed mass-loss trend? Please clarify.

Thank you for highlighting the confusion regarding the use of the term "present-day". We used it several times to refer to the (observed) state of the ice sheet at the start of our simulations in 1990, but agree that we could have been more clear. We will be more specific throughout the text:

*Besides, the initialized system should represent the observed state as closely as possible for an accurate representation of ice sheet–atmosphere interactions and ensuing estimates of the GrIS contribution to sea level.*

*[… ] into an equilibrium state resembling  recent observations, as represented by the BedMachine v3 dataset (Morlighem et al., 2017).*

*For the ISM initialization, we combine a glacial–interglacial spin-up with a data assimilation technique, to capture the ice sheet response to past climatic conditions and represent its recently observed geometry as closely as possible.*

*However, as is often the case, the  ice sheet geometry obtained from the glacial–interglacial spin-up is slightly oversized and too thick near the margin […]*

*[…] to accurately represent the  observed ice thickness at the start […]*

*[…] and once run continuously for the period 1950 to 1990 on the recently observed topography (Morlighem et al., 2017) […]*

*As can be seen in Fig. 2, the differences between the initialized and observed  topography (Morlighem et al., 2017)[…]*

*Therefore, considering the exploratory nature of this study, the timescale and the accurately represented observed ice sheet geometry with low remaining model drift […]*

3.1  Initialized ice sheet topography

We explicitly simulate the period between 1990 and present-day by directly applying the IPSL-CM6A-LR forcing to MAR from 1990 onwards, the start of our coupled simulations.

**l. 96:** Mention that initialization should ideally reflect not only the present-day state but also current mass loss trends in response to recent 'historical' forcing (see also comment l. 118). See for example Rahlves et al. (2025) for including historical trends of ice mass loss in projections.

For consistency and to avoid model drift at the start of our model simulations, or at the transition between a historical period and the present-day, our coupled MAR-GISM model is initialized using IPSL-CM6A-LR forcing for the period 1960-1990 during which the ice sheet is assumed to have been in a steady-state. The coupled simulations thus start from this steady state in 1990 and are further forced by IPSL-CM6A-LR output. As such, a historical forcing does not apply here. Given our focus on ice sheet – atmosphere interactions and the millennial timescale of the simulations, we believe this is not crucial here. However, we agree to cite Rahlves et al. (2025) and mention the historical forcing nevertheless (see next comment regarding line 99).

**l. 99:** To me it is not clear what you mean by "*additional assumptions*". Please clarify.
What we meant is that since our simulations start from a steady-state in 1990, we do not handle the period between 1990 and 2025 separately but directly apply the IPSL-CM6A-LR forcing to MAR for this period (see also previous comment). We will be more specific:
*This way we do not need any  assumptions regarding the state of the ice sheet between 1990 and present, or a historical run preceding the future simulations as in Rahlves et al. (2025).*

**l. 110:** Consider explicitly showing the sliding law equation.
Thank you for this suggestion. Given the substantial length of the manuscript and the methodology section in particular, we prefer to refrain from adding the sliding law equation here, as it will not contribute to a better understanding of the presented research and

results. For interested readers, we have referred to the work describing the specific higher-order model version and sliding law in detail.

**I. 111:** Please specify what is meant by "*geometric input*" (bedrock topography and/or ice sheet geometry).

Thank you for mentioning that this was not explicitly stated. We will specify this:

*The geometric input for the model consists of the BedMachine v3 dataset* *for bedrock topography and ice sheet surface elevation or ice thickness* *(Morlighem et al., 2017),…*

**I. 116:** Define "*acceptable*" here. How is the acceptable discrepancy level determined, and what happens if it is exceeded?

What we mean is that the resolutions should not be too different, and it would not be ideal to run GISM on a much higher resolution. We were referring to common sense rather than a determined discrepancy level. We will explain this more clearly:

*Besides, ideally the GISM resolution should not be too high with respect to the MAR resolution, Additionally, it is important to maintain a reasonable level of discrepancy between both model resolutions throughout the coupled simulations, and to facilitate the efficient initialization of the ice sheet and coupled model into an equilibrium state resembling  recent observations, as represented by the BedMachine v3 dataset (Morlighem et al., 2017).*

**I. 118:** You describe the system as initialized "*into an equilibrium state resembling present-day observations.*" In reality, the ice sheet is not in equilibrium at present. How do you account for this (see also comment I. 96)?

We will correct this (see also answers to comments regarding Sect. 2.3 above, and lines 96 and 99):

*[…] to facilitate the efficient initialization of the ice sheet and coupled model into an equilibrium state resembling recent observations, as represented by the BedMachine v3 dataset (Morlighem et al., 2017).*

**Sect. 2.3:** Since the initialization is a complex but central part of your setup, I recommend a schematic figure/flowchart showing the main steps (spin-up, data assimilation, coupled initialization), inputs/forcings, parameters adjusted vs. fixed, and the MAR–GISM information exchange (including frequency).

Thank you for this suggestion. We agree that this would be useful, but we do not want to overload the current paper and instead plan to go into more detail in another manuscript in which we will focus more on the ice sheet response and shorter-term sea level contribution, for which this procedure and the initialized ice sheet state are (even) more relevant.

**I. 163:** Although you mention here that the isostatic bedrock adjustment is disabled after the spin up, I suggest reiterating this in Sect. 2.5 when describing the future simulations.

We will add the following sentence to Sect. 2.5.1:

*The ice sheet topography is annually updated in GISM for all three coupled simulations, but the*

*glacial isostatic bedrock adjustment is not considered (Sect. 2.3.1).*

**l. 165:** Add a brief explanation of how the positive degree-day approach works.
We will add the following explanation:
*With this approach, the amount of (energy available for) melt is determined based on the sum of mean daily temperatures above 0°C, i.e. the number of positive degree days. Accumulation is considered as the fraction of precipitation falling below a certain threshold, denoted as the snow fraction limit, here 1°C.*

**Sect. 2.3.2**: Reiterate here that GISM is forced with MAR SMB during data assimilation (currently mentioned earlier).
We suggest adding the following sentence:
*Throughout both steps, GISM is forced by the MAR SMB for our reference period (1961-1990).*

**l. 184**: Regarding peripheral glaciers: if data assimilation is not applied there, does this mean they are initially too large? How are they treated in sea-level contribution calculations? Do they make a difference? Please clarify.
Instead of performing the data assimilation for the peripheral glaciers, we adopt their observed ice thickness and apply SMB anomalies. We will better explain this and move it to the end of the paragraph for clarity:
*For the detached peripheral glaciers surrounding the ice sheet, identified based on the PROMICE aerophotogrammetric map of Greenland ice masses (Citterio and Ahlstrøm, 2013), the data assimilation was not performed. For these areas the observed ice thickness was adopted and SMB anomalies were applied throughout the coupled simulations.*

In section 3.2 it is mentioned that their contribution (total of 2.58 cm s.l.e.) is included in the reported sea level values, but we will repeat this when clarifying how the sea level contribution was calculated (see corresponding comment line 256).

**l. 190:** The phrase *"unvarying parameters"* is misleading. Suggest rephrasing as: *"Holding these parameters fixed is justified over short-term periods …"* Also specify what "short-term" vs. "long-term" means, and where your simulation timeframe fits.
Thank you for pointing out that this term can be misleading. To keep it short and given the explanation in the previous sentence, we suggest rephrasing it to "fixed" parameters:
*After the data assimilation, in absence of a better approach, the optimized basal sliding coefficient and enhancement factor are held constant throughout the coupled simulations, as is the geothermal heat flux. In general, these  fixed parameters are justifiable for short-term projections but inevitably become more contestable over the course of time (e.g. Goelzer et al., 2013; Le clec'h et al., 2019a, 2019b).*
The 'optimal' timing of such fixed parameters is already addressed in a separate paragraph in the discussion (Section 4.3), so we prefer not to repeat this here.
*In addition, it is unlikely that the optimized two-dimensional fields for the BSC and EF in the ISM remain valid over a period exceeding several hundred years. Over such timescales their values will likely be impacted by the changing overlying ice thickness and basal hydrological conditions (Leclec'h et al., 2019a, 2019b). On the other hand, the fixed ice temperature is not expected to substantially affect the presented results, as the rate of ice melt in our simulations exceeds the rate*

*at which ice temperature is altered through advection or the propagation of atmospheric temperature perturbations into the ice.*

**l. 195:** Be more precise: *"… computed on the ice-sheet topography as simulated by GISM."*
Thank you, we will implement this suggestion.

**Sect. 2.5:** What is the first year of the forward simulation (after the initialization procedure is complete)? I suppose it is 1991, but I think it would be good to mention it explicitly.
The first year of the coupled simulations is 1990. We will mention it explicitly.
*The first year of these simulations is 1990.*

**l. 216**: Does *"similar to …"* refer only to the zero-way run, or to all three? If the latter, consider: *"Similar to …, we consider three coupling types …"*
This could indeed be interpreted in two ways, so we agree with the suggestion for clarification:
*As illustrated in Fig. 1,  three different coupling types between the ice sheet and the RCM were considered: a so-called two-way (2wC), one-way (1wC), and a zero-way coupled (0wC) simulation.*

**l. 234:** Fig. 2 shows differences in ice thickness, not the initialized topography itself. Please rephrase.
Thank you for noticing this. We will remove the reference to Fig. 2 here.

**l. 241 ff.:** I find this sentence difficult to understand. I suggest rewriting for clarity.
We thank the reviewer for pointing out that these sentences are difficult to understand and suggest rephrasing the paragraph as follows:
*For all  three coupled simulations, after 2300 the GCM forcing for MAR is held constant until the year 3000 by randomly sampling the yearly IPSL-CM6A-LR output  from the period 2250 to 2300, during which the forcing temperature stabilizes (Fig. 3). In any case, as stressed by Delhasse et al. (2024), even when the temperature stabilizes  it is important to repeat the GCM forcing fields for MAR to prolong  the 1wC and 0wC simulations, rather than repeating the  MAR output (SMB and runoff), since the meltwater retention capacity and thereby the ablation area and SMB still require several decades to stabilize once warming stops.*

**Sect. 2:** In your methods section I am missing a description of how you are treating the ice-ocean boundary. Please include a brief explanation.
We will mention that the ice – ocean interactions are not regarded in our coupled ice sheet – atmosphere simulations:
*Lastly, given the ice sheet model resolution and the research focus on long-term ice sheet – atmosphere interactions, dynamic outlet glacier retreat is not explicitly considered here.*

**l. 252:** *"…since only SMB was extrapolated..."* What about runoff?
Thank you for noticing that indeed also the MAR runoff was extrapolated to the GISM

topography throughout the coupled simulations. We will add this to the sentence here.

**l. 256:** Clarify how sea-level contribution is calculated and what the reference period is.

We will add the following clarification at the beginning of section 2.6:

*For the conversion of ice volume change to sea level contribution since the start of our simulations in 1990, we consider only the ice above floatation (Goelzer et al., 2020a), include volume changes from the peripheral glaciers and ice caps, assume an ice density of 916.7 kg m$^{-3}$ and apply a conversion factor of 361.8 Gt per millimetre of barystatic sea level rise (Morlighem et al., 2017).*

**l. 264:** Specify *"fourth iteration"* of what.

We could indeed have been more specific here and suggest the following clarification:

*The ice sheet topography from the fourth MAR-GISM initialization iteration is used as the initial topography for the coupled simulations starting in 1990.*

**l. 305:** Instead of *"the first part of the simulation,"* write *"During the first X years …"*

We will rephrase this.

*During the first three centuries, the ice sheet – climate evolution in the simulations is dominated by the strong warming predicted by IPSL-CM6A-LR under the SSP5-8.5 scenario (Fig. 3).*

**l. 310:** I find the expression "compensation of differences" unclear. Differences of what? Suggest rephrasing to *"spatial compensation within SMB fields."* Also, I think the expressions "over- and underestimation" only work if there is a reference. Consider rewriting to "overestimated compared to…" and "underestimated compared to…".

Thank you for the suggestion, we will add some details:

*Especially around 2200 it becomes clear that a spatial compensation of differences within the SMB fields is at play, with ablation in the 1wC simulation being overestimated (by 1 to >3 m w.e.) compared to the 2wC simulation within 60 km from the ice sheet margins (i.e. two MAR grid cells), referred to hereafter as the ice-marginal zone, and slightly underestimated over the interior compared to the 2wC simulation (by generally 10 to 20 mm w.e.).*

**l. 335:** Refer to a figure if this is illustrated.

We will add a map showing these overall wind speed changes over the entire ice sheet for the 2wC versus the 1wC and 0wC simulations and refer to it in the text.

**l. 336:** Clarify causality: e.g. *"Changes in wind speed impact the surface energy balance, reducing runoff and lowering SMB (Fig. 6)."*

Thank you for this suggestion. Yet, since the figure and focus concern the SMB-elevation gradient rather than SMB in itself, we favor a small adjustment:

*As demonstrated by a transect from the ice sheet interior to the margin (Fig. 6), these changes in wind speed impact the surface energy balance, resulting  in lowered runoff and a reduced or even inversed SMB–elevation gradient in the ice-marginal zone in the 2wC simulation.*

**l. 375:** The causal link in this sentence is unclear. Please clarify how the density of the upper ice layers is used as an indicator of snowfall melt. In addition, it would be helpful to specify the actual density values of these layers to support the statement.

Thank you for indicating that this was not clear. The density of the upper snow/ice layers indicates whether the surface is covered by snow, firn or ice. We will be more specific:

*Another reason for the very similar ice sheet contribution to sea level by 2300, is the similar SMB evolution for all simulations up to 2300 on both the MAR and GISM grids (Fig. 9). Besides, as Figure 10 shows, by that time the ablation area already covers 100 % of the ice sheet area  in all simulations. This is also reflected in the density of the upper snow/ice layers up to 10 m depth in MAR that exceeds 910 kg m⁻³ by 2200,  indicate that  most of the snowfall melts before densifying to firn in all simulations (not shown here).*

**l. 403:** Remind the reader why low-lying margins retreat less rapidly in the 1wC simulation.

Thank you for addressing the confusion here. This was meant to be a mere observation to accompany the figure, rather than a process of which the reader should be reminded. We will rephrase the sentence to make this clearer.

*Between 2250 and 2650 the mean surface elevation is lower for the 1wC simulation compared to the 2wC simulation,  because the low-lying ice margin retreats less rapidly in this simulation after 2250. The same applies to the 0wC simulation between 2150 and 2250, yet after this time the higher parts of the ice sheet dwindle less rapidly than in the other two simulations.*

**l. 411:** Insert "loss" after mean surface elevation and change "stronger" to "strong".

We will apply these changes.

**l. 413:** Specify the year of *"At the time of this intensification …"*

Thank you for requesting this specification, we will clarify this:

*At the time of this intensification, around 2500, …*

**Sect. 3.6:** Consider merging 3.6.1 and 3.6.2 instead of using sub-subsections.

Thank you for this suggestion. However, these subsections describe different processes in the climate system, namely changes in clouds (3.6.1) and precipitation (3.6.2). The subsections and their titles are meant to indicate the observed processes and differentiate between them, so we prefer to keep the sections and subtitles as is.

**l. 455:** Clarify that *"increase"* is relative to the 1990–2019 reference period.

We will add this clarification at the end of the sentence:

*By 2300 the increase relative to the 1990-2019 reference period in both solid and liquid clouds together is 73 % higher and the LWD increase is 5.8 % higher for the 2wC simulation than for the 1wC and 0wC simulations, compared over the same retreating ice mask (Fig. 14).*

**l. 466:** Explain why this is specifically a result of the melt–elevation feedback.

**l. 467:** I could not fully follow how this is shown in the text. Please clarify.

We suggest the following rephrasing to clarify both sentences in the text:

*After 2300, regardless of the constant climate forcing, these trends continue in the 2wC simulation as a result of the thickening atmospheric column following the melt–elevation feedbackBy 3000 the amount of liquid and solid clouds increase by +529 % and +85 %, respectively, compared to the start of the simulations. Though the increments after 2300  are thus smaller than those induced by the climate forcing until 2300, they are still substantial.*

**l. 481:** Omit "important".

We will omit it.

**l. 489:** Rephrase: *"… and the increase in bare ice exposure."*
Thank you for the observant correction, we will rephrase it as suggested.

**l. 491:** Add: *"… and the associated increase in near-surface air temperature."*
We agree with the reviewer that as a result of the decreasing surface elevation, also the near-surface air temperature will increase. However, we prefer not to add the suggestion, since this might cause confusion, given that the GCM forcing temperature remains constant after 2300 throughout the prolonged simulations, in all simulations.

**l. 503:** Replace "snowfall" with "precipitation".
We will replace this.

**l. 504:** The two clauses are not logically connected. Consider rewriting to: *"However, as Fig. 14 demonstrates, not all snowfall is transformed into rainfall. Instead, decreasing snowfall also reduces total precipitation."*
Thank you for the suggestion. To our feeling however, the original sentence in essence conveys the same meaning as the proposed revision: namely the incomplete transformation of snowfall into rainfall and the consequent reduction in total precipitation. We will consider splitting the sentences for clarity:
*However, as Fig. 14 demonstrates, over time not all snowfall is persistently transformed into rainfall. Instead, the decrease in snowfall also leads to a  reduction in total precipitation.*

**l. 526:** Rephrase: *"Similar to Delhasse et al. (2024), during the first three centuries …"*
As the simulations by Delhasse et al. (2024) only span two centuries we suggest the following reformulation to improve the textual cohesion:
* At the beginning of our simulations,  the changes in the near surface wind speed have a negative feedback effect on ablation at the ice sheet margins, an effect that was also observed by Delhasse et al. (2024).*

**l. 528:** Rephrase: *"This results in a similar ice-sheet contribution to sea-level rise across our three coupled simulations up to 2300."*
Thank you for improving the readability with this suggestion, we will implement it.

**l. 542:** Add "per year" to "120 runoff days".
This paragraph will be rephrased according to a suggestion by Reviewer 1, yet we will update this for other instances throughout the text.

**l. 557:** When comparing to Aschwanden (2019) and Gregory (2020), stress that setups differ (coupled vs. uncoupled, forcing strategies). This distinction should be made clear throughout the discussion.
We will rewrite the sentence to make this more clear:
* In our two-way coupled simulation, the GrIS has almost entirely disappeared within the next millennium.  similar to earlier findings for a high-warming climate forcing though with a different experimental set-up and without full coupling, using a general circulation*

*model (Aschwanden et al., 2019; (Gregory et al., 2020) or corrected climatologies from an RCM (Aschwanden et al., 2019)..*

**l. 569:** The phrasing *"… in which the climate did not continue to warm up to 2200 …"* is not entirely clear. It would help to clarify whether the intended meaning is that previous studies assumed climate stabilization before 2200, or simply that warming was not extended in their scenarios. Consider rephrasing for precision.

Thank you for indicating that this was not entirely clear. We will rephrase the sentence to highlight that in the other studies the simulations were extended by assuming a stabilized climate after 2100.

*We find an underestimation of the sea level contribution of 10.41 % by 2150 or 14.41 % by 2200 when not including the melt–elevation feedback (i.e. 0wC simulation)., whichThis is somewhat higher than the ~10 %9.3 % by 2150 reported in Le clec'h et al. (2019b), and the 10.5 % by 2200 reported in Delhasse et al. (2024), since in these these three other studies in which the a stabilized climate was assumed to extend the simulations did not continue to warm beyond 2100up to 2200 (Edwards et al., 2014; Leclec'h et al., 2019b; Delhasse et al., 2024).*

**l. 576:** The sentence *"Though their applied horizontal model resolutions …"* is difficult to follow. I suggest rephrasing or splitting into shorter sentences to make the causality clearer.

We appreciate the suggestion and will split the sentences for clarity:

*This is consistent with the observations by Delhasse et al. (2024) for their MAR-PISM simulations with different large-scale forcing (CESM2). Though their applied On the one hand their horizontal model resolutions are similar to ours, namely 25 km for MAR and 4.5 km for PISM, and it such that it remains unclear to what extent these resolutions affect the strength of the observed negative wind feedback. Yet, on the other hand, the occurrence of the feedback and its poor reproduction by the offline extrapolation can thus be said to be independent of the coupled ISM and large-scale forcing.*

**Sect. 4.3:** Briefly discuss missing representation of ice–ocean interactions as a limitation. Outlet-glacier dynamics currently contribute significantly to GrIS mass loss and will likely remain important over several centuries. Also note feedbacks between SMB and dynamic processes (e.g. thinning reduces flux vs. ocean warming accelerates retreat). Even if not in the scope of your study, this deserves a short mention.

Thank you for pointing this out. We will expand the discussion accordingly and add the following paragraph to Sect. 4.3:

*Although ice – ocean interactions are not the focus here, it should be noted that ice discharge at outlet glacier fronts is expected to accelerate due to warming ocean temperatures and increased surface runoff (Slater et al., 2019), at least as long as the ice sheet remains in contact with the ocean, which is up to the 2340s in our 2wC simulation. Since dynamic outlet glacier retreat was not included here, the (rate of) ice mass loss in our coupled simulations might thus be somewhat too low during the first centuries. On the other hand, part of the ice at the ocean boundary might be removed by SMB-driven surface melting before it reaches the calving front, such that the present-day observed discharge rates cannot merely be extrapolated over time (Fürst et al., 2015) and the impact is thus likely more limited than such extrapolations would suggest.*

**l. 592:** The argumentative line of this paragraph is not clear to me. Can you rewrite to make your thought clearer?

We will remove this paragraph and incorporate the first sentence with the previous paragraph:

*The applied 30 km resolution for MAR is still relatively coarse, as for example the 30 – 60 km ice-marginal zone wherein the near-surface wind speed changes are observed spans only two MAR grid cells. However, even at this resolution it is clear from the presented simulations that the location, type and amount of precipitation are very strongly topographically controlled, highlighting the need for an RCM to accurately represent local atmospheric dynamics. In fact, the most accurate way to represent all ice sheet–atmosphere interactions would be to run both the ice sheet and RCM at the same horizontal resolution. However,* *. Running the RCM at the same resolution as the ISM thus*  *currently remains unreasonable in terms of computational resources for millennial-scale simulations, like the ones presented here.*

~~*Besides, as with all modelling research, the trade-off between the required computational resources, and the spatial as well as temporal resolution of the model (output) is inevitable. In this case, we were only able to identify an (accelerated) expansion of the area subjected to at least 120 runoff days, and the solar elevation angle as a limiting factor for further intensification of the melt-elevation feedback thanks to the daily resolution of the MAR output. Consequently, it can be argued that our attempt to balance these elements was fruitful.*~~

**l. 615:** Consider briefly mentioning expected effects of GrIS on large-scale circulation, and cite relevant studies (e.g. Haubner et al., 2025).

We agree to include more references, like the one suggested by the reviewer. However, given the already substantial length of the manuscript and the fact that findings related to large-scale atmospheric and oceanic changes following the GrIS decline vary considerably between studies (depending for example on model coupling strategies, model types and resolutions, surface properties,…) we prefer to refrain from discussing these potential effects.

*Lastly, our coupled model set-up does not represent the impact of the GrIS decline on the large-scale atmospheric and ocean circulation over the northern hemisphere (e.g. Davini et al., 2015; Madsen et al., 2022; Andernach et al., 2025; Haubner et al., 2025) but this is far beyond the scope of the study.*

**l. 644:** Since you start a new paragraph here, I suggest opening with *"Beyond the 2300 timescale …"*.

We agree and will update this.

**Figures**

**Fig. 2:** Omit the title (information is already in the caption) and add a description to the color bar (e.g. *"difference in ice thickness (m)"*). If possible, adjust the color scheme for detached peripheral glaciers to avoid overlap with the palette, where white indicates zero difference.
Thank you. We will apply these changes to improve the figure.

**Fig. 4:** Consider using a color bar for ice thickness (grey tones) as well as for the bedrock topography.
Ok, we will show a colour bar for the surface elevation (grey tones) and bedrock topography.

**Fig. 5:** Omit the title (repeated in the caption) and label all subpanels (a–e). In the upper panels, outline the selected regions in the same colors used for the corresponding SMB values in the lower panels, to improve readability. Add descriptions to the color bars (e.g. *"difference in SMB"*). In the caption, clarify whether the SMB evolution refers to area-mean SMB or integrated SMB, and specify whether *"mean SMB"* on the y-axis denotes areal mean or annual mean.
Thank you, we will update the figure and caption to improve its readability according to these suggestions.

**Fig. 6 & 7:** Consider reducing the number of points along the transects to make the plots less cluttered. As a matter of style, I suggest placing subpanel labels outside the plots.
Thank you for these suggestions.
We choose not to reduce the number of points along the transects, as they show the continuity of the changes taking place at the ice sheet margin, both through space and time. Reducing the number of points would interfere with this continuity, seemingly minimize the robustness of the findings, and/or further complicate the interpretation of the plots.
We have considered several label placings in- and outside the plots, yet to reduce the amount of whitespace and maximize the subplot sizes the current label positions at the top inside the graph proved to be best.

**Fig. 9, 10, 11, 12:** Consider merging these into one figure, since they share the same timescale.
Thank you for this suggestion. However, given that the figure order will slightly change according to a suggestion by Reviewer 1 and for reasons of readability we prefer to keep the figures separately.

**Fig. 13:** Remove the title; this information is already given in the caption.

Thank you, we will remove the title.

**Fig. 15:** Add descriptions to the color bars (e.g. *"total precipitation," "precipitation change,"* and *"rain fraction (increase)"*).
We will move the titles to the colour bar descriptions, as suggested.

**Technical Corrections**

**l. 56:** Include commas: *"It has, for example, been reported…"*
Thanks, we will.

**l. 130:** Change to: *"The main difference to …"*
We will rephrase the sentence.
*Compared to*  *version 3.11 described in Kittel et al. (2021), the main differences include corrections…*

**l. 197:** Avoid *"… until the differences … no longer change …"*. Instead, I suggest rephrasing as: *"until the differences become small/insignificant/approach zero"*
We will rephrase this:
*…until the differences in SMB and ice sheet topography between two consecutive initializations* *become insignificant*.

**l. 230:** (and other occasions): Replace *"on Fig. 2"* with *"in Fig. 2"*
Thank you, we will replace this.

**l. 427:** Replace *"a clear link with"* with *"a clear link to"*.
Ok.

**l. 722:** Add missing *"d"* in *"Land surface induced regional climate change…"*.
Thank you.

**References**

Below, we have inserted the references that we will add to our reference list based on the updates above.

Davini, P., von Hardenberg, J., Filippi, L. and Provenzale, A.: Impact of Greenland orography on the Atlantic Meridional Overturning Circulation. Geophys. Res. Lett., 42, 871–879, https://doi.org/10.1002/2014GL062668, 2015.

Goelzer, H., Coulon, V., Pattyn, F., de Boer, B., and van de Wal, R.: Brief communication: On calculating the sea-level contribution in marine ice-sheet models , The Cryosphere, 14, 833–840, https://doi.org/10.5194/tc-14-833-2020, 2020a.

Haubner, K., Goelzer, H., and Born, A.: Limited global effect of climate-Greenland ice sheet coupling in NorESM2 under a high-emission scenario, EGUsphere [preprint], https://doi.org/10.5194/egusphere-2024-3785, 2025.

Madsen, M. S., Yang, S., Aðalgeirsdóttir, G., Svendsen, S. H., Rodehacke, C. B., and Ringgaard, I. M.: The role of an interactive Greenland ice sheet in the coupled climate-ice sheet model EC-Earth-PISM, Clim. Dynam., 59, 1189–1211, https://doi.org/10.1007/s00382-022-06184-6, 2022.

Rahlves, C., Goelzer, H., Born, A., and Langebroek, P. M.: Historically consistent mass loss projections of the Greenland ice sheet, The Cryosphere, 19, 1205-1220, https://doi.org/10.5194/tc-19-1205-2025, 2025.

Slater, D. A., Straneo, F., Felikson, D., Little, C. M., Goelzer, H., Fettweis, X., and Holte, J.: Estimating Greenland tidewater glacier retreat driven by submarine melting, The Cryosphere, 13, 2489–2509, https://doi.org/10.5194/tc-13-2489-2019, 2019.

---

## Author Response (AR2)

Dear Editor,

We would like to thank you and both reviewers for taking the time to review our work and for all the valuable comments that have helped us to improve our manuscript.

Below, we address the final technical comment (in blue with track changes below).
In addition, we have updated our Data availability statement and provided the DOI to the now publicly available main output data.

**Technical correction:**
I do have one last minor comment that I would like to see addressed in the final paper. In Figure 14, the caption states: liquid clouds (water vapour path). The term in brackets should be (cloud) liquid water path, as the water vapour path includes the water in the atmosphere that is not condensed into a cloud instead of the total amount of water in the form of liquid cloud.

Thank you for your attention to detail. We have updated Figure 14 and the corresponding values in Section 3.6.1 to show the Liquid Water Path instead of Water Vapour Path; this does not alter our findings.

*By 2300 the increase relative to the 1990-2019 reference period in both solid and liquid clouds together is 38 % higher (not shown here) and the LWD increase is 5.8 % higher for the 2wC simulation than for the 1wC and 0wC simulations, compared over the same retreating ice mask (Fig. 14).*

*By 2300, as a result of the climate forcing, the amount of liquid clouds over the retreating ice sheet increases remarkably by +2017 to 313 % for the 1wC or 0wC and 2wC simulation, respectively, while the amount of solid clouds increases by only +55 to 58 % in all simulations.*

*By 3000 the amount of liquid and solid clouds increase by +4198 % and +85 %, respectively, compared to the start of the simulations.*

*Figure 14: Percentual increase in solid clouds (ice water path) and liquid clouds (liquid  path)  (a), percentual increase in longwave downward (LWD) radiation (b), and multiplication factor of runoff (c) with respect to the 30 year mean at the start of the simulations (1990–2019).*